# Extracellular vesicle engineering using a small scaffold protein

Wenjing Yan[1,2,13], Shizhi Wang [1,13] ✉, Haibin Hao[3,13], Hong Lin[4], Chen Wang [5], Shuqian Xie[1], Xing Zhang[1], Yiran Lu[1], Xin Ding[1], Xue Chen[1], Haohan Liu[1], Guiyuan Zhang[6], Dong Wei[6], ChangYan Ma[7], Cheng Tang[8], Xiuting Li[9], Bingjia Yu[9], Jing Hu[1], Zhongze Gu [6], Evan Yi-Wen Yu [10] ✉, Weiqin Li[3] ✉, Jiang Xia [11] ✉ & Hao Zhang [3,6,12] ✉

Extracellular vesicles (EVs) are promising drug-delivery vehicles owing to their biocompatibility and low immunogenicity. Genetic engineering of a membrane-bound EV-sorting scaffold protein empowers EVs by installing targeting moieties on the surface and enriching therapeutic cargo in the lumen. However, the choice of scaffold proteins with simple structures and short sequences is limited. Here, we conduct mass spectrometry-based proteomic studies and identify ENPP1 as a superior scaffold protein. Furthermore, we show that a truncated 144-amino acid variant, EN144, efficiently loads diverse therapeutic cargoes and outperforms conventional scaffolds. By fusing EN144 to the IL-6 decoy receptor gp130, we create engineered decoy EVs that potently inhibit inflammatory IL-6 trans-signaling. In mouse models, these EVs reduce inflammation, improve survival in sepsis, and, when targeted to cartilage, alleviate tissue damage in osteoarthritis. Our work establishes EN144 as a minimal, high-performance scaffold for EV engineering and demonstrates its broad therapeutic potential for inflammatory diseases.

Extracellular vesicles (EVs) have emerged as a promising avenue for targeted drug delivery. These nanoscale particles, naturally secreted by cells, facilitate vital intercellular transport processes[1]. Compared to other delivery modalities like nanoparticles and liposomes, EVs possess high biocompatibility, extended circulation half-lives, minimal toxicity, and the ability to traverse biological barriers[2]. EVs can encapsulate biomolecules via exogenous or endogenous loading. Exogenous loading, applicable to pre-isolated EVs, involves physical treatment (sonication, electroporation, chemical conjugation) and is suitable for small payloads like microRNAs and low-molecular-weight compounds. These procedures, however, may incur RNA precipitation and EV damage[3]. Alternatively, endogenous loading leverages the overexpression of biomolecules fused to EV-sorting proteins naturally expressed in EVs during their biogenesis within cells. This approach preserves the membrane integrity of EVs and enables the loading of large cargoes such as proteins. However, the loading efficacy relies on

[1]Key Laboratory of Environmental Medicine Engineering, Ministry of Education, School of Public Health, Southeast University, Nanjing, China. [2]School of Public Health, Shandong Second Medical University, Weifang, China. [3]Department of Critical Care Medicine, Jinling Hospital, Affiliated Hospital of Medical School, Nanjing University, Nanjing, China. [4]School of Medicine, Southeast University, Nanjing, China. [5]School of Pharmacy, Jiangsu University, Zhenjiang, China. [6]State Key Laboratory of Digital Medical Engineering, School of Biological Science and Medical Engineering, Southeast University, Nanjing, China. [7]Department of Medical Genetics, Nanjing Medical University, Nanjing, China. [8]Department of Orthopaedics, Nanjing First Hospital, Nanjing Medical University, Nanjing, China. [9]School of Public Health Administration, Jiangsu Health Vocational College, Nanjing, China. [10]Key Laboratory of Environmental Medicine and Engineering of the Ministry of Education, and Department of Epidemiology & Biostatistics, School of Public Health, Southeast University, Nanjing, China. [11]Department of Chemistry, The Chinese University of Hong Kong, Hong Kong SAR, China. [12]EVLiXiR Biotech Inc., Nanjing, China. [13]These authors contributed equally: Wenjing Yan, Shizhi Wang, Haibin Hao. ✉e-mail: shizhiwang2009@seu.edu.cn; evan.yu@maastrichtuniversity.nl; liweiqindr@nju.edu.cn; jiangxia@cuhk.edu.hk; scottzzhang09@gmail.com

the abundance of the sorting protein[4–6]. Also, some EV-sorting proteins may be expressed only in a subpopulation of the purified EVs, affecting the final product's homogeneity[7]. Therefore, EV-sorting proteins with a uniformly high expression level and robust engineerability as the scaffold protein are highly desired. Recent research employing fluorescent enzyme-based assays has uncovered that an EV-sorting protein, TSPAN2, surpasses CD63, exhibiting a higher signal level of luciferase reporter ThermoLuc within EVs (93 vs. 89%)[5]. Notably, these proteins belong to the multifunctional tetraspanin superfamily of multi-pass transmembrane proteins. As tetraspanins are known to be involved in the initiation, promotion, and progression of tumorigenesis[8], their use as scaffold proteins raises concerns. Significant endeavors have been made to identify alternative candidates, which resulted in the discovery of PTGFRN, BASP1, and TSPAN14 as highly potential scaffold proteins[6,9]. However, most of these studies employ a single method for EV extraction when screening for scaffold proteins, overlooking the inherent risk of potential protein contamination associated with any particular extraction technique. Therefore, a holistic approach to identifying scaffold proteins, ideally with minimalist structure and sequence, homogenous expression, and efficient cargo loading, is highly desirable.

On another note, inflammation is a complex medical condition that orchestrates diverse physiological and pathological responses to stimuli, primarily geared toward tissue repair and pathogen clearance[10]. Chronic inflammation may develop when the delicate balance is interrupted, and diseases such as sepsis and osteoarthritis (OA) may be triggered. Sepsis arises from a malfunction in the body's systemic inflammatory and immune response to infection, resulting in organ dysfunction[11]. The exaggerated inflammatory reaction, characterized by cytokine storms, has long been recognized as the primary driver of sepsis-related mortality[12]. OA, a degenerative disorder, is marked by the progressive loss of articular cartilage accompanied by chronic inflammation[13]. A crucial aspect in the development of OA is the disruption of cytokine homeostasis, where pro-inflammatory cytokines play a pivotal role. By activating catabolic enzymes like matrix metalloproteinase 13 (MMP13), these cytokines contribute to the degradation of cartilage and other vital intra-articular structures[14]. Aberrant or heightened production of cytokines, including interleukin 6 (IL-6), tumor necrosis factor-alpha (TNF-α), and IL-1β, potentially leads to organ failure and even death[15]. Currently, anti-cytokine treatment has emerged as a promising therapeutic approach for inflammatory diseases[16]. For example, monoclonal antibodies targeting inflammatory cytokines, such as Tocilizumab, have been approved as first-line therapies for treating rheumatoid arthritis.

Among all the cytokines, IL-6 is essential in the inflammatory cascade of sepsis and OA, thus raising the speculation of targeting the IL-6 signaling pathway for anti-inflammatory therapy against these diseases[17–19]. In the classic signaling mechanism, IL-6 binds to the membrane-bound IL-6 receptor, forms a complex that engages with IL6ST (gp130), and triggers intracellular signaling cascades[20]. Conversely, in trans-signaling, IL-6 combines with soluble IL-6 receptor (sIL-6R) to form an IL-6:sIL-6R complex, which subsequently interacts with gp130, activating the JAK signaling pathway[20]. Recent studies have revealed IL-6 as a dual-nature cytokine exhibiting both anti-inflammatory and pro-inflammatory effects through classic signaling and trans-signaling pathways, respectively[10]. However, antibodies targeting IL-6 or IL-6R, such as tocilizumab, cannot differentiate between these two pathways[21]. Thus, therapeutic approaches that preserve the anti-inflammatory properties of classic signaling while selectively inhibiting the pro-inflammatory trans-signaling of IL-6 are highly beneficial but challenging. One such strategy involves leveraging soluble gp130 (sgp130) shed from the plasma membrane. Sgp130 is a natural inhibitor of IL-6 trans-signaling by sequestering the IL-6:sIL-6R complex[22]. Recent studies report that TNFR1-enriched EVs function as a TNF-α decoy, inhibiting TNF-α-induced p38 MAPK

phosphorylation[23]. Decoy EVs with neutralizing functions are emerging therapeutic tools[24–26]. We then envision that EVs displaying gp130 may inhibit IL-6 trans-signaling and present an anti-inflammatory therapy for sepsis and OA treatment.

Here, we screen proteins expressed on the surface of EVs separated from Expi239F cells (Expi239F-EVs) using three distinct EV extraction strategies and proteomics analysis, and identify an effective EV-sorting protein, ENPP1. Notably, we identify a truncated version of ENPP1 (EN144), with only 144 amino acids, as one of the shortest and structurally simplest scaffold proteins. EVs carrying overexpressed EN144 (EN144-EVs) show improved loading of biomolecules in the lumen, high safety and stability, and preferential in vivo distribution to the liver and spleen in animals. EN144-EVs displaying gp130 in the lumen and the targeting motif on the surface simultaneously through engineered EN144 demonstrate a potent anti-inflammatory effect in animal models of sepsis and OA.

## Results

### Identification of EV-sorting proteins in Expi293F-EVs

Because of their robust growth in chemically defined serum-free media, Expi293F cells (derivatives of the 293 cell line) are an ideal chassis for large-scale, controlled production of biologics that meet the quality requirements of clinical applications[27]. We purified EVs from Expi293F cells, designated as Expi293F-EVs, by following the established protocol[28]. Expi293F-EVs exhibited cup-shaped double-membrane structures, typical of mammalian-cell-derived EVs, with a size distribution of $67 \pm 30$ nm (Supplementary Fig. 1a, b). EV-specific marker proteins, such as Hsp70, TSG101, and CD9, but not the endoplasmic reticulum marker Calnexin or the core component of nucleosome (Histone H3), were found in the Expi293F-EVs preparation (Supplementary Fig. 1c), suggesting the absence of nuclear fragments and ER-derived contaminants. We employed three different and independent strategies to isolate EVs: ultracentrifugation, magnetic bead-based EVtrap technology[29], and a purification process integrating tangential flow industrial extraction[30] with size exclusion chromatography. Next, we conducted comprehensive proteomic analyses on samples from the three isolation methods. Different sources of EVs were used to mitigate any possible bias or false hits of each isolation method (Fig. 1a). A panel of proteins was selected as candidates for EV-sorting proteins and divided into four categories: (1) abundant proteins in Expi293F-EVs, (2) membrane proteins, and (3) proteins that have not been reported as EV-sorting proteins[5,6,9,31], and (4) proteins involved in EV biogenesis and release, including Rab7a, VTA1, Syntenin-1, and IST1. A total of 15 candidate proteins were identified, including ENPP1, Rab7a, STX7, CXADR, EPCAM, TFRC, VTA1, Syntenin-1, SNAP23, AT1A1, STX4, PDL1, VAMP2, CD55, and IST1 (Supplementary Table 4). Notably, among the list, Rab7a is not a membrane-bound protein.

Next, we assess the EV-sorting capability of the 15 candidate proteins by fusing them with the green fluorescent protein EGFP (Supplementary Table 4). Briefly, EGFP was appended to either the N- or C-terminus of the candidates, stable Expi293F cell lines were established using plasmids encoding EGFP fusion, and EVs carrying the fluorescent candidate proteins were isolated and characterized (Fig. 1b, Supplementary Fig. 2a). All the engineered EVs showed particle sizes of 50–150 nm, suggesting that overexpressing the candidate proteins did not significantly affect the EV packaging (Supplementary Fig. 2b). Based on Western blot (WB) analysis, EVs from Expi293F cells overexpressing ENPP1, Rab7a, STX7, CXADR, SNAP23, PDL1, EPCAM, and VTA1 exhibited elevated EGFP expression levels compared to other groups, surpassing that of EVs derived from Expi293F cells overexpressing PTGFRN (named PTGFRN-EVs, Fig. 1b, Supplementary Fig. 3). Among these positive hits, EVs from Expi293F cells overexpressing ENPP1 (ENPP1-EVs) and Rab7a (Rab7a-EVs) demonstrated the highest EGFP levels. Conversely, EVs derived from Expi293F cells

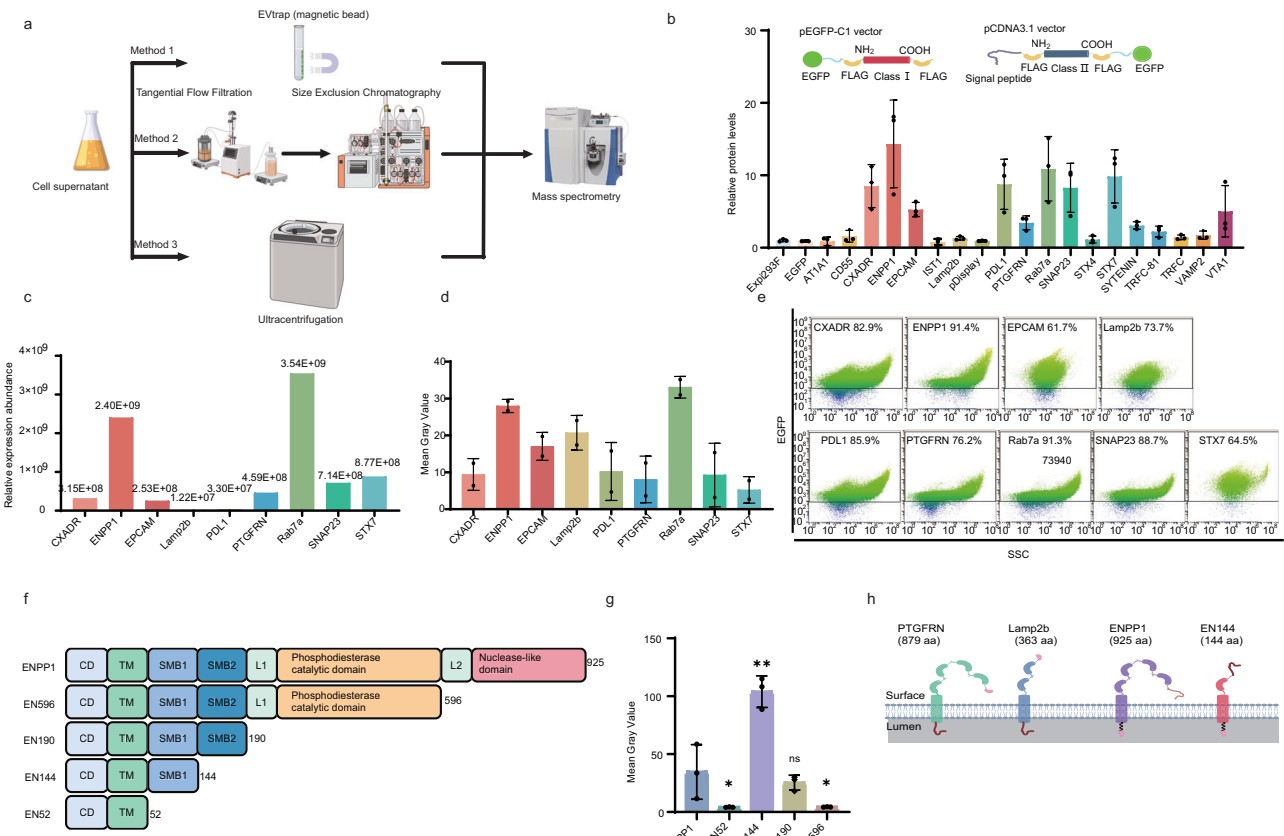

**Fig. 1 | Screening for EV-sorting proteins and identification of EN144 as the smallest EV-specific scaffold protein. a** Schematic illustration showing the workflow. Briefly, three methods (ultracentrifugation, magnetic bead-based EVtrap technology, and a purification process integrating tangential flow industrial extraction) were used to purify Expi293F-EVs, which were subsequently subjected to LC-MS/MS analysis. This figure was created using MedPeer (medpeer.cn). **b** Schematic illustration of the sorting proteins and their expression levels quantified by Western blot analysis. Class I sorting proteins include STX4, ENPP1, TFRC, TFRC-81 (the transmembrane sequence of TFRC), VAMP2, and CD55, which were inserted into the pEGFP-C1 vector. Class II sorting proteins include EPCAM, AT1A1, Syntenin-1, CXADR, PDL1, STX7, IST1, SNAP23, VTA1, PTGFRN, and Lamp2b, inserted into the pCDNA3.1-EGFP vector. EV-sorting protein Rab7a, VTA1, Syntenin-1, and IST1 are also included. EGFP levels of the overexpressed sorting proteins in purified EVs were quantified by Western blotting. Briefly, EVs were collected from transfected cells for Western blot analysis ($3.0 \times 10^9$ particles; Fig. S3) and quantification.

$n = 3$ independent experiments. **c** Relative quantification of EGFP protein levels in each group of EVs based on mass spectrometry. **d** Semi-quantitative analysis of EGFP protein in engineered EVs. The relative load levels of EGFP in engineered EVs were analyzed semi-quantitatively based on the band grayscale values shown in Fig. S4. $n = 2$ independent experiments. **e** Flow cytometric analysis showing the percentages of EGFP-positive EVs among the isolated EV populations. **f** Schematic illustration showing truncated ENPP1 variants. CD cytoplasmic domain, TM transmembrane domain, SMB somatomedin B-like domain. **g** Semi-quantitative analysis of FLAG protein in engineered EVs, as determined by band intensity measurements from Fig. S6B. $n = 3$ independent experiments. **h** Schematic illustration showing the comparison of the architecture of the selected scaffold proteins. This figure was created using MedPeer (medpeer.cn). Two-tailed $t$-tests were used to assess statistical significance between two groups, and the data are presented as mean ± SD. Statistical significance indicated as *$P < 0.05$, **$P < 0.01$. Source data and exact $p$ value are provided as a Source data file.

overexpressing TFRC, TFRC-81 (the transmembrane sequence of TFRC), Syntenin-1, AT1A1, STX4, VAMP2, CD55, IST1, pDisplay, and Lamp2b showed relatively low EGFP signals (Fig. 1b, Supplementary Fig. 3). Additionally, EVs from Expi293F overexpressing unfused EGFP failed to yield detectable protein bands, indicating that EGFP protein does not spontaneously sort into EVs without fusion to an EV-sorting protein (Fig. 1b, Supplementary Fig. 3).

Next, the seven top-performing candidate proteins alongside two control sorting proteins, PTGFRN and Lamp2b, were selected for further assessment using mass spectrometry. When analyzed from an equal number of EV particles, ENPP1-EVs and Rab7a-EVs exhibited the highest raw EGFP label-free quantification (LFQ) signals, 5.23-fold and 12.07-fold higher than that of PTGFRN-EVs, respectively (Fig. 1c). Semi-quantitative protein analysis indicated that the levels of EGFP protein loaded into ENPP1-EVs and Rab7a-EVs were approximately three- and four-fold higher, respectively, than those in PTGFRN-EVs (Fig. 1d, Supplementary Fig. 4). Nanoscale flow cytometry confirmed that ENPP1 and Rab7a overexpression led to abundant and uniform EGFP distribution in EVs, with a loading capacity exceeding 90% (Fig. 1e).

These findings indicate that ENPP1 and Rab7a can serve as potent EV-sorting proteins to facilitate efficient cargo loading.

Subsequently, we assess the entry of ENPP1-EVs, Rab7a-EVs, and PTGFRN-EVs into Expi293F cells. Although Rab7a-EVs exhibited the highest EGFP loading (Fig. 1d), they were internalized less efficiently than ENPP1-EVs and PTGFRN-EVs (Supplementary Fig. 5a). Therefore, we focused on ENPP1-EVs in the following experiment and explored ENPP1 as a potential scaffold protein for EV engineering (Rab7b, in contrast, is not membrane-bound, so it is an EV-sorting protein, but not a scaffold protein). WB analysis revealed a time-dependent increase in the entry of ENPP1-EVs into recipient cells, which saturated at 18 h (Supplementary Fig. 5b, c). As smaller and simpler EV-sorting scaffold proteins (<50 kDa) facilitate cargo engineering[9], we next aim to test ENPP1 truncations by cutting at the domain boundaries (Fig. 1f). Transfection efficiencies of truncated and full-length ENPP1 plasmids ranged from 38% to 58% (Supplementary Fig. 6a). EVs expressing the 144-amino-acid truncation, EN144 (EN144-EVs), exhibited a marked enhancement in cargo loading, a three-fold increase over ENPP1-EVs (Fig. 1g, Supplementary Fig. 6b). In addition to the Expi293F cell line,

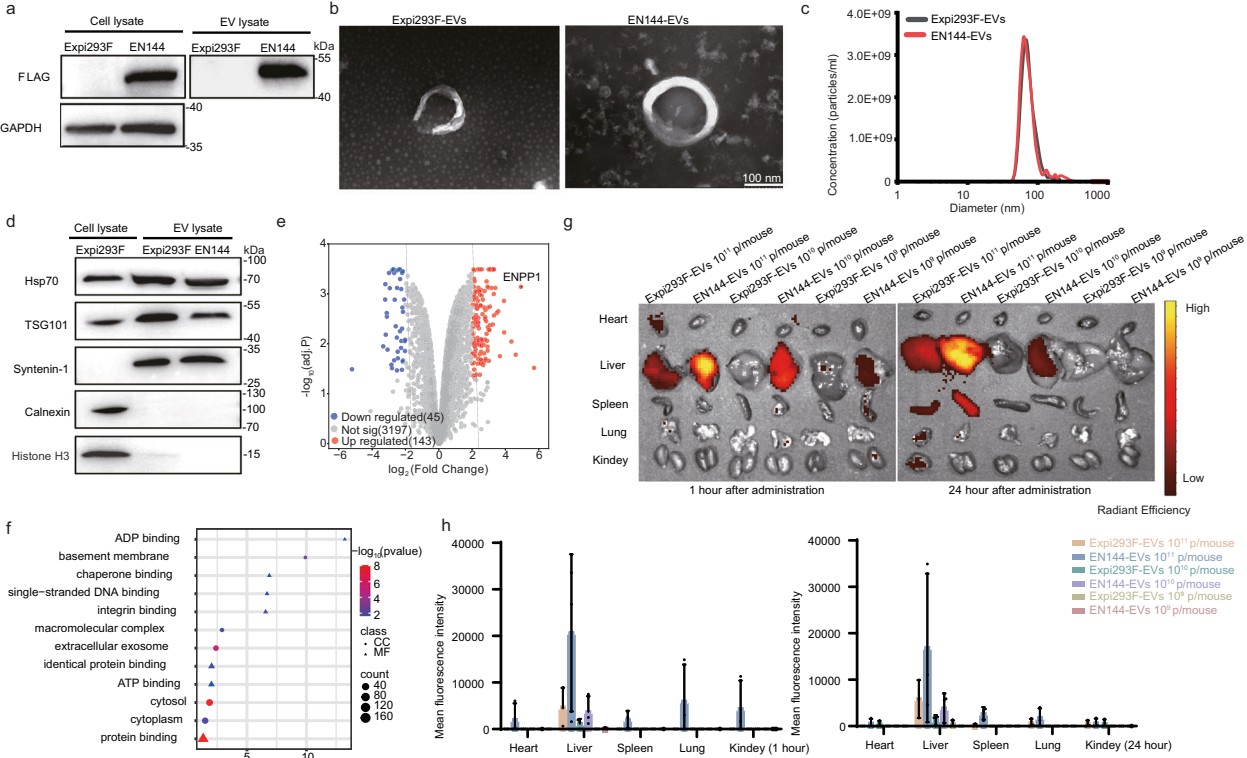

**Fig. 2 | Proteomic analysis and in vivo distribution of EN144-EVs. a** Western blot analysis showing that EN144-EVs enrich the overexpressed EN144 based on the FLAG tag. **b** Representative TEM images of Expi293F-EVs and EN144-EVs. Scale bar: 100 nm. **c** Particle size distribution of Expi293F-EVs and EN144-EVs based on Resistive Pulse Sensing (RPS) analysis. **d** Western blot analysis showing the expression of exosome markers Hsp70, TSG101, and syntenin-1 in EN144-EVs and Expi293F-EVs, but not calnexin and histone H3. Each sample contains 10 μg of cell lysate or $3.0 \times 10^9$ EV particles. **e** The volcano plot showing 188 differentially expressed proteins (DEPs) in EN144-EVs compared to Expi293F-EVs. (Selection threshold: $|\log_2(FC)| > 2$, adj.$P < 0.05$, with 143 upregulated DEPs in red dots and 45 downregulated DEPs in blue). The differential expression analysis was performed using a two-sided moderated $t$-test within the limma package in R, with Benjamini-Hochberg FDR adjustment for multiple comparisons. **f** Gene Ontology (GO) and Kyoto Encyclopedia of Genes and Genomes (KEGG) pathway analysis (adj.$P < 0.05$) of the 188 DEPs was performed using the DAVID database (https://davidbioinformatics.nih.gov/). CC cellular component, MF molecular function. **g** Representative fluorescent images of tissues showing the distribution of intra-venously injected DiR-labeled Expi293F-EVs and EN144-EVs at different doses ($1.0 \times 10^9$, $1.0 \times 10^{10}$, or $1.0 \times 10^{11}$ particles) in C57BL/6 mice post-injection after 1 h and 24 h. **h** Quantification of EVs based on the fluorescence signal of DiR. The data are presented as mean ± SD. Source data are provided as a Source data file.

other adherent cell lines are frequently employed as sources of EVs. Based on the ratio of EN144 in EVs to all the EN144 in the cell lysates (the total expression levels varied in different cells), we found that a similar percentage of EN144 was enriched in EVs when we over-expressed EN144 in cell lines, including 293T cells, synovial fluid-derived mesenchymal stem cells (SF-MSCs), HeLa cells, and HcerEpic cells. These results demonstrated that EN144 exhibited comparable sorting efficiency in various cell types (Supplementary Fig. 7a, b). In contrast to PTGFRN, Lamp2b, and ENPP1, EN144 has only 144 amino acids with a significantly simpler transmembrane architecture (Fig. 1h). Taken together, based on multiple screening strategies, we identified a truncated version of ENPP1, comprising only 144 amino acids, as one of the most efficient EV-sorting proteins and, to our knowledge, the smallest and simplest EV-specific scaffold protein.

### Biocompatibility and bioavailability test

Next, we characterized the physical properties, toxicity, and in vivo distribution of EN144-EVs. WB results indicated EN144 was highly expressed in the lysates of the EN144-transfected Expi293F and EN144-EVs (Fig. 2a). TEM imaging confirmed their spherical morphology and lipid bilayer structure (Fig. 2b). Resistive Pulse Sensing (RPS) analysis revealed that EN144-EVs have similar sizes as Expi293F-EVs (72 vs. 66 nm, Fig. 2c). Both EN144-EVs and Expi293F-EVs expressed char-acteristic biomarkers Hsp70, TSG101, and Syntenin-1 but not Calnexin or Histone H3 (Fig. 2d). LC-MS/MS-based proteomics was performed to

assess the impact of EN144 on EV protein profiles, revealing 143 upregulated and 45 downregulated genes in EN144-EVs compared to Expi293F-EVs ($|\log_2(FC)| > 2$, adj.$P < 0.05$; Fig. 2e). Gene Ontology (GO) and Kyoto Encyclopedia of Genes and Genomes (KEGG) pathway analyses revealed enrichment in processes related to extracellular exosomes, protein binding, and integrin binding, suggesting minimal disruption to normal physiological functions (adj.$P < 0.05$; Fig. 2f). Notably, 13 proteins (OPRD1, GFER, KCNC2, CYP1B1, ABTB2, PTPN13, TAS2R40, PRKN, TNF, PINK1, AHR, PPP1R9A, and JOX2) associated with cellular response to toxic substances (GO:0097237) were absent in both EV types, suggesting that EN144-EVs may have low toxicity.

The cytotoxicity of EN144-EVs was next evaluated in several cell lines (RAW 264.7, 293T, SF-MSC, and rat primary chondrocytes) using varying concentrations of EVs (ranging from $5.0 \times 10^8$ to $5.0 \times 10^{10}$ particles/mL). After 24 or 48 h, LDH release, ROS generation, cell via-bility, apoptosis, and migration were evaluated. Several data points showed signs of cell disruption: for example, the LDH levels in 293T cells were significantly elevated at $5.0 \times 10^8$ particles/mL EN144-EVs (143 vs. 100%, $P = 0.047$, Supplementary Fig. 8a), and a modest increase in apoptosis was observed in SF-MSCs at $1.0 \times 10^{10}$ particles/mL (3.88 vs. 2.90%, $P = 0.004$, Supplementary Fig. 8b). However, these signs did not persist at higher doses (Supplementary Fig. 8a, b). Fur-thermore, high concentrations of EN144-EVs did not significantly decrease cell viability (Supplementary Fig. 8c, d), and ROS levels remained unaffected across all concentrations (Supplementary Fig.

8e). Cell migration was minimally affected by EN144-EVs (Supplementary Fig. 8f). Collectively, these findings suggest that intracellular uptake of EN144-EVs will not cause significant cytotoxicity to the cells we have tested.

Next, we tracked the distribution of EVs in vivo following tail vein injection. We first confirmed that EVs were fluorescently labeled following the DiR-labeling procedure, and the free DiR dye was completely removed. DiR-labeled Expi293F-EVs and EN144-EVs were injected via the tail vein into the mouse circulation system. Fluorescent imaging using the in vivo imaging system (IVIS) showed rapid accumulation of both Expi293F-EVs and EN144-EVs in the liver. Intriguingly, we observed a greater accumulation of EN144-EVs in the liver and spleen compared to Expi293F-EVs (Fig. 2g, h). The reasons, however, are unknown.

An in-depth evaluation of in vivo toxicity was conducted following single intravenous administrations of varying doses of Expi293F-EVs and EN144-EVs ($1.0 \times 10^{11}$, $1.0 \times 10^{10}$, and $1.0 \times 10^{9}$ particles/animal) into mice. Hematological assessments 24 h post-treatment revealed minimal impact on blood parameters, including the levels of white blood cells, red blood cells (RBC), hemoglobin (HGB), granulocytes (Gran), lymphocytes (Lymph), and monocytes (Mon). A modest elevation in RBC percentage (9.980 vs. 4.410, $P = 0.046$) was observed in Expi293F-EVs-treated mice compared to the PBS control group (Supplementary Fig. 9a–c). Both groups' histopathological examinations of the heart, liver, spleen, lungs, and kidneys revealed no treatment-related abnormalities (Supplementary Fig. 10a). Similarly, blood biochemical markers (ALT, ALP, AST, BUN, CREA, and LDH) remained unchanged after the treatment (Supplementary Fig. 9d–f). Next, we investigated the in vivo toxicity of repeated administration of EN144-EVs. Briefly, mice were dosed with the highest concentration thrice weekly (Supplementary Fig. 9g). Monitoring body weight and organ coefficients revealed good overall health, with only a slight increase in the spleen coefficient in the EN144-EVs group compared to PBS (0.006 vs. 0.005, $P = 0.028$) (Supplementary Fig. 9h, i). Subsequent hematological, biochemical, and histopathological evaluations detected no significant differences between EN144-EVs-treated and PBS-treated animals (Supplementary Figs. 9j, k and 10b). Taken together, these data confirm that EN144-EVs are generally safe for systemic injections.

## Surface and luminal cargo loading by EN144 engineering

Next, we installed a chondrocyte-affinity peptide (CAP, DWRVIIPPPPRPSA) on the surface of EVs by engineering the scaffold proteins to achieve chondrocyte targeting for the treatment of OA[32]. Four plasmids were constructed by fusing CAP with different EV scaffold proteins, yielding ENPP1-CAP, EN144-CAP, Lamp2b-CAP, and PTGFRN-CAP (Fig. 3a). Subsequently, chondrocyte-targeting EVs, EN144-EV$^{CAP}$, ENPP1-EV$^{CAP}$, PTGFRN-EV$^{CAP}$, and Lamp2b-EV$^{CAP}$, were isolated from transfected cells. These engineered EVs showed sizes ranging from 50 to 150 nm (Supplementary Fig. 11a, b). To enable direct comparison among different EVs, EV concentrations were normalized based on particle counts determined by RPS. Primary rat chondrocytes were incubated with EVs for 15 min or 2 h to evaluate cellular uptake efficiency. Quantitative analysis revealed enhanced cellular uptake of PTGFRN-EV$^{CAP}$ (80.38 ± 0.79), ENPP1-EV$^{CAP}$ (107.6 ± 19.78), and EN144-EV$^{CAP}$ (101.6 ± 15.32) in the 2-h incubation group, with ENPP1-EV$^{CAP}$ and EN144-EV$^{CAP}$ exhibiting the highest fluorescence intensities (Fig. 3b). Notably, after 24-h incubation, the cellular uptake rates of ENPP1-EV$^{CAP}$ and EN144-EV$^{CAP}$ reached approximately 90%, significantly surpassing those of PTGFRN-EV$^{CAP}$ (87.97% ± 0.38%), Lamp2b-EV$^{CAP}$ (69.37% ± 0.97%), and Expi293F-EV (47.93% ± 1.27%, Fig. 3c). To determine whether the observed ENPP1-EV$^{CAP}$/chondrocyte interaction was mediated by CAP-specific binding, primary chondrocytes were pre-treated with CAP peptide (1, 3, 10 µg) for 2 h to block potential binding sites. Interestingly, pretreatment with 10 µg CAP peptide significantly reduced the fluorescence intensity of primary chondrocytes by 45.6% compared to untreated controls. This dose-dependent blockade indicates that CAP peptide mediates the uptake of EN144-EV$^{CAP}$ by primary chondrocytes (Supplementary Fig. 12).

Next, we compared the loading efficiency of mRNA cargo into different EVs. We devised a two-component system leveraging the MS2 phage shell protein (MS2)-LOOP pair and the archaeal ribosomal protein (L7Ae)-C/D Box pair. MS2 is known to bind LOOP motifs, while L7Ae recognizes the C/D Box sequence. By fusing EN144 with MS2 or L7Ae and inserting LOOP or C/D Box sequences into EGFP mRNA's 3'-UTR, we achieved selective mRNA loading to EVs. A control group with mRNA expressing free, non-fused EGFP was included to provide an estimation of the random, passive mRNA encapsulation level in EVs (Fig. 3d). qRT-PCR analysis revealed that compared to EVs with the MS2-LOOP pair, the EV group with the L7Ae-C/D Box pair contains the highest EGFP mRNA content (Fig. 3e, Supplementary Fig. 13a). Thus, we selected the L7Ae-C/D Box as a guiding cue for enriching mRNA in EVs. In addition, we compared the EGFP mRNA loading capacity of EVs with EN144, Lamp2b, and PTGFRN as the scaffold by constructing EN144-EV$^{EGFP}$, Lamp2b-EV$^{EGFP}$, and PTGFRN-EV$^{EGFP}$. EN144-EV$^{EGFP}$ demonstrated the highest EGFP mRNA content compared to Lamp2b-EV$^{EGFP}$ and PTGFRN-EV$^{EGFP}$ (Fig. 3f, g, Supplementary Fig. 13b, c). Cellular uptake studies revealed a peak uptake at 18 h for EN144-EV$^{EGFP}$, achieving a higher level of intracellular EGFP mRNA than other EV preparations (Supplementary Fig. 13d, e). To determine whether the observed EGFP signal originated from active mRNA translation rather than from pre-existing protein passively carried in EVs, LC-MS/MS was used to quantify EGFP levels in three types of EVs (Expi293F-EVs, EGFP-C/D box EVs, and EN144-EV$^{EGFP}$) as well as in recipient cells after co-culture (Supplementary Fig. 13f, g). The results demonstrated that passive loading of EGFP protein was detected in both the EN144-EV$^{EGFP}$ and EGFP-C/D box EVs groups (Supplementary Fig. 13f). However, EGFP expression was significantly elevated in recipient cells co-cultured with EN144-EV$^{EGFP}$ (Supplementary Fig. 13g). These findings indicate that the EGFP signal in the recipient cells primarily resulted from active translation of EGFP mRNA delivered by EN144-EV$^{EGFP}$, rather than from pre-packaged EGFP protein in the EVs. Collectively, these results demonstrate that EN144-EV$^{EGFP}$, utilizing the L7Ae-C/D box system, efficiently loads EGFP mRNA in EVs and delivers the mRNA into cells.

For protein loading within the lumen of EVs, one can directly fuse the protein of interest with the scaffold through a covalent bond or assemble the protein of interest with the scaffold protein via a specific, non-covalent interaction mediated by a pair of interacting domains. We systematically evaluated multiple protein heterodimerization strategies for EV cargo loading, including a GFP-GFP nanobody interacting pair[33], a light-dependent CIBN-CRY2 interacting pair[34], an NbALFA$^{PE}$-ALFA interacting pair[35], and a PRB-PDAR interacting pair[36]. We fused one binding partner, GFP, CIBN, PDAR, and NbALFA$^{PE}$, with EN144, and Cas9 with the corresponding binding partner, Nanobody, CRY2, PRB, and ALFA, respectively (Fig. 3h). These four plasmids were then transiently transfected into Expi293F cells. As expected, sgRNAs were detected in transfected cells and their derived EVs using qRT-PCR (Fig. 3i, Supplementary Fig. 14a). The qRT-PCR results clearly showed that the GFP-Nanobody system emerged as the most efficient for loading sgRNA into EVs compared to other dimerization systems (Fig. 3i). Next, we evaluated the capacity of engineered EVs to deliver sgRNA into cells. The GFP-Nanobody system exhibited the greatest uptake efficiency, with intracellular levels of exogenous sgRNA exceeding those delivered by CIBN-CRY2, PRB-PDAR, and NbALFA$^{PE}$-ALFA systems by approximately 4.32-, 5.57-, and 18.71-fold, respectively (Fig. 3j). To further validate that Cas9 was loaded into EVs, we verified Cas9 expression in cells and secreted EVs (EN144-EV$^{M-CRISPR/Cas9}$) using the GFP-Nanobody system (Supplementary Fig. 14b). NTA results revealed that EN144-EV$^{M-CRISPR/Cas9}$ and Expi293F-EVs had distinct diameters, which peaked at approximately 77 and 66 nm, respectively (Supplementary Fig. 14c). To confirm the function of CRISPR/Cas9 in

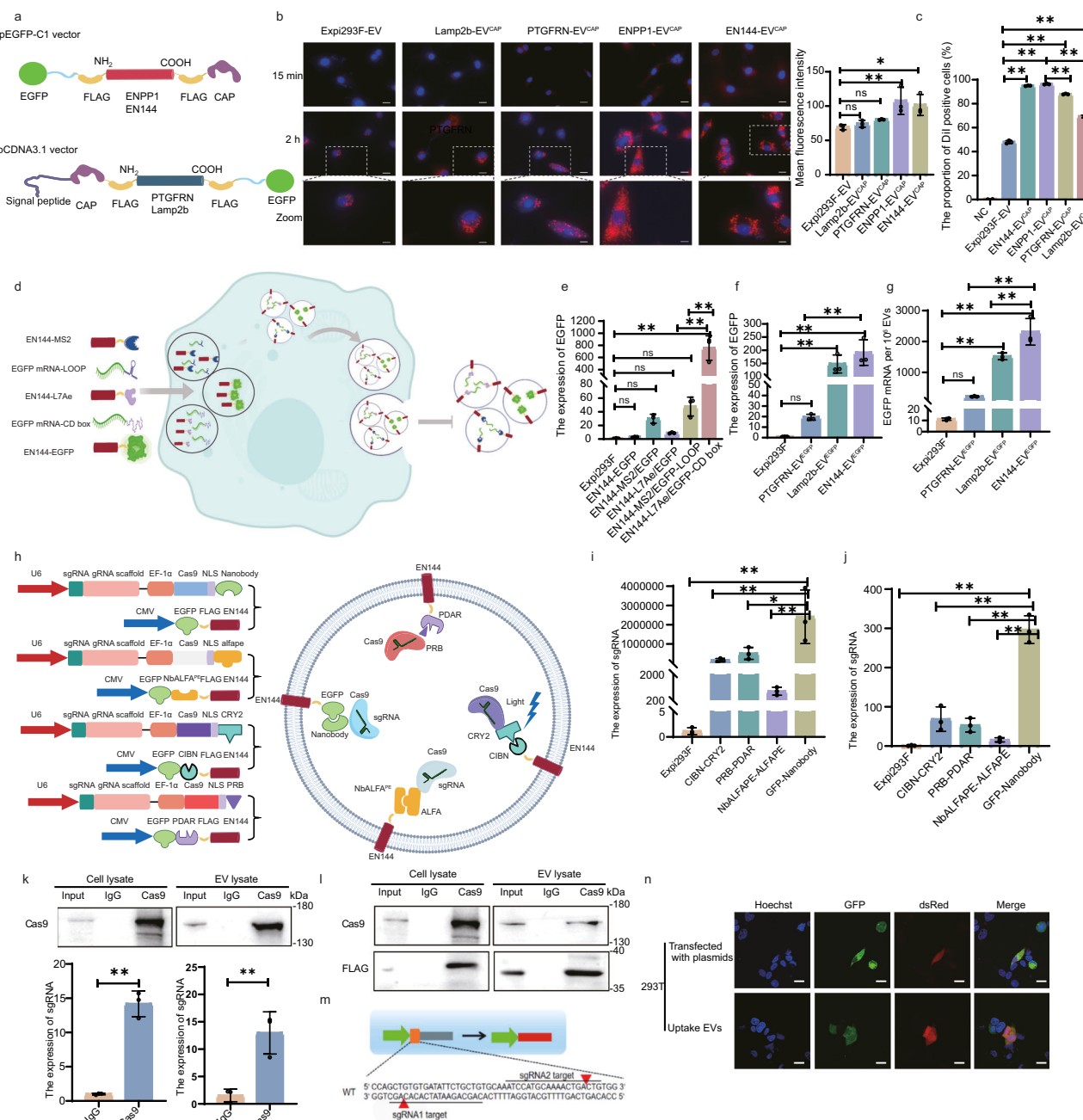

**Fig. 3 | Surface and luminal cargo loading on EN144-EVs. a** Schematic of constructing chondrocyte-targeting EVs by fusing the CAP peptide to Lamp2b or PTGFRN's N-terminus, or to ENPP1 or EN144's C-terminus. Confocal microscopy (**b**) and flow cytometry (**c**) show ENPP1-EV$^{CAP}$ and EN144-EV$^{EGFP}$ are internalized more efficiently and rapidly than other EV types. Scale bars: 100 μm; 50 μm (Zoomed images). $n$ = 3 independent experiments. **d** Schematics show two self-assembly systems (MS2-LOOP and L7Ae-C/D Box) for loading EGFP mRNA. **e** qRT-PCR quantification of EGFP mRNA loaded into EVs. Groups include: unmodified Expi293F-EVs; EVs from cells overexpressing EN144-EGFP (EN144-EGFP); and EVs from cells co-transfected with various plasmid pairs for the MS2-LOOP (EN144-MS2/EGFP, EN144-MS2/EGFP-LOOP) or L7Ae-C/D box (EN144-L7Ae/EGFP, EN144-L7Ae/EGFP-CD box) loading systems. qRT-PCR quantification of EGFP mRNA (**f**) and estimation of mRNA copy numbers (**g**) per EV particle. Each sample has 1.0 × 10$^{10}$ EV particles. **h** Schematic of CRISPR/Cas9 loading into EVs via self-assembling protein pairs. **i** Levels in EVs loaded via the four protein-interaction pairs. **j** Levels in

recipient 293T cells after 24 h exposure to the corresponding CRISPR/Cas9-harboring EVs. **k** IP using anti-Cas9 antibody confirmed the presence of Cas9 and its associated sgRNA in both producer cell lysates and purified EN144-EVs (using the GFP-Nanobody pair). IgG IP served as a control. **l** Western blot for FLAG-tagged EN144 after Cas9-IP further verified the specific interaction between the CRISPR/Cas9 complex and the engineered EV membrane protein. **m** Schematic of a stop-dsRed reporter plasmid, where successful CRISPR/Cas9 delivery removes a translational stop cassette to activate red fluorescence. **n** Fluorescence microscopy showed that EVs loaded with CRISPR/Cas9 via the GFP-Nanobody system successfully edited reporter cells, inducing dsRed expression. Scale bars: 10 μm. The data are presented as mean ± SD. The $p$ values (K) were calculated using a two-tailed $t$-test. The $p$ values (**b, c, e, f, g, i, j**) were analyzed by one-way ANOVA followed by Dunnett's multiple comparisons test. *$P < 0.05$, **$P < 0.01$. These figures (**a, d, h, m**) were created using MedPeer (medpeer.cn). Source data and exact $p$ value are provided as a Source data file.

EVs, we first demonstrated the physical binding of sgRNA to Cas9 in EVs through RIP assays, indicating the existence of RNP complexes (Fig. 3k). Subsequent co-IP analysis confirmed the loading of RNP complexes into EVs via the GFP-Nanobody system, as indicated by a Cas9 pull-down assay using an antibody against FLAG-tagged EN144 (Fig. 3l).

Having established the successful loading of Cas9 into EVs via the GFP-Nanobody system, we evaluated its functional integrity. We generated a stop-dsRed reporter cell line, where a stop sequence halts dsRed gene expression. Stop-dsRed is a genetic modification that involves adding a stop sequence to the dsRed gene, effectively halting its expression. However, introducing sgRNA1 and sgRNA2 precisely cleaves this stop sequence, thereby allowing the dsRed gene expression as intended (Fig. 3m). Confocal imaging confirmed the successful setup of reporter cells and plasmids, as evidenced by the expression of GFP and Red in stop-dsRed cells. Importantly, EVs containing CRISPR-Cas9 components induced detectable red fluorescent signals in stop-dsRed cells, demonstrating the function of RNP in EVs (Fig. 3n). Sanger sequencing followed by TIDE (Tracking of Indels by Decomposition) analysis further revealed that the editing efficacy in the stop-dsRed gene upon EV treatment ranged from 3.1 to 13.6% (Supplementary Table 5). These data indicate that RNP in EVs is functional in recipient cells.

## Decoy EVs alleviate systemic inflammation in LPS-induced septic mouse

Cytokines IL-6 and TNF-$\alpha$ are key mediators of inflammatory regulation. Next, we design EVs that display cytokine receptors. Glycoprotein 130 (also known as gp130, IL6ST, IL6R-beta, or CD130) is a type I cytokine receptor that binds to IL-6. Based on previous studies[25,26], the C-terminus of EN144 was fused to mouse gp130 (mgp130), comprising the extracellular, transmembrane, and partial cytoplasmic domains, and transiently expressed in Expi293F cells. Differential ultracentrifugation yielded engineered EVs displaying EN144-mgp130 (EN144-EV$^{mgp130}$, Fig. 4a). RPS analysis revealed EN144-EV$^{mgp130}$ diameters of 73 nm, comparable to typical EV sizes (Fig. 4b). To validate the efficacy of EN144-EV$^{mgp130}$, an in vitro inflammation model was established using LPS-stimulated RAW 264.7 cells. Co-culturing with 1000 ng/mL LPS for 3 h significantly upregulated TNF-$\alpha$ and IL-6 at both RNA ($P < 0.01$) and protein levels, imitating the inflammation conditions (Supplementary Fig. 15a–c). Quantitative analysis revealed that EN144-EV$^{mgp130}$ achieved significantly greater suppression of TNF-$\alpha$ and IL-6 in inflammatory RAW 264.7 cells (Fig. 4c, d). Encouraged by these data, we evaluated the therapeutic potential of EN144-EV$^{mgp130}$ in an LPS-induced sepsis mouse model. Septic mice exhibited heightened IL-6 and TNF-$\alpha$ expression in the liver, spleen, and lungs at 3 h post-LPS injection (Supplementary Fig. 15d–f). EN144-EV$^{mgp130}$ was found to be enriched in the liver, lungs, and spleen of septic mice 6 h after EVs were injected intravenously (Supplementary Fig. 15g). Consistently, EN144-EV$^{mgp130}$ treatment significantly reduced the IL-6 expression levels in the liver, lungs, and spleen (Fig. 4e). In vitro and in vivo investigations demonstrated that EN144-EV$^{mgp130}$ significantly attenuated IL-6 and TNF-$\alpha$ expression levels in a LPS-induced inflammation model.

Conservativeness analysis of gp130 across multiple species revealed high conservation levels in humans, rhesus monkeys (97.6%), rats (78.63%), and mice (77.18%), all exceeding 75%. This evolutionary conservation prompted further investigation into the therapeutic efficacy of human gp130 (hgp130). EN144-hgp130 was transfected into Expi293F cells, and hgp130 expression was detected in cells and EVs (EN144-EV$^{hgp130}$, Fig. 4f). Notably, despite comparable particle sizes, EN144-EV$^{hgp130}$ exhibited significantly higher loading efficiency than EN144-EV$^{mgp130}$ (Fig. 4f, g). Similar to the therapeutic effects of EN144-EV$^{mgp130}$, EN144-EV$^{hgp130}$ was observed to significantly reduce both IL-6 protein and RNA levels in inflammatory cells (Fig. 4h, i). Furthermore, we observed that EN144-EV$^{hgp130}$ exhibited a dose-dependent inhibitory effect on IL-6. Specifically, EN144-EV$^{hgp130}$ decreased IL-6 levels by 30% at a concentration of $5.0 \times 10^8$ particles/mL and by 55% at $1.0 \times 10^{10}$ particles/mL (Supplementary Fig. 16a). Building on these encouraging in vitro findings, the therapeutic potential of decoy EVs was subsequently evaluated in vivo using a sepsis model. Administration of varying EV doses demonstrated that intravenous injection of EN144-EV$^{mgp130}$ or EN144-EV$^{hgp130}$ ($1.0 \times 10^{10}$ particles/mouse) significantly improved survival outcomes in septic mice, achieving survival rates exceeding 80% within 120 h post-administration (Fig. 4j, k). Notably, when administered at the highest dose of $5.0 \times 10^{10}$ particles/mouse, both preparations achieved complete (100% up to 120 h after injection) survival protection (Fig. 4j, k). Comparative analysis revealed that EN144-EV$^{hgp130}$ (80%) and EN144-EV$^{mgp130}$ (80%) demonstrated an enhanced therapeutic efficacy in septic mice, with survival rates higher than those achieved by sgp130 treatment at 10 μg (40%) and 1 μg (20%) doses (Fig. 4l). Moreover, mice receiving decoy EVs therapy exhibited accelerated weight recovery (72 h) compared to the sgp130-treated group (96 h post-treatment, Fig. 4m). The decoy EVs exhibited superior protection against LPS-induced inflammation, as evidenced by reduced IL-6 levels in the liver, lung, and spleen (Supplementary Fig. 16b, c). We then assessed the impact of various treatments on the histopathology of sepsis mice. Mice in the PBS group exhibited massive inflammatory cell infiltration in the liver, lungs, and spleen, and elevated histologic scores. Conversely, decoy EVs treatment mitigated inflammation in the liver, lungs, and spleen, effectively alleviating LPS-induced inflammation (Supplementary Fig. 16d, e). Taken together, we demonstrate the anti-inflammatory efficacy of two protein-carrying EVs using EN144 as the scaffold protein, EN144-EV$^{mgp130}$ and EN144-EV$^{hgp130}$, in a cell-based model and an animal model of sepsis.

To investigate the regulatory role of EN144-EV$^{mgp130}$ in the IL-6 trans-signaling pathway, we first characterized its binding properties in vitro. Consistent with the canonical theory that gp130 activation requires the preformed IL-6/IL-6R complex[22], EN144-EV$^{mgp130}$ did not bind to IL-6 or IL-6R individually, and binding was observed exclusively with the IL-6/IL-6R complex (Supplementary Fig. 17a). In control experiments, EN144-EVs showed no binding to IL-6, IL-6R, or the IL-6/IL-6R complex (Supplementary Fig. 17a). These results demonstrate that EN144-EV$^{mgp130}$ specifically interacts with the IL-6/IL-6R complex. Building on these findings, we evaluated the biological effects of EN144-EV$^{mgp130}$ in a murine model. Treatment with EN144-EV$^{mgp130}$ significantly reduced serum levels of IL-6, sIL-6R, and pro-inflammatory cytokines (IL-1$\beta$ and TNF-$\alpha$; Fig. 4n–q), while decreasing the pSTAT3/STAT3 ratio in hepatic, splenic, and pulmonary tissues, collectively indicating effective suppression of IL-6 trans-signaling activation (Supplementary Fig. 17b–g). Notably, expression levels of classic signaling markers—acute-phase proteins serum amyloid A (SAA) and C-reactive protein (CRP)—remained unaltered, excluding nonspecific interference of EN144-EV$^{mgp130}$ with the canonical pathway (Fig. 4r, s). These results demonstrate that EN144-EV$^{mgp130}$ selectively inhibits the IL-6 trans-signaling pathway through targeted engagement of the IL-6/IL-6R complex, without appreciably affecting classical IL-6 signaling.

## Chondrocyte-targeting decoy EVs to repair cartilage damage

The sepsis model evaluated the systemic administration of EN144-EV$^{hgp130}$ in a non-targeted way. Next, we seek to assess the anti-inflammatory efficacy of EN144-EV$^{hgp130}$ for targeted delivery of protein to chondrocytes in the cartilage as a potential treatment for OA. Cartilage is a dense tissue that is difficult to penetrate. A CAP sequence was attached to the C-terminus of EN144-hgp130, resulting in EN144-hgp130-CAP. Subsequently, the plasmid was transiently transfected into Expi293F cells, and EN144-EV$^{hgp130-CAP}$ were isolated by ultracentrifugation. Based on the FLAG tag fused to the C terminus of the EN144-hgp130 sequence, we detected the fusion protein in all the transfected cells and EVs, and no notable change in EV sizes was observed (Fig. 5a, Supplementary Fig. 18a, b). The chondrocyte-

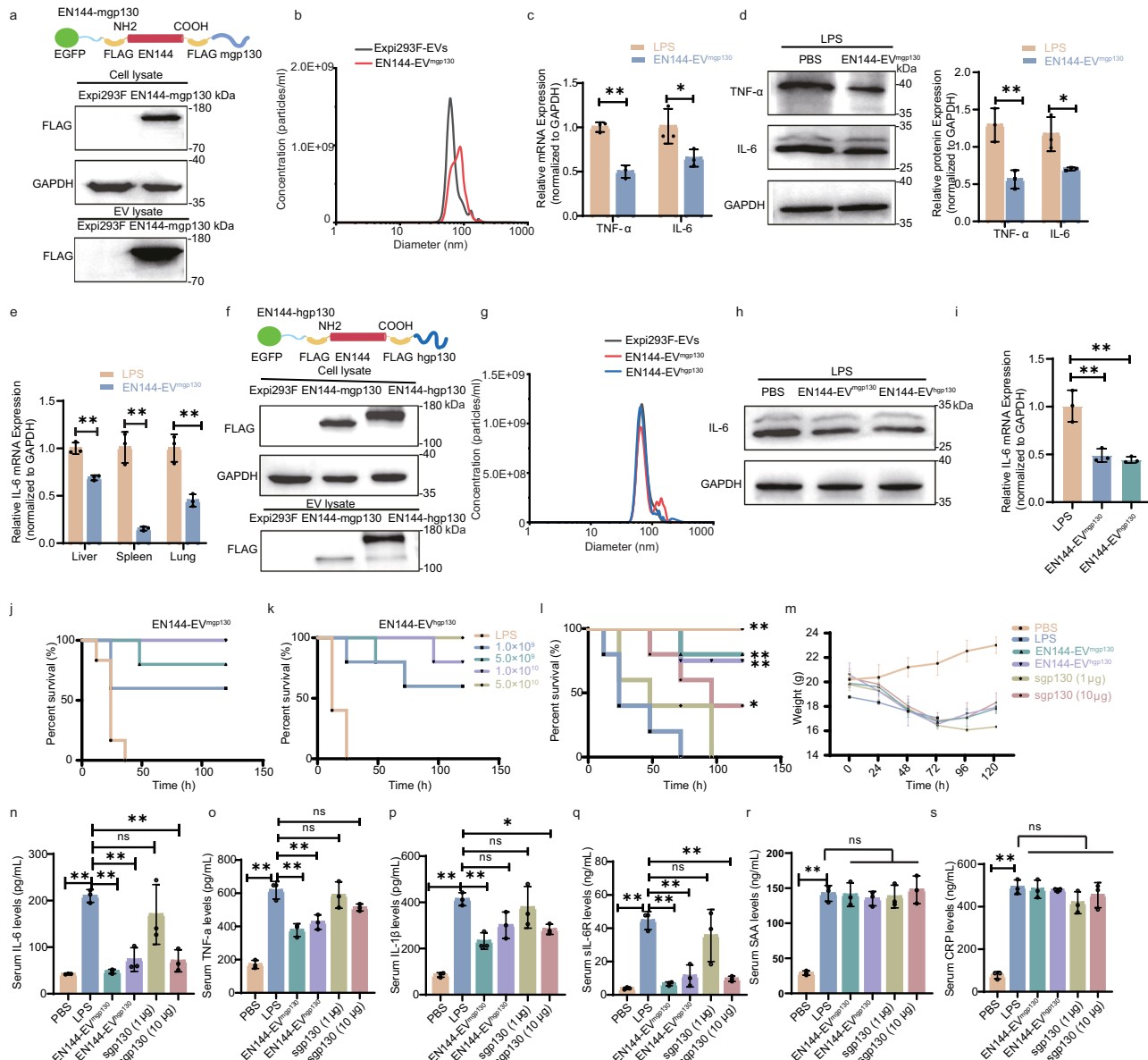

**Fig. 4 | Anti-inflammatory effects of EN144-EV^hgp130 in a mouse sepsis model.** Engineered EVs displaying species-specific gp130 (mouse mgp130 or human hgp130) were constructed and characterized, confirming protein expression (**a**) and typical EV size distribution (**b**). This figure was created using MedPeer (medpeer.cn). mRNA (**c**) and protein (**d**) levels of TNF-α and IL-6 in RAW 264.7 cells treated with EN144-EV^mgp130 post-LPS induction. Error bars were derived from 3 independent experiments. **e** mRNA expression of IL-6 in liver, spleen, and lung tissues of septic mice 6 h after intravenous injection of 1.0 × 10^10 particles EN144-EV^mgp130. **f** Expression of FLAG-tagged proteins in cell lysates (top) and EVs (bottom). This figure was created using MedPeer (medpeer.cn). **g** Size distribution of Expi293F-EVs, EN144-EV^mgp130, and EN144-EV^hgp130. Protein (**h**) and mRNA (**i**) levels of IL-6 in LPS-induced RAW 264.7 cells following co-culture with EN144-EV^hgp130 or EN144-EV^mgp130. $n = 3$ independent experiments. Survival rates of septic mice treated

with varying doses of EN144-EV^mgp130 (**j**) and EN144-EV^hgp130 (**k**). $n = 5$ mice/group. Survival rates (**l**) and body weight (**m**) changes of septic mice after tail vein injection of EN144-EV^mgp130 (1.0 × 10^10 particles/mouse), EN144-EV^hgp130 (1.0 × 10^10 particles/mouse), sgp130 (1 μg), or sgp130 (10 μg). $n = 5$ mice/group. Statistical analysis was performed using the Log-rank (Mantel-Cox) test. Serum levels of inflammatory cytokines and acute-phase proteins in septic mice treated with EN144-EV^mgp130, EN144-EV^hgp130, low-dose sgp130, or high-dose sgp130, including (**n**) IL-6, (**o**) TNF-α, (**p**) IL-1β, (**q**) soluble IL-6 receptor (sIL-6R), (**r**) serum amyloid A (SAA), and (**s**) C-reactive protein (CRP). $n = 3$ mice/group. The data are presented as mean ± SD. The $p$ values (**c**, **d**, **e**) were calculated using a two-tailed $t$-test. The $p$ values (**i**, **n**, **o**, **p**, **q**, **r**, **s**) were analyzed by one-way ANOVA followed by Dunnett's multiple comparisons test. *$P < 0.05$, **$P < 0.01$. Source data and exact $p$ value are provided as a Source data file.

targeting capacity of engineered EVs was evaluated using DiI-labeled EVs co-cultured with primary chondrocytes. Quantitative analysis by confocal microscopy demonstrated significantly enhanced binding of EN144-EV^hgp130-CAP to chondrocytes (115.4 ± 10.77) compared to EN144-EV^hgp130 (86.79 ± 0.12) at 2-h post-incubation (Fig. 5b). Prolonged 24-h co-culture revealed superior internalization efficiency of EN144-EV^hgp130-CAP, with 95.33% ± 0.31% DiI-positive cells versus 62.90% ± 0.66% in the EN144-EV^hgp130 control group (Fig. 5c). Moreover,

intra-articular injections of EN144-EV^hgp130-CAP demonstrated higher fluorescence intensity in the joints at 24 and 168 h than the group treated with EN144-EV^hgp130, and minimal systemic distribution to other organs (Supplementary Fig. 18c). These data show that chondrocyte-targeting EN144-EV^hgp130-CAP had significantly higher cellular uptake in vitro and cartilage-specific retention in vivo.

After validating the chondrocyte-targeting capability, we next investigated the anti-inflammatory efficacy of EN144-EV^hgp130-CAP in

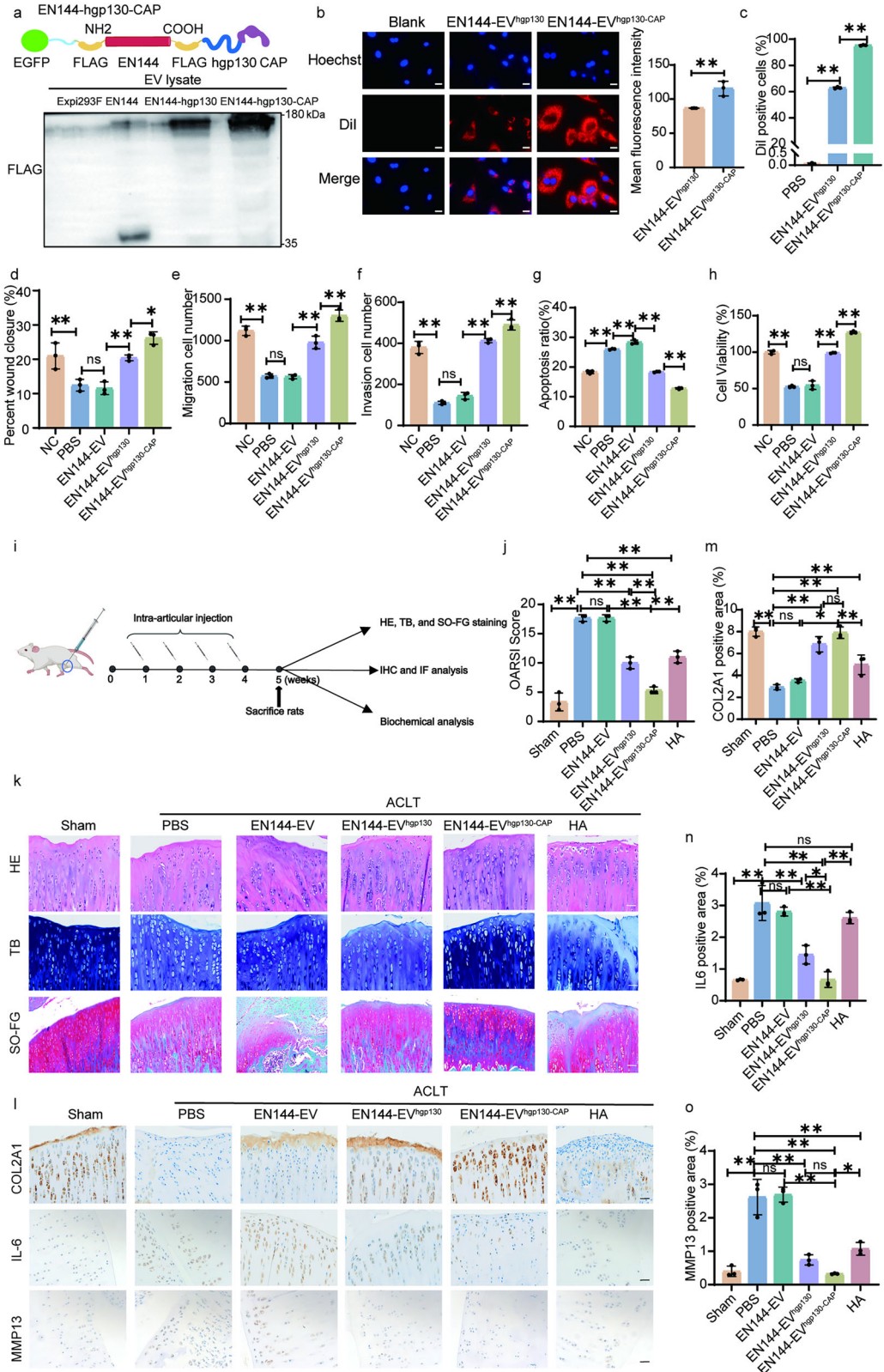

chondrocytes. EN144-EV$^{hgp130}$ or EN144-EV$^{hgp130-CAP}$ significantly promoted the proliferation and migration of inflammatory chondrocytes and reduced apoptosis compared with other treatment groups (Fig. 5d–h, Supplementary Fig. 19a–c). Notably, the CAP peptide on the surface of EN144-EV$^{hgp130-CAP}$ delivered hgp130 more efficiently into inflammatory chondrocytes, promoting chondrocyte repair more effectively than EN144-EV$^{hgp130}$ (Fig. 5d–h, Supplementary Fig. 19a–c),

highlighting the potential advantage of the targeting effect. The above results indicated that EN144-EV$^{hgp130-CAP}$ exerts enhanced chondroprotective effects compared to other treatments on inflammatory chondrocytes, thereby confirming the high anti-inflammatory efficacy of EN144-EV$^{hgp130-CAP}$ in vitro.

The efficacy of EN144-EV$^{hgp130-CAP}$ in vivo for cartilage repair and potential OA treatment was evaluated. OA conditions were induced in

**Fig. 5 | Construction of EN144-EV^hgp130-CAP for targeted repair of cartilage damage in OA rats. a** Construction of EN144-hgp130-CAP plasmid and assessment of FLAG protein levels in cell lysates or engineered EVs. This figure was created using MedPeer (medpeer.cn). Fluorescence imaging (**b**) and flow cytometry (**c**) show that CAP modification significantly enhances EV uptake by chondrocytes. Scale bars: 100 μm; 50 μm (Zoomed images). **d–h** Effects of various EV preparations (EN144-EVs, EN144-EV^hgp130, or EN144-EV^hgp130-CAP) on IL-1β-induced chondrocytes. **d** Cell migration based on a scratch assay, **e** cell migration based on Transwell assay, **f** cell invasion based on Transwell assay, **g** cell apoptosis, and **h** cell viability based on CCK-8 assay. NC non-inflamed chondrocytes (no IL-1β treatment). **i** Schematic illustration showing cartilage repair by EV treatment in vivo. Briefly, a rat model of OA was established based on the ACLT method. EV preparations (EN144-EVs,

EN144-EV^hgp130, and EN144-EV^hgp130-CAP) of $1.0 \times 10^{10}$ EV particles or HA (10 μg) were injected intra-articularly into OA rats weekly for 4 weeks. The rats were sacrificed in the fifth week, and the cartilages were dissected for analysis. This figure was created using MedPeer (medpeer.cn). **j** OARSI scores indicating cartilage damage severity in different treatment groups ($n = 3$). **k** Representative images of H&E, TB, and SO-FG staining for histological analysis. The sham group serves as a control. Scale bars: 200 μm (top panel), 50 μm (bottom panel). **l–o** Immunohistochemical expression levels of COL2A1, IL-6, and MMP13 in cartilage tissues from various treatment groups ($n = 3$). Scale bar: 50 μm. The data are presented as mean ± SD. The $p$ values (**b, c, d, e, f, g, h, j, m, n, o**) were calculated using a two-tailed $t$-test. *$P < 0.05$, **$P < 0.01$. Source data and exact $p$ value are provided as a Source data file.

rats using the anterior cruciate ligament transection (ACLT) surgical technique. The rats were divided into groups receiving intra-articular injections of various substances, including PBS, hyaluronic acid (HA), EN144-EVs, EN144-EV^hgp130, or EN144-EV^hgp130-CAP. A control group undergoing a sham surgery was also included in the study. The injections were administered once a week for four consecutive weeks, and rats were euthanized in the fifth week for further analysis (Fig. 5i). Based on the OA Research Society International (OARSI) score, the EN144-EV^hgp130-CAP treatment group exhibited the best recovery, as indicated by the lowest score. Comparing the EN144-EV^hgp130-CAP treatment group and the EN144-EV^hgp130 group, the targeting property of EVs provided a more pronounced chondroprotective effect (Fig. 5j).

Next, we conducted immunohistological staining to verify the therapeutic efficacy. Hematoxylin and eosin (H&E), Safranin O fast green (SO-FG), and toluidine blue (TB) staining were performed to assess cartilage degeneration in each group. The ACLT group exhibited cartilage thinning and decreased proteoglycan levels, whereas HA and EN144-EV^hgp130 treatment partially ameliorated these features (Fig. 5k). The EN144-EV^hgp130-CAP treatment group showed the most pronounced alleviation of cartilage degeneration and abrasion in OA rats (Fig. 5k). Immunohistochemistry (IHC) analysis revealed elevated COL2A1 and reduced IL-6 and MMP13 expression in the treatment groups compared to the ACLT group. Notably, the EN144-EV^hgp130-CAP group showed the highest COL2A1 level and the lowest levels of IL-6 and MMP13, which were superior to the levels in the EN144-EV^hgp130 and HA groups (Fig. 5l–o). These results indicate that EN144-EV^hgp130 and EN144-EV^hgp130-CAP groups promoted cartilage repair, with the EN144-EV^hgp130-CAP treatment being the most effective.

OA is characterized as a chronic inflammatory condition driven primarily by the actions of innate immune cells, particularly macrophages located in the synovial tissue. Immunofluorescence (IF) imaging of the synovium revealed an elevated presence of M1-like macrophages (iNOS⁺) and a reduced proportion of M2-like macrophages (CD206⁺) in OA rats. Quantitative analysis demonstrated a higher percentage of M2-like macrophages in all treatment groups than in the PBS group (Supplementary Fig. 20a, c, d). Notably, treatment with EN144-EV^hgp130-CAP resulted in decreased IL-6 expression in synovial tissue, comparable to the Sham group (Supplementary Fig. 20b, e). To evaluate the safety of intra-articular injection of EN144-EV^hgp130-CAP in vivo, rat organs (heart, liver, spleen, lungs, kidneys, and blood) were collected post-treatment. No significant pathological changes were observed in the EN144-EV^hgp130-CAP treatment group compared to the Sham group (Supplementary Fig. 21a–g). These results suggest that the EN144-EV^hgp130-CAP treatment effectively alleviates the OA inflammation by promoting the polarization of M2-like macrophages. Therefore, EN144-EV^hgp130-CAP improves the microenvironment of synovial tissue, reduces cartilage tissue inflammation, and thereby promotes cartilage tissue regeneration.

## Discussion

Efficient loading and targeted delivery of therapeutic molecules into EVs are pivotal for advancing EV-based drug delivery systems[9].

Researchers predominantly leverage cell engineering to fuse therapeutic proteins with vesicle-enriched proteins, thereby integrating cargos into EVs[37]. Compared to EVs derived from other cell types, Expi293F-EVs utilized in this study have garnered significant attention in preclinical research due to their high yield, absence of serum contamination, and minimal inter-batch variability[38]. To mitigate the potential risk of protein contamination in any given EV enrichment method, we employed three distinct and independent enrichment methods: ultracentrifugation, a magnetic bead-based EVtrap technology, and an industrial-scale approach combining tangential flow filtration with size exclusion chromatography for EV isolation. Rigorous screening and evaluation of various EV-sorting proteins were conducted to assess their efficiency in loading biomolecular cargo into EVs, thereby broadening the repertoire of viable sorting proteins. In our screening, ENPP1 emerged as an effective sorting protein, which showed higher efficiency than PTGFRN, already in clinical trials[39], and Lamp2b, widely used in preclinical studies[40]. Furthermore, EN144, a truncated variant of ENPP1, features a simple structure consisting of merely 144 aa and a substantially lower molecular weight than PTGFRN and Lamp2b. The loading of diverse biomolecules, including Cas9 proteins, mRNA, and targeted peptides, both on the surface and within EVs, was successfully achieved. Notably, EN144-modified EVs hosting hgp130 (EN144-EV^hgp130) mitigated inflammation and improved survival in a sepsis mouse model compared to the sgp130 treatment. Additionally, EN144-EV^hgp130-CAP reduced chondrocyte inflammation and improved the synovial tissue microenvironment (Fig. 6). Consistent with Expi293F-EVs[27], EN144-EVs exhibited low in vivo toxicity, underscoring their safety as drug delivery vectors.

ENPP1, a member of the ectonucleotide-pyrophosphatase/phosphodiesterase family, was initially detected in ciliated EVs of the nematode Cryptobacterium hidradiensis by Barr's group through proteomic profiling[41], subsequently extending its presence to diverse cell-derived EVs[42,43]. Our research pioneers the utilization of ENPP1 for endogenous cargo loading in EVs, effectively enriching EGFP within EVs. As a type II transmembrane glycoprotein, ENPP1 encompasses diverse structural domains, including cytoplasmic, transmembrane, catalytic, nuclease-like, SMB1, SMB2, and linker regions (L1, L2)[44]. These domains govern distinct biological functions, with the catalytic and nuclease-like domains crucial for bone mineralization regulation, while SMB domains facilitate homodimerization and plasma membrane translocation[45]. Notably, almost all biological actions of ENPP1 are associated with the catalytic and nuclease-like structural domains[46]. Given the ubiquitous involvement of these domains in ENPP1's activities, we devised EN144, a truncated variant devoid of these functional domains, to ensure maximal safety during drug delivery. Remarkably, EN144's compact size translates into a three-fold enhanced sorting capacity for biomolecules into EVs compared to ENPP1. Consequently, EN144 is designed to enhance EV loading efficiency and to minimize the potential biological side effects of ENPP1.

Despite their enhanced biomolecule loading, Rab7a-EVs exhibit reduced cellular uptake efficiency. Prior research implicates EV composition, membrane fluidity alterations, fusion protein exposure, and

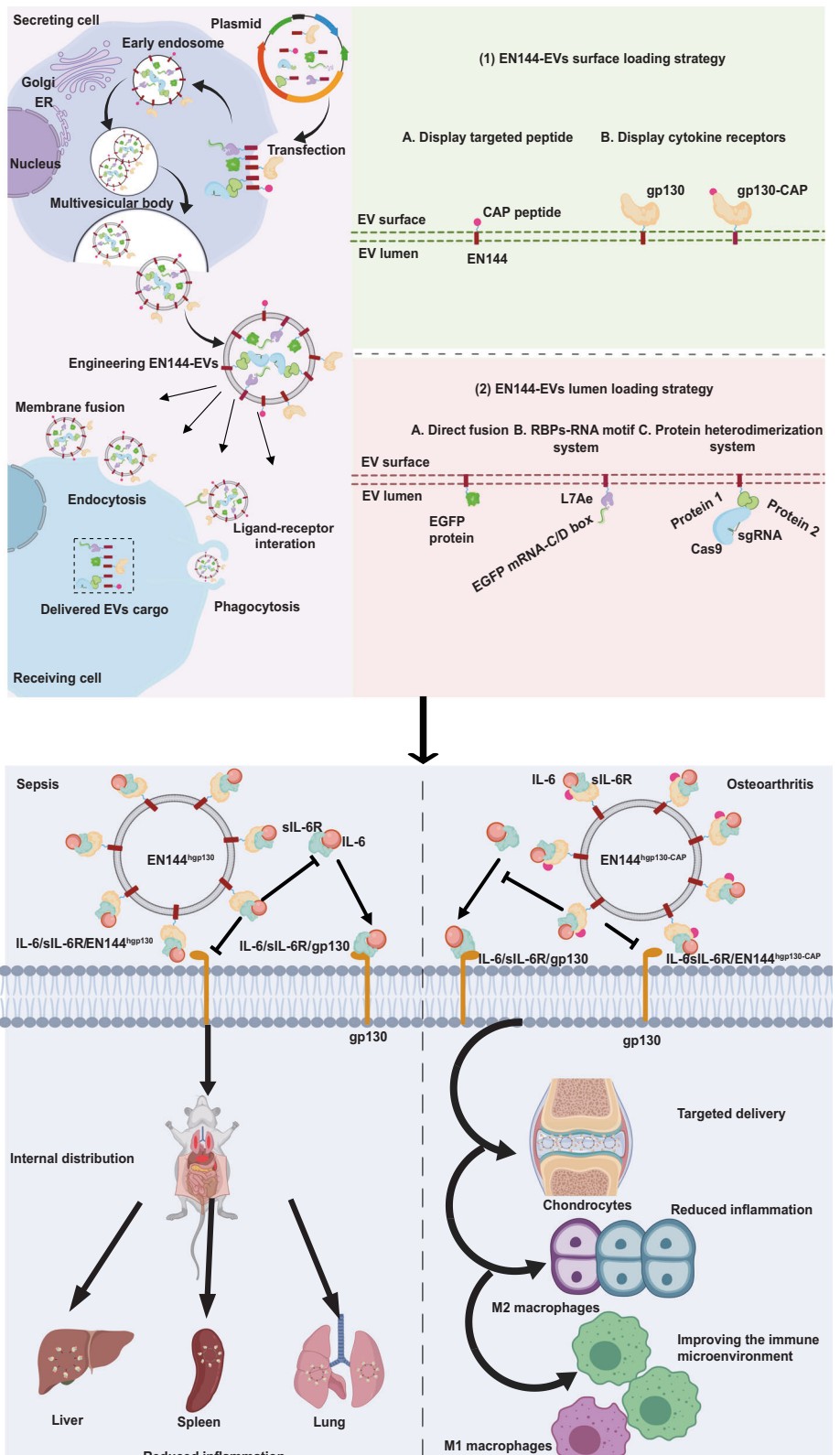

**Fig. 6 | A versatile platform for generating engineered EVs with defined therapeutic properties based on EN144.** This figure was created using MedPeer (medpeer.cn).

EV size or aggregation as factors influencing EV signaling and cargo transport[47]. Rab7a overexpression may modify the native structure and membrane protein profile, thereby impeding cellular uptake. Furthermore, Rab7a's pivotal role in endosomal-lysosomal regulation, its binding to autophagic vesicles, and its promotion of autophagosome maturation may expedite phagocytosis and degradation of Rab7a-EV-encapsulated biomolecules[48].

The biodistribution of EVs, as a pivotal factor in evaluating their therapeutic potential and safety, is paramount for next-generation drug delivery vehicles. Intravenous injection, the prevalent route for

systemic EV administration, has been extensively studied in preclinical disease models[49]. After i.v. injection in mice, EVs derived from various cell types mainly accumulate in the liver, spleen, kidney, and lung[50]. Concordantly, our study revealed that Expi293F-EVs and EN144-EVs were similarly amassed in the liver and spleen 24 h post-injection. Rapid clearance by the mononuclear phagocyte system, primarily by liver and splenic macrophages, accounts for this preferential accumulation[49,51]. Previous studies suggest that surface proteins of EVs affect uptake by different cells, thereby altering the distribution of EVs. For instance, CD47 inhibits macrophage-mediated EV uptake[52], while specific integrins enrich EVs in the liver[53]. Moreover, variations in glycosylation, proteins, lipids, and nucleic acids also contribute to distinct biodistribution patterns[54–56]. These factors likely underlie the enhanced liver and spleen accumulation, along with prolonged retention, of EN144-EVs compared to Expi293F-EVs. Surprisingly, despite these differences, EN144-EVs maintained the safety profile of Expi293F-EVs, exhibiting no significant toxicity upon multiple administrations[57].

Engineering of EN144-EVs facilitates the display of diverse proteins (e.g., CAP peptide, gp130) and the luminal loading of Cas9 protein and EGFP mRNA. Studies underscore the capability of RNA-binding proteins to recognize specific RNA sequences, enhancing targeted RNA encapsulation in EVs. The MS2 stem−loop and L7Ae−C/D box systems have been explored for RNA enrichment[58,59], with our findings indicating the L7Ae−C/D box as optimal for EGFP mRNA loading into EN144-EVs.

The efficacy of CRISPR/Cas9 therapeutics is hindered by ineffective and unsafe delivery methods[34]. EVs emerge as promising carriers for safe and efficient CRISPR/Cas9 delivery[34]. Efforts to improve Cas9 loading in EVs include the fusion of GFP-GFP Nanobody with CD63 and Cas9, respectively, and light-induced dimerization strategies, achieving ~25 Cas9 molecules per EV[33,34]. Our study demonstrates that for the EN144 scaffold, the GFP-Nanobody system yields a higher capacity and delivery efficiency than other tested dimerization systems. The findings demonstrate that the protein heterodimerization on the EN144 scaffold enabled the efficient loading of Cas9 in EVs, underscoring the versatility and potential of the EN144-EVs engineering platform.

As EVs' pivotal roles in diverse physiological processes emerge, their therapeutic potential is vigorously investigated. To evaluate the therapeutic applications of EN144-EVs, we developed EN144-EV^mgp130 to act as a decoy for TNF-α and IL-6, respectively. Moreover, in sepsis patients, while TNF-α levels significantly decline, IL-6 concentrations persist, indicating IL-6 as a more viable anti-inflammatory target[60]. Additionally, gp130 is a receptor for the IL-6 family of cytokines (CNTF, CT-1, LIF, OSM, and IL-27)[61,62]. Given the broad ligand specificity of gp130, this may contribute to the superior anti-inflammatory efficacy of EN144-EV^mgp130. In addition, the soluble gp130 protein lacks species specificity[22,63], elucidating EN144-EV^hgp130's effectiveness in reducing inflammation across mouse sepsis and rat OA models.

In summary, EN144-EVs emerge as a safe and versatile drug delivery platform that is amenable to diverse engineering strategies, including the surface presentation of targeting ligands and decoy receptors, as well as luminal encapsulation of biomolecules. Our findings underscore the therapeutic potential of decoy EVs in sequestering cytokines, presenting a promising approach for treating sepsis and OA. By integrating protein therapeutics with natural EVs that traverse tissue barriers, engineered EVs are poised to revolutionize the field of biotherapeutics.

## Methods

### Materials and reagents

Expi293F cells (#A14528), Penicillin-Streptomycin (#15070063), Lipofectamine 2000 (#1668019), and Protein A/G Magnetic Beads (#88802) were acquired from Thermo Fisher Scientific, MA, USA. RAW 264.7 cells (#KGG2201-1), 293T cells (#KGG3101-1), and dihydroethidium (DHE) probe (#KGA7502-5) were provided by KeyGEN Biotech Inc., China. Fetal Bovine Serum (FBS, #SA112.02) was sourced from CellMax, China. Additionally, Dulbecco's Modified Eagle's Medium (DMEM, #BL304A) and Phosphate Buffered Saline (PBS, #BL302A) were acquired from Biosharp, China. The Rat IL-1β protein (#80023-RNAE) and Mouse gp130/IL6ST protein (50135-M02H) were purchased from SinoBiological, China. The Mouse IL-6 protein (#CG39) and Mouse IL-6RA protein (#C06P) were acquired from Nonoprotein, China. The enzyme-linked immunosorbent assay (ELISA) kit used to detect CRP (#YM-2810A1), sIL-6R (#YM-6368A1), SAA (#YM-3116A1), IL-6 (#YM-2899A1), IL-1β (#YM-2776A1), and TNF-α (#YM-2868A1) were provided by YOUMENG BIOLOGY, China. Albumin Bovine (BSA, #9048-46-8) and collagenase II (#9001-12-1) were acquired from Bio-Froxx, Germany. The PVDF membranes (#ISEQ00010), 0.22 μm filter membranes (#SLGPR33RB), and Lipopolysaccharide (LPS, #L2630) were purchased from Sigma-Aldrich, MO, USA. Chemiluminescence kit (#E422-02-AB), FreeZol Reagent Kit (#R711-02), Prestained Protein Marker (#MP102-02), and SYBR Green PCR Mix (#Q712−02) were obtained from Vazyme Biotech Co., Ltd., China. The DiR (#C019), DiI dyes (#C017), and Hochest dyes (#FP027) were purchased from ABP Biosciences, MD, USA. Matrigel™ Basement Membrane Matrix (#356234) was provided by BD Biosciences, CA, USA. HA (#9067- 32-7, Molecular Weight ≥1.8 MDa, intrinsic viscosity ≥27.2 DL/G) were obtained from Aladdin, China. The DAB chromogen Kit (#G1212-200T) was obtained from Servicebio, China. PEI MW40000 (#40816ES), 2 × Hieff® PCR Master Mix (#10102ES03), Annexin V-YSFluorTM (#40304ES60) were purchased from Yeasen, China. LDH Cytotoxicity Assay Kit (#C0017), Antifade Mounting Medium with DAPI (#P0131), PMSF (#ST2573), Protease inhibitor cocktail for general use (#P1005), RNase inhibitor (#R0102), and Proteinase K (#ST535) were provided by Beyotime, China. Evtrap (EV02-01-01) was obtained from EVLiXiR, China. A comprehensive list of antibodies used in this study is detailed in Supplementary Table 1.

### Cell culture

The cell lines RAW 264.7, 293T, HcerEpic, HeLa, Rat Primary Chondrocytes (Chondrocytes), and SF-MSCs were maintained in DMEM media supplemented with 10% FBS and 1% penicillin-streptomycin. SF-MSCs were a gift from Dr. Yujie Liang[64]. Chondrocytes were extracted according to a previous protocol[32]. Expi293F cells were cultured in HEK293 Media under continuous shaking at 120 rpm. All cell types were housed in humidified incubators at 37 °C and 5% $CO_2$.

### Liquid chromatography-tandem mass spectrometry (LC-MS/MS)

LC-MS/MS processing and analysis were performed as previously described[32]. The peptide samples were subjected to analysis using the QE HF-X instrument (Thermo Fisher Science, USA). For the proteomic analysis, EVs lysates ($3.0 \times 10^9$) underwent chromatographic separation on the nLC 1200 system (Thermo Scientific, USA), equipped with a C18 chromatographic column (2.2 μm, 100 Å; Michrom Bioresources). The chromatographic separation was performed at a flow rate of 300 nL/min, utilizing an aqueous solution of 0.1% formic acid (solvent A) and an 80% acetonitrile solution containing 0.1% formic acid (solvent B) as the mobile phases.

The chromatographic column was initially equilibrated with 100% of liquid A, followed by a gradient elution: 2–8% B for 0–3 min, 8–40% B for 3–81 min, 40–95% B for 81–83 min, and maintained at 95% B for 83–90 min. Data acquisition was performed in data-dependent acquisition mode, with MS1 settings of 60,000 resolution, AGC target of 3e6, maximum inject time of 30 ms, and a scan range of 400–1200 $m/z$. For MS2, the settings included 30,000 resolution, AGC target of 1e5, maximum inject time of 50 ms, loop count of 20, collision energy of 28, and dynamic exclusion of 40 s.

The raw data were searched using Proteome Discoverer 2.3 software, configured with trypsin/P as the enzyme, allowing up to 3 missed cleavages. Fixed modifications included carbamoyl methylation (+57.02 Da), while variable modifications encompassed methionine oxidation (+15.99 Da) and acetylation (+42.01 Da). Both protein and peptide false discovery rates (FDRs) were set to 1% to ensure high data quality.

To detect the expression of EGFP in engineered EVs, EV lysates derived from a standardized input of $3.0 \times 10^9$ particles (as determined by RPS) underwent proteomic analysis. LFQ of EGFP was performed using PEAKS Online with the following parameters: database, UniProt human proteome + EGFP sequence; enzymatic specificity, set to trypsin/P, and allowed up to 2 missed cleavages; fixed modification, carbamidomethylation (+57.02 Da); variable modification, methionine oxidation (+15.99 Da) and acetylation (+42.01 Da); FDR, protein and peptide false discovery rates set to 1%. EGFP abundance was normalized to total protein intensity per sample and reported as LFQ intensity. The raw LFQ intensity for EGFP was used for cross-sample comparison. To account for potential minor technical variations during mass spectrometry runs, the data were also processed with a total protein intensity normalization in the PEAKS software; however, all biological interpretations are made based on the comparison of signals originating from an equal number of EV particles.

## Transfection and stable cell line selection

To generate stable cell lines, the target plasmids were transfected into Expi293F cells using electroporation (Celetrix, USA) or PEI MW40000 transfection reagents. Specifically, Expi293F cells ($1.0 \times 10^7$) were mixed with the plasmid (20 μg) and transferred to a 200 μL electrotransfer tube. Cell Line mode was then selected, and a single electro transformation at 1125 V was performed. Unlike electroporation, PEI transfection was performed by mixing the plasmid (20 μg) and PEI MW40000 transfection reagent (1 mg/mL, 60 μL) for 15 min and then adding it uniformly dropwise to the medium containing Expi293F cells ($1.0 \times 10^7$). Stable cell lines were screened by G418 and regular passaging until the viability of the cell lines was restored to more than 90%, and then detected the expression of target genes[6].

293T, HcerEpic, HeLa, and SF-MSC cells were seeded in 100 mm culture dishes ($1-2 \times 10^6$ cells/dish) 24 h prior to transfection. Cells at 70–80% confluency were transfected with 10 μg of EN144 plasmid using 20 μL PEI MW40000 via incubation of DNA-PEI complexes (formed in serum-free DMEM, 15 min RT), followed by dropwise addition; complete medium replacement was performed at 6 h post-transfection. Conditioned media were collected at 48 h and 72 h for EVs isolation.

## EVs isolation and characterization

According to the previously established protocols, EVs were extracted and characterized, and the viability of cells before EVs isolation was maintained above 95%[28]. Morphological analysis was conducted using TEM (HITACHI, Japan), where EVs were deposited onto Formvar-coated nickel grids, negatively stained with 2% uranyl acetate, air-dried, and visualized. RPS was employed for the determination of particle size and concentration. Using a NanoCoulter counter (Resun Technology, China), a chip with a measurement range of 60–200 nm was installed, and 200 μL of PBS buffer was rapidly dispensed into each sample well on both sides of the detection card, ensuring the system was free of air bubbles or leaks. Before sample analysis, a background verification was performed, confirming that the particle count in PBS was fewer than 5 over 60 s. In each sample, 100 EV particles were selected for size analysis. To detect marker proteins in EVs, WB analysis was performed to detect the expression levels of TSG101, Hsp70, CD9, and Calnexin in cells or EVs. First, the lysed cells (10 μg) or EVs ($3.0 \times 10^9$) proteins were quantified by NanoDrop (Thermo Scientific, USA), and then the proteins were separated by 8–15% SDS-PAGE.

Subsequently, the proteins were transferred to 0.22 μm or 0.45 μm PVDF membranes and closed with 1% BSA. Next, incubation of primary and secondary antibodies was performed, and finally, an enhanced chemiluminescence solution was used to show protein bands.

## Plasmid design and construction

All protein sequences were obtained from UniProt to generate chimeric proteins (protein sequences available in Supplementary Table 2). During the plasmid construction process, the following elements were utilized: the FLAG tag (DYKDDDDK), a general signal peptide (MGWSCIILFLVATATGVHS), a protective peptide (GNSTM), a translation-promoting Kozak sequence (GCCGCCACC), and a flexible linker peptide (GSGS). The sorting protein sequence with N-terminal located in the EVs lumen was EGFP-flexible peptide-FLAG tag protein-sorting protein-FLAG tag protein, and this sequence was constructed in the pEGFP-C1 vector. The sorting proteins with a C-terminal located in the EVs lumen had the following sequence: promoting translation peptide-universal signal peptide-protective peptide-FLAG tag protein-Sorting protein-FLAG tag protein-flexible peptide-EGFP, and this sequence was constructed in the pCDNA3.1 vector. The PDGFR sequence was incorporated into the pDisplay plasmid, enabling EGFP fusion to PDGFR within the plasmid. Expression was initiated by the CMV promoter on the carrier, with EGFP labels on the carrier protein, and G418 resistance genes used for the detection and screening of positive cells.

## Cellular uptake of EVs

To evaluate the targeting efficiency of EVs, glass coverslips were placed at the bottom of 12-well plates, and each well was seeded with $8.0 \times 10^4$ cells and cultured overnight. $3.0 \times 10^9$ counts of DiI-labeled EVs were added to each well and co-cultured with cells for 15 min or 2 h. Subsequently, the medium was aspirated, and cells were washed twice with PBS. Cells were then fixed using 200 μL of 4% paraformaldehyde (PFA) for 20 min at room temperature. After washing, cells were further incubated with 500 μL serum-free DMEM medium containing Hoechst 33342 at 37 °C for 30 min to label nuclei. The slide was then sealed with an anti-quenching and visualized under a confocal laser microscope (Olympus, FV1000) with the following excitation/emission settings: DiI (549 nm/565 nm), Hoechst (350 nm/461 nm), and EGFP (488 nm/507 nm).

## LDH cytotoxicity assay, cell proliferation, apoptosis, migration assay, and reactive oxygen species (ROS)

To assess the cellular impacts of varying EN144-EVs concentrations, a series of cytotoxicity and phenotyping experiments was employed. Prior studies have established the non-cytotoxic nature of Expi293F-EVs across a concentration (range $5.0 \times 10^8$ to $5.0 \times 10^9$ particles/mL) in HepG2 cells, prompting our investigation into the cytotoxicity of EN144-EVs at concentrations (including $5.0 \times 10^8$, $1.0 \times 10^9$, $5.0 \times 10^9$, $1.0 \times 10^{10}$, and $5.0 \times 10^{10}$ particles/mL) on RAW 264.7, 293T, Chondrocytes, and SF-MSCs.

For the LDH assay, cells were seeded at 5000 cells/well in a 96-well plate and incubated overnight. Following 24 h exposure to various EN144-EVs concentrations, cells were centrifuged at $400 \times g$ for 5 min, and the supernatant, post-LDH release reagent treatment, was analyzed after further centrifugation. The resulting supernatant (120 μL) was transferred to a new plate and mixed with LDH assay solution (60 μL), incubated at room temperature for 30 min, and absorbance measured at 450 nm to quantify LDH release.

For the level of ROS in cells by EN144-EVs, the DHE probe was employed. DHE is a marker of intracellular peroxides, and once oxidized, it enters the DNA of the cell nucleus, allowing the nucleus to emit a bright red fluorescence. After the cells were exposed to different concentrations of EN144-EVs for 24 h, the medium was replaced with fresh medium (1 mL/well), and the DHE probe (1 μL/well) was

added. After incubation for 30 min in the dark, the proportion of fluorescent cells was quantified by flow cytometry.

The effects of EN144-EVs on cell proliferation, apoptosis, and migration were evaluated using standardized protocols for CCK-8, flow cytometry, scratch, and transwell assays, respectively, consistent with previous publications[28,65].

### Quantitative real-time PCR (qRT-PCR)

To elucidate the impact of various treatments on gene expression profiles, qRT-PCR was conducted. Total RNA was efficiently extracted from EVs, cells, and tissues using the FreeZol Reagent, adhering strictly to the manufacturer's procedure. This RNA was then converted to cDNA through reverse transcription utilizing the RT MasterMix Kit. Subsequently, qRT-PCR was performed with SYBR Green PCR Mix for relative quantification. The $2^{-\triangle\triangle CT}$ method facilitated the analysis of results. The primers relevant to the study were listed in Supplementary Table 3.

### RNA immunoprecipitation (RIP) assay

Cells and EVs were lysed in a buffer containing 10 mM Tris, 200 mM NaCl, 30 mM EDTA, 0.5% Triton-X 100, 0.5% NP40, 1 mM PMSF, 1% protease inhibitor cocktail, and 10 U/mL RNase inhibitor for 30 min on ice. Following centrifugation at $16,000 \times g$ for 10 min at 4 °C, anti-Cas9 antibody or IgG was incubated with 50 μL Protein A/G Magnetic Beads for 30 min at 4 °C. Overnight incubation at 4 °C ensued with the supernatants and beads. Subsequently, immunoprecipitated complexes were eluted from the beads using a lysis buffer. These complexes were then processed for either WB analysis using RIPA buffer or RNA analysis using FreeZol Reagent.

### Co-Immunoprecipitation (co-IP)

Lysed cells and EVs were supplemented with either 3 μg of anti-FLAG antibody or anti-IgG antibody and incubated overnight at 4 °C. Subsequently, 50 μL of Protein A/G Magnetic Beads were added to the mixture and incubated at room temperature for 2 h. Unbound complexes were thoroughly removed through two wash steps using a total of 500 μL of wash buffer. The antigen-antibody complexes were then eluted with lysis buffer and further processed with RIPA buffer for WB analysis.

### Sanger sequencing

Sanger sequencing was employed to assess the editing efficiency at the target genomic loci. PCR amplification was performed in a 20 μL reaction mixture containing 3 μL of template DNA, 1 μL each of forward and reverse primers (10 μM), 10 μL of 2 × Hieff® PCR Master Mix, and 5 μL of ddH$_2$O. The amplification protocol consisted of an initial denaturation at 95 °C for 5 min, followed by 40 cycles of denaturation at 95 °C for 25 s, annealing at 62 °C for 30 s, and extension at 72 °C for 20 s, with a final hold at 4 °C. The resulting DNA fragments were separated by electrophoresis on a 2% agarose gel. Subsequently, the purified PCR products were subjected to Sanger sequencing, and the editing efficacy was analyzed using the TIDE algorithm[66]. Editing efficiency was determined by sequencing analysis of the target locus in genomic DNA extracted from the FACS-sorted, red fluorescent-positive cell population. It represents the frequency of indel mutations among all stop-dsRed copies in the fluorescent 293T cells.

### Enzyme-linked immunosorbent assay

Serum levels of IL-6, sIL-6R, CRP, SAA, TNF-α, and IL-1β were quantified using a two-antibody sandwich ELISA. Briefly, 96-well microplates were pre-coated with capture antibodies specific to each target protein. Standards (recombinant proteins) and serum samples were loaded into designated wells, followed by incubation with horseradish peroxidase (HRP)-conjugated detection antibodies. After sequential incubation and thorough washing, tetramethylbenzidine (TMB) substrate was added for chromogenic development. The enzymatic reaction was terminated by adding a stop solution, converting the blue chromogen into a stable yellow product. Optical density was measured at 450 nm using a microplate reader, and analyte concentrations in mouse serum were calculated based on standard curves.

### In vivo distribution of EVs

EVs were labeled with the near-infrared fluorescent dye DiR (10 mM) by incubating $1.0 \times 10^{11}$ particles/mL EVs with 5 μL of DiR dye at 37 °C for 30 min. Unbound dye was removed using a 0.22 μm filter, and the DiR-labeled EVs were resuspended in sterile PBS. Mice or rats received either tail vein or intra-articular injections of DiR-EVs. At designated time points post-injection, animals were anesthetized and transferred to an IVIS Spectrum imaging system (PerkinElmer). Fluorescent signals were acquired at excitation/emission wavelengths of 754/778 nm. For ex vivo tissue analysis, euthanized animals were dissected, and major organs (e.g., heart, liver, spleen, lung, kidneys, legs) were placed on black cardboard for fluorescence imaging under identical optical settings.

### Histopathological examination and immunofluorescence (IF) staining

Histopathological examination and IF were performed to assess the therapeutic effect of EVs, as described previously[28]. Briefly, after the sample is embedded and trimmed, it is sliced. The sections are then sequentially immersed in xylene for 15 min, followed by another 15 min in xylene, and then in absolute ethanol I and II for 2 min each. Finally, they are immersed in 95% ethanol, 70% ethanol, and double-distilled water for 2 min each to complete the dewaxing process. Sections were stained with H&E, TB, and SO-FG. For IHC, paraffin-embedded sections were stained with anti-MMP13, anti-IL-6, or anti-COL2A1, followed by staining using the DAB chromogen kit. In the IF experiment, tissue sections were first subjected to antigen retrieval and blocking procedures. Subsequently, primary antibodies (iNOS and CD206) were added and incubated at 4 °C for 24 h. Following this, the sections were washed in PBS solution, and fluorescent secondary antibodies were applied and incubated in the dark at room temperature for 1 h. Finally, DAPI staining was performed, and the sections were blocked.

### In vivo toxicity assay

To detect EN144-EVs toxicity in vivo, male C57BL/6 mice (18–24 g) were randomly administered PBS, Expi293F-EVs, and EN144-EVs intravenously. Based on treatment protocols from clinical studies involving COVID-19 patients, each patient received an intravenous injection of $3.26 \times 10^{11}$ amniotic fluid-derived EVs[67]. Considering a human body weight of 70 kg and a mouse body weight ranging from 18–24 g, along with an extrapolation factor of 9.1 between mice and humans, the calculated dosage for each mouse is approximately $0.763-1.02 \times 10^9$ particles EVs. To assess the toxicity of EN144-EVs administered intravenously within a 100-fold dosage range, three groups were established: a high-dose group receiving $1.0 \times 10^{11}$ particles/mouse, a mid-dose group receiving $1.0 \times 10^{10}$ particles/mouse, and a low-dose group receiving $1.0 \times 10^9$ particles/mouse. Each group consisted of 6 mice, and 24 h post-injection, we collected tissues and blood samples from the mice for toxicity evaluation.

The high-dose EN144-EVs were injected three times weekly for 1 week to evaluate the toxicity of repeated administration of EN144-EVs. The body weight of mice was measured from day 1 to day 6 following the initial EN144-EVs injection. Whole blood samples were collected for routine and biochemical analyses. Subsequently, samples of the heart, liver, spleen, lungs, and kidneys were collected, fixed in 4% PFA, and embedded in paraffin for subsequent histological experiments.

## Animal housing conditions

C57BL/6 mice and SD rats were housed in the Specific Pathogen-Free barrier facility at the Animal Center of Southeast University. Under the supervision of the Animal Ethics Committee, all animals were co-housed under strictly controlled environmental conditions, including a 12-h light/dark cycle, a temperature of $22 \pm 2\,°C$, and a humidity of $50 \pm 10\%$, to minimize variation. The facility maintained sterility through the use of disinfected equipment, autoclaved feed and water, and strict air-shower entry protocols.

## Animal models

The LPS was used to induce a sepsis model, and male C57BL/6 mice (18–22 g) were selected for the experiment. Each mouse was given 10 mg/kg of LPS via intraperitoneal injection. After LPS induction, different drugs were administered through tail vein injection, and the time of death and changes in body weight of the mice were recorded.

To establish an animal model of OA, male SD rats weighing 190–210 g were subjected to ACLT as previously described. Briefly, the rats were anesthetized with 3% isoflurane. For the sham surgery (control group in this study), both knees were exposed after a medial capsule incision of the patellar tendon, and the skin incision was closed with sutures. For ACLT, the ACL of both knees was cut after opening the joint capsule. The skin incision was closed after tissue debris was removed by rinsing with saline. Four weeks after ACLT, the rats were randomly divided into five groups ($n = 5$ rats/group) receiving different treatments: PBS, HA (the widely clinically used injectable conservative drug for OA), EN144-EVs, EN144-EV$^{hgp130}$, and EN144-EV$^{hgp130-CAP}$. The rats received intra-articular injections through the joint cavity once a week for 4 weeks and were sacrificed in the fifth week to collect cartilage and internal organs for histological analysis.

## Study approval

All animal experiments were authorized by the Animal Ethics Committee of Southeast University under reference number 20220301031.

## Statistics and reproducibility

Unless otherwise specified, all experiments were performed in three independent replicates, and the displayed WBs, gels, and fluorescence images were selected as representative images from three independent replicate experiments. Statistical evaluations were conducted utilizing GraphPad Prism version 8.02 (GraphPad Software, USA). Two-tailed $t$-tests assessed statistical significance between two groups with parametric data. For comparisons among multiple groups, either one-way ANOVA or Brown-Forsythe ANOVA test was used, followed by Dunnett's or Dunnett's T3 multiple comparisons test, respectively. $P < 0.05$ was deemed statistically significant. Data are presented as mean ± standard deviation (SD).

## Reporting summary

Further information on research design is available in the Nature Portfolio Reporting Summary linked to this article.

## Data availability

The mass spectrometry proteomics data have been deposited to the ProteomeXchange Consortium via the PRIDE partner repository with the dataset identifier PXD057965. Source data is available for Figs. 1b–d, g, 2a, b, e–h, 3b, c, e–g, i–l, 4a, c–f, h–s and 5a–o, and Supplementary Figs. S1c, S3, S4, S5, S6, S7, S8, S9, S11, S12, S13, S14, S15, S16, S17, S18, S20 and S21 in the associated source data file. Source data are provided with this paper.

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

## Acknowledgements

This work was supported by the Fundamental Research Funds for the Central Universities 3225002202A1 (E.Y.), Major Project of Jiangsu Health Vocational College JKA202202 (S.W.) & JKD202405 (S.W.), Postgraduate Research & Practice Innovation Program of Jiangsu Province KYCX24_0489 (H. Liu), Medical Foundation of Southeast University 4060692202/021 (E.Y.), Zhishan Young Scholar Award at the Southeast University 2242023R40031 (E.Y.), Shandong Province Medical and Health Science and Technology Project 202525021171 (W.Y.). Post-doctoral Fellowship Projects from Jinling Hospital, Affiliated Hospital of Medical School, Nanjing University (97391 to H.Z. and 97157 to H.H.) are acknowledged. The authors also acknowledge Nanjing Jiangbei New Area Biopharmaceutical Public Service Platform Co., Ltd. for mass spectrometry analysis. This work was also partially funded by an ITF TCFS grant GHP/074/20SZ (J.X.) and internal grants from CUHK (a CRIMS grant, a Faculty Impact Case Fund, a direct grant, and an SIEF Fund to J.X.).

## Author contributions

H.Z. and J.X. conceived and designed the study. H.Z. and S.W. supervised and acquired funding for this project. W.Y., H.H. and S.W. wrote the first draft. H. Lin, C.W., S.X., X.Z., Y.L. and X.D. performed the experiments. X.C., H. Liu, D.W., C.M., C.T., X.L., J.H., Z.G. and B.Y. analyzed the data. G.Z. provided mass spectrometry expertise. W.J., S.W., E.Y., W.L., J.X. and H.Z. revised the manuscript.

## Competing interests

This research has been filed for a patent under application number CN116769718A with W.Y. and H.Z. named as inventors. J.X. and H.Z. co-founded EVLiXiR Biotech Inc., where H.Z. serves as CEO and J.X. as chief scientist. All other authors declare no conflicts of interest.
