## [Transparent Peer Review file · Nature Communications]

Extracellular vesicle engineering using a small scaffold protein

Corresponding Author: Professor Jiang Xia

Version 0:

Reviewer comments:

Reviewer #1

(Remarks to the Author)

This manuscript describes a discovery-based project and genetic engineering of cells to endogenously express an EV sorting protein/ scaffold. The authors claim that when EVs possess this novel sorting protein, the EVs have enhanced ability to target a cell (when co-expressed with a specific cell targeting protein) and load therapeutic cargo. Based on proteomic studies, the authors purport that a sorting protein, ENPP1, is superior to other sorting proteins previously identified, and that a truncated 144 aa ENPP1mversion (EN-144) is even more optimal. The authors then tested the EN-144 expressing EV in several paradigms. The author tests the EN-144 in collaboration with gp130, acts as a decoy in IL-6/IL-6 receptor system in models of sepsis and osteoarthritis, a degenerative disorder that is highly inflammatory. Cell targeting proteins (CAP) are employed in EVs that target chondrocytes in the osteoarthritis model. A weakness is that the studies to prove targeting are not rigorous. The authors conclude that then EN-144 engineered EVs can target chondrocytes and ultimately antagonize inflammation. While the concept of the project is exciting and thoughtful in regard to developing a new therapeutic, the manuscript is very dense and lacks sufficient details and controls in several experiments that dampens this reviewer's enthusiasm for the work.

Overall Strengths

- The authors used 3 distinct methods to isolate EVs which adds to rigor of this EV study and strengthens their mass spec results.
 - Discovery approach - performed comprehensive proteomic analyses for sorting proteins that might improve the current standard.
 - Identifying mechanisms that can assist with targeted drug delivery by EV have great therapeutic potential in many diseases, not just inflammation
 - In Vivo studies designed to test the safety of the EN-144 EVs
 - Used two distinct models for biological efficacy – sepsis and osteoarthritis
- Focus on GP130 that has a broader range than TNFR1

Weaknesses

- It is odd that whole cell lysates of EXpi293F do not express CD9 (Fig. 1S) and that CD9 is only observed in the EVs.
- In Figure 2, the GAPDH immunoblotting is not consistent and difficult to interpret.
- In Figure 4, the timing of the EV uptake is not clear; was live cell imaging employed? There are no clear methods included for the EV uptake studies.
- The Lamp2bEV high powered image lacks focus compared to the other images. Thus, it is very difficult to make quantitative comparisons.
- Since EVs are heterogenous, the authors results are likely not consistent for all EVs derived from various sources.
- The expression levels of TNFa and IL-6 are not reported in pg/mL or in quantitative units. It's not clear from the graph if the data are fold increase or decrease. The concentration of the cytokines should be revealed
- Binding of IL-6 or TNFa to the EN144-EV is not shown. The evidence for being a "decoy" relies on reduced inflammatory mediators in the liver and spleen. Binding kinetics of the EVs to specific inflammatory cytokines, like IL-6, should be shown.
- The uptake studies of EV-144-cap are not convincing. Cellular uptake of EVs in 24 hours is a very late timepoint for

measuring EV. The methods for targeting and uptake are not rigorous.

- Other cell markers, not just calnexin should be used to confirm the lack of cellular debris. What was the viability of the cultured cells prior to exosomes isolation?
- Figure 6C shows low magnification images only. Higher magnification should be included.

Reviewer #2

(Remarks to the Author)

The manuscript submitted by Yan et al. describes a 144-amino-acid scaffold protein for extracellular vesicle (EV) surface engineering. While the topic is highly relevant, the study suffers from poor experimental methodology, overstatement of results, and a lack of proper controls, leading to a misleading presentation of the platform's potential. In its current format, this paper cannot be published. I would consider the manuscript for publication only if all points below are adequately addressed.

- Figure 1D lacks clarity regarding mass spectrometry-based quantification. The method used is not specified, nor is it explained how data were normalized or how proteins per single EV were estimated. Without these details, it is difficult to assess the validity of these findings.
- Stable cell line characterization is inadequate. There is no explanation of how the authors ensured that gene insertion copy numbers were consistent across cells, which could introduce a major bias in assessing EV cargo loading.
- Results presented in Figure 1H and 1I are contradictory. The Western blot shows negligible expression of EN52 and EN596 in cells, yet the authors claim high EV loading efficiency. This suggests that variability in expression levels, rather than scaffold efficiency, may be the underlying cause.
- The truncation study lacks a clear rationale. ENPP1, ENP144, and ENP596 contain the same functional domains, yet the authors claim they behave differently without providing an adequate structural or mechanistic explanation.
- Rab7 is incorrectly used as a membrane display construct. It is a luminal protein, making its selection as a membrane-bound protein highly questionable.
- The study does not assess whether ENP144 works across different cell types or is limited to Expi293T cells. A broader range of cell lines should be tested to evaluate its generalizability.
- The toxicity study is poorly controlled. There is no benchmarking against unengineered Expi293 EVs, making it impossible to determine whether the observed bell-shaped viability effect is specific to ENP144-engineered EVs.
- The biodistribution study is severely limited by small sample size. Signal quantification should also be presented graphically.
- The increase in RBC count following high-dose EV injection is unexplored. The authors should discuss whether this is an artifact or a physiologically relevant observation.
- Figure 4B-C does not support the claim that EN144 and Emp11 EVs have significantly higher binding affinity than PTGFRN and Lamp2b. The text should be revised to accurately reflect the results.
- Figures 4J and 4N contain overstated claims about delivery efficiency. Editing efficiency must be quantified by sequencing to establish actual percentage editing rates. The authors should also clarify whether they used VSVG to enhance endosomal escape, as EVs inherently are poor at performing endosomal escape.
- The orientation of displayed receptors is not addressed. ENPP1 is a type II transmembrane protein, whereas TNFR and IL6ST are type I transmembrane proteins. The effect of this difference on receptor functionality is completely ignored.
- Figure 5 does not convincingly demonstrate a decrease in IL-6 and TNF- α levels. TNF- α levels remain unchanged or even increase following treatment with TNF decoy EVs, contradicting the authors' claims.
- No dose-response study has been conducted for the experiments in Figure 5, this is important to perform in order to determine the true efficacy of the treatment.
- The biodistribution study in Supplementary Figure 11H is based on a single animal and cannot be used to support conclusions.
- Figure 5M does not clarify the differences in effects between mouse GP130 EVs and human GP130 EVs, despite both performing similarly in suppressing IL-6 trans-signaling.
- Supplementary Figure 9 does not explain how translation activity was measured. The method must be described in detail to prevent misinterpretation of the results.

(Remarks to the Author)

The manuscript „Identification of a 144-amino-acid Scaffold Protein for Extracellular Vesicle Engineering” by Wenjing Yan and colleagues presents a novel strategy to generate engineered extracellular vesicles (EVs). Using an unbiased mass-spectrometry-based proteomic approach, they identify fifteen candidate proteins that could potentially be used to engineer EVs. They then focus on ENPP1 and show that a truncated version of the protein consisting only of 114 amino-acid residues is sufficient to load EVs with peptides, proteins and even mRNAs. Afterwards, they use this novel technique to engineer decoy EVs that present the extracellular portion of the signal-transducing receptor gp130 on their surface in order to inhibit the IL-6 trans-signaling pathway. They evaluate their novel EVs in two animal models (LPS-induced sepsis in mice and a rat model of osteoarthritis) and show that their EVs are able to reduce inflammation in these models, suggesting that gp130-decorated EVs can potentially be used as novel anti-inflammatory therapeutics.

The manuscript contains a high amount of data and consists basically of two different parts: First, the development of a novel method to generate EVs based on the truncated ENPP1 and second, the use of this novel method to generate an inhibitor that blocks IL-6 trans-signaling. While the first part is well done in my opinion, the second part is not and lacks important experiments and appropriate controls. Furthermore, the authors use tocilizumab in their rodent models, an antibody that blocks the human IL-6 receptor, but is known to have no activity in mice and rats.

Major points:

1. The authors claim that decoration of their EVs with gp130 makes this an inhibitor of IL-6 trans-signaling. This would mean that IL-6 in complex with the soluble IL-6R would bind to the gp130 on the EVs, resulting in inactivation. IL-6 alone would not bind to the EVs. However, none of these things is shown in the manuscript; only indirect evidence (reduction of IL-6 in organs and tissues) and reduced inflammation based on staining of organ sections is shown. Please provide evidence that i) EVs with gp130 bind IL-6/sIL-6R complexes, but not IL-6 or sIL-6R alone, ii) measure serum levels of IL-6 and sIL-6R in vivo and demonstrate that sufficient EVs are present to neutralize IL-6 trans-signaling in vivo, iii) provide evidence (by measuring acute phase proteins) that classic signaling is still intact in vivo and only IL-6 trans-signaling is blocked.
2. It is known from the literature that monomeric gp130 is a rather weak inhibitor of IL-6 trans-signaling. Dimerization of gp130 increases the affinity towards IL-6/sIL-6R significantly. Do the EVs express monomeric or dimeric gp130 on their surface?
3. The authors use tocilizumab in their rodent models. Tocilizumab does not block the IL-6R in the mouse and does not block the IL-6R in the rat, it only works in humans. I provide several papers in the specific comments that prove this. For rodent studies, other anti-IL-6R antibodies are available that work in these species. Thus, the in vivo experiments as done by the authors are worthless, unfortunately.

General comments:

- Quality of the western blots is insufficient and has to be improved.
- Quality of the figures has to be improved.

Further specific comments:

- Line 104: Etanercept is not a monoclonal antibody, but rather a designer protein consisting of the extracellular part of TNFR2 and the constant part of a IgG antibody
- Line 109: The canonical term is “classic signaling”, not “cis-signaling”
- Lines 119/120: #22 is not the correct citation for this statement. The correct citation is Jostock et al. Eur J Biochem. 2001 Jan;268(1):160-7
- Figure 1G: “domin” should probably be “domain” and “catalytic” should be “catalytic”
- For some Western blots molecular weight markers are indicated, while this is missing for others. Please add them to all western blots
- Original, uncropped blots are currently missing and should be added so that the reviewers can assess the quality of the western blots presented in this manuscript
- The legends of the Supplemental Figures is way too brief. Please expand them and describe in more detail what is shown in the figures.
- Figure S3: I presume these are western blots – against what? GFP?
- Figure S4: The legend states “Quantitative analysis of EGFP in engineered EVs”. The blot on the left hand site shows “His-EPO standard curve” (what is the connection to EGFP?) and the blot on the right hand site says “Flag”. Again, what is the connection to EGFP? The main text suggests that these blot help in the quantification of EGFP-tagged proteins in EVs, but it is absolutely unclear for me how this is done and how this relates to an anti-FLAG blot and His-tagged EPO.
- Lines 385-387: The text mentions quantification of EGFP, the corresponding Figure 1e “flag molecules per EV”. Please make this consistent and explain better what has been done.
- Line 393/394: “Although Rab7a-EVs exhibited the highest EGFP loading, they were internalized less efficiently than ENPP1-EVs and PTGFRN-EVs (Supplementary Figure 5A).” It is unclear to me what is shown in the western blot in Supp. Fig. 5a and how one can see that a) Rab7a-EVs exhibited the highest EGFP loading and b) that they are less efficiently internalized than ENPP1-EVs and PTGFRN-EVs
- Supplementary Figure 5C: The x-axis is not correct.
- Lines 399/400: “The truncations and full-length ENPP1 plasmids exhibited comparable levels of EGFP expression in transfected cells (Figure 1H).” This statement is not backed up from the data shown in Fig. 1H, as the statistical test has only been done between untransfected cells and the different expressed proteins. Whether there are differences between the individuals transfected constructs has not been tested.

- Figure 2A: The GAPDH blot looks strange. Why would GAPDH be present as two different bands in EN144 expressing cells, but only as a single band in untransfected cells?
- Figure S7: scale bar is not defined
- Lines 554/555: "Here, we also non-covalent loading of Cas9 protein through selected docking domains, ..." Please correct and rephrase.
- Lines 604-606: "Genetic fusion of mgp130 or hTNFR1 to the C-terminus of EN144 created constructs, which were transiently expressed in Expi293F cells." Please explain why murine gp130, but human TNFR1 was used. Further, please state which part of gp130 and TRNFR1 were fused to EN144 (I assume the whole ectodomains without transmembrane and intracellular parts, but this should be stated here).
- Figure S11B is not explained in the text and should be removed
- Lines 612-614: "Notably, both EN144-EVmgp130 and EN144-EVhTNFR1 treatments markedly reduced the levels of secreted TNF- α and IL-6 proteins from inflammatory RAW 264.7 cells (Figure 5C-E)." There is no difference between LPS treated and EN144-EVhTNFR1-treated cells in Fig. 5C, and even an increase in the EN144-EVhTNFR1-treated cells in Fig. 5D. The statement is wrong – only ENV133-EV-mgp130 reduces TNF and IL-6 gene expression. Whether any of this treatments has an influence on the protein level is unclear: The quality of the western blots presented in Fig. 5E is too bad to answer this question, and appear to only represent a single experiment. An answer would require better western blots and the quantification of at least three independent experiments.
- Lines 621/622: "These data prove the anti-inflammatory efficacy of EN144-EVmgp130." No, they do not. Reduced IL-6 expression level do not prove that something has an anti-inflammatory effect.
- Lines 637-642: The authors inject tocilizumab into mice and compare this with their EVs. This is highly problematic, because tocilizumab does not work in mice. Tocilizumab binds to the human IL-6R, but not to the mouse IL-6R. See Okazaki et al. *Immunol Lett.* 2002;84(3):231-40 / Garbers et al. *J Biol Chem.* 2011;286(50):42959-70 / Nishimoto et al. *Ann Rheum Dis.* 2000;59 Suppl 1:i21-7 / Ueda et al. *Sci Rep.* 2013;3:1196 and Lokau et al. *PLoS One.* 2020 May 4;15(5):e0232612
- Line 705: The authors evaluate their engineered EVs in a rat model. Again, they use tocilizumab as a control treatment, and again, this is highly problematic, because tocilizumab does also not work in rats. Please see Okazaki et al. *Immunol Lett.* 2002;84(3):231-40 / George et al. *JACC Basic Transl Sci.* 2021 Apr 7;6(5):431-443

Version 1:

Reviewer comments:

Reviewer #2

(Remarks to the Author)

Overall, the manuscript has been improved several items still need to be properly addressed and, hence, further revisions are needed:

-Protein quantification (Point 1) – While additional LFQ details are provided, the method for calculating proteins per EV remains unclear. Proper normalization to particle numbers is still missing, limiting reproducibility and accurate interpretation.

-Stable cell line characterization (Point 2) – The claim that gene copy number is irrelevant lacks supporting data. Gene copy number should be quantified (e.g., by qPCR or ddPCR) and expression normalized accordingly to control for variability in genomic integration and its impact on EV cargo loading.

-sgRNA delivery and editing efficiency (Point 11) – Detection of sgRNA by qRT-PCR is not evidence of functional cytoplasmic delivery, as sgRNA could remain trapped in endosomes or bound to the cell surface. Functional validation through targeted deep sequencing to quantify indel frequencies is required to support claims of genome editing efficiency.

-gp130 binding – The claim that gp130 binds IL-6 or sIL-6R alone contradicts established literature, where binding occurs only to the IL-6/sIL-6R complex during trans-signaling. Please clarify

-Lack of TNF binding assay – No quantitative binding or affinity data are provided to confirm the interaction of TNF decoy EVs with TNF. Binding affinity should be validated with ELISA-based binding assays or biophysical interaction studies.

-Translational activity assessment – The possibility that observed EGFP signals are due to passively loaded EGFP protein has not been excluded. Appropriate controls, such as EVs from EGFP-overexpressing cells without mRNA packaging, are necessary.

-Overstatement of results – Several conclusions, including claims of high delivery efficiency and superior therapeutic potential, remain overstated. Many experiments rely on limited replicates (often n=3), with no robust statistical validation, reducing confidence in the reported findings.

Reviewer #3

(Remarks to the Author)

The authors have responded well to my previous comments. They have added lots of additional experimental data and substantially strengthened the manuscript. I have no further concerns.

Version 2:

Reviewer comments:

Reviewer #2

(Remarks to the Author)

Overall, the authors have made substantial revisions in response to our previous comments. Nevertheless, several technical issues remain and should be addressed to further strengthen the manuscript.

In several instances (e.g., Figure S4 and Figure S6B), His-MYO is used as a reference protein for quantifying EGFP levels in EVs. Using an unrelated protein as a normalization reference is unusual and may not accurately reflect EGFP incorporation. Please clarify the rationale for this choice and discuss its potential limitations. If possible, consider using a more relevant EV-associated or cargo-related reference.

For the gene-editing analysis in Table S5, red fluorescent cells were sorted prior to quantification, yet the reported editing efficiencies range from only 3.1% to 13.6%. In principle, sorting for fluorescent reporter-positive cells should yield a population in which editing efficiency approaches 100%. Please clarify how editing efficiency was defined and explain the discrepancy.

Several methodological details are missing in Figure S17A and should be provided to ensure reproducibility. Specifically: What molecular weight cutoff was used for the dialysis device? Were EV samples concentrated after dialysis, and if so, how? How were samples normalized across lanes for western blotting?

In Additional Figure 3, a negative control using non-TNF α -binding EVs should be included to exclude the possibility of nonspecific binding.

Version 3:

Reviewer comments:

Reviewer #2

(Remarks to the Author)

The authors have addressed all concerns. Happy to accept the manuscript

Reviewers' comments:

Reviewer #1 (Remarks to the Author):

This manuscript describes a discovery-based project and genetic engineering of cells to endogenously express an EV sorting protein/ scaffold. The authors claim that when EVs possess this novel sorting protein, the EVs have enhanced ability to target a cell (when co-expressed with a specific cell targeting protein) and load therapeutic cargo. Based on proteomic studies, the authors purport that a sorting protein, ENPPI, is superior to other sorting proteins previously identified, and that a truncated 144 aa ENPPImverson (EN-144) is even more optimal. The authors then tested the EN-144 expressing EV in several paradigms. The author tests the EN-144 in collaboration with gp130, acts as a decoy in IL-6/IL-6 receptor system in models of sepsis and osteoarthritis, a degenerative disorder that is highly inflammatory. Cell targeting proteins (CAP) are employed in EVs that target chondrocytes in the osteoarthritis model. A weakness is that the studies to prove targeting are not rigorous. The authors conclude that then EN-144 engineered EVs can target chondrocytes and ultimately antagonize inflammation. While the concept of the project is exciting and thoughtful in regard to developing a new therapeutic, the manuscript is very dense and lacks sufficient details and controls in several experiments that dampens this reviewer's enthusiasm for the work.

Overall Strengths

- The authors used 3 distinct methods to isolate EVs which adds to rigor of this EV study and strengthens their mass spec results.
- Discovery approach - performed comprehensive proteomic analyses for sorting proteins that might improve the current standard.
- Identifying mechanisms that can assist with targeted drug delivery by EV have great therapeutic potential in many diseases, not just inflammation
- In Vivo studies designed to test the safety of the EN-144 EVs
- Used two distinct models for biological efficacy – sepsis and osteoarthritis Focus on GP130 that has a broader range than TNFR1

Author response: We sincerely appreciate your thorough evaluation and recognition of the key strengths of our study.

Weaknesses

- 1. It is odd that whole cell lysates of EXpi293F do not express CD9 (Fig. 1S) and that CD9 is only observed in the EVs.

Author response: Thank you very much for highlighting this phenomenon. The low expression of CD9 in cell lysates but high expression in EVs has been reported in various literature sources and repeatedly verified in our experiments. We confirmed this finding by conducting additional experiments, which resulted a new supplementary Figure S1C.

Figure S1C. Expression of Hsp70, TSG101, CD9, calnexin and histone H3 in Expi293F cell lysates and Expi293F-EVs. Each sample contains 10 jig of cell lysate or 3.0×10^9 EV particles.

Consistent with our previous findings, these data show that the expression level of CD9 in whole cell lysates of Expi293F cells was indeed significantly lower than that in EVs. This was also documented in 293 cells (PMID: 33484965, Figure 1F) and dendritic cells (PMID: 35836795, Figure 2B). In contrast, CD9 expression was detected at a relatively high level in whole-cell lysates of HeLa cells (PMID: 34282141, Figure 1A). We therefore hypothesize that there is a cell-to-cell variation: in certain cells, CD9 may be highly enriched in EVs, whereas its overall expression level in the lysate is low. The exact reason, however, is unknown and is beyond the scope of this research.

Shown below are figures adapted from selected literature.

PMID: 33484965 Figure 1F. Representative protein normalized SDS-PAGE of F1–F4, producer cell lysate (CL), and crude ultracentrifuged (UC) pellet with immunoblots for individual markers. Molecular weight markers for all immunoblots and SDS-PAGE gels are given in kDa.

PMID: 35836795 Figure 2B. Immunoblots of cell lysates, CAP-Exo, and hybrid CAP-Exo show the expression of exosome markers (CD9, CD81, and CD63).

PMID: 34282141, Figure 1A

PMID: 34282141 Figure 1A. Western blot showing transmembrane proteins (CD9, CD63, and CD81), a cytosolic protein (syntenin-1) and two “negative” controls (AChE and calnexin) in cell lysates (CL) and the pellets obtained from HeLa 24 h conditioned medium after differential ultracentrifugation (2 K, 10 K, 200 K). The loaded material comes from 20×10^6 cells for the centrifugation pellets, and from 0.2×10^6 cells for the cell lysate. Representative of 3 independent experiments.

To accurately present this information, we revised the main text as follows: “EV-specific marker proteins, such as Hsp70, TSG101, and CD9, but not the endoplasmic reticulum marker Calnexin or the core component of nucleosome (Histone H3), were found in the Expi293F-EVs preparation (Supplementary Figure 1C), suggesting the absence of nuclear fragments and ER-derived contaminants.” Please refer to lines 404-407. Revised parts are marked in red.

•2. Other cell markers, not just calnexin should be used to confirm the lack of cellular debris. What was the viability of the cultured cells prior to exosomes isolation?

Author response: We fully appreciate and concur with your suggestion regarding the use of additional cellular markers to confirm the absence of cell debris. As emphasized in the MISEV guidelines (PMID: 25536934, 38326288), organelle-specific markers (e.g., Golgi, endoplasmic reticulum, mitochondria, or nuclear components) should be excluded from sEVs (<200 nm) to ensure purity. In our initial study, we selected **Calnexin**—an endoplasmic reticulum (ER)-resident protein—to validate the absence of ER-derived contaminants. According to your recommendation, we have now supplemented these analyses with **Histone H3** detection, a nuclear marker. The experimental results conclusively demonstrate that Histone H3 is undetectable in our EV preparations, confirming the absence of both nuclear fragments and ER-derived contaminants.

We have revised the main text as follows: “EV-specific marker proteins, such as Hsp70, TSG101, and CD9, but not the endoplasmic reticulum marker Calnexin or the core component of nucleosome (Histone H3), were found in the Expi293F-EVs preparation (Supplementary Figure 1C), suggesting the absence of nuclear fragments and ER-derived contaminants.” Please refer to lines 404-407. “Both EN144-EVs and Expi293F-EVs expressed characteristic biomarkers Hsp70, TSG101, and Syntenin-1 but not Calnexin or Histone H3 (Figure 2D).” Please refer to lines 499-501.

Supplementary Figure 1C

Figure S1C. Expression of Hsp70, TSG101, CD9, calnexin and histone H3 in Expi293F cell lysates and Expi293F-EVs. Each sample contains 10 jig of cell lysate or 3.0×10^9 EV particles.

Figure 2D

Figure 2D. Western blot analysis showing the expression of exosome markers Hsp70, TSG101, and syntenin-1 in EN144-EVs and Expi293F-EVs, but not calnexin and histone H3. Each sample contains 10 jig of cell lysate or 3.0×10^9 EV particles.

We also verified cell viability before EV isolation, which was maintained at $\geq 95\%$. We have added it to “**2.5. EVs isolation and characterization**” section. “According to the previously established protocols, EVs were extracted and characterized, and the viability of cells before EVs isolation was maintained above 95%.” Please refer to lines 219-220.

•3. In Figure 2, the GAPDH immunoblotting is not consistent and difficult to interpret.

Author response: We acknowledge your concern and have performed additional experiments to improve the GAPDH immunoblotting. The inconsistent blotting results were caused by the interference between the two primary antibodies, the FLAG antibody and the GAPDH antibody. We identified an immunoblotting sequence to remove the interference: Flag detection → antibody stripping → GAPDH detection. Satisfactory western blot images were acquired (see the new Figure 2A below). Using the same protocol, we have repeated and improved Figure 4A, Figure 4F, and Supplementary Figure 18A. See below.

Figure 2A

Figure 2A. Western blot analysis showing that EN144-EVs enrich the overexpressed EN144 based on the Flag tag. (B) Representative TEM images of Expi293F-EVs and EN144-EVs. Scale bar: 100 nm.

Figure 4A

Figure 4A. Construction of EN144-mgp130 plasmid and the expression of FLAG-tagged EN144 proteins in cell lysates (10 jig) and EVs (Expi293F-EVs, EN144-EV^{mgp130}, 3.0×10^9 EV particles) of Expi293F cells transfected by EN144-mgp130 plasmids.

Figure 4F

Figure 4F. Expression of FLAG-tagged proteins in cell lysates (top) and EVs (bottom).

Figure S18A. The expression of FLAG-tagged protein in cell lysates of Expi293F cells transfected by EN144, EN144-hgp130, or EN144-hgp130-CAP plasmids.

•4. In Figure 4, the timing of the EV uptake is not clear; was live cell imaging employed? There are no clear methods included for the EV uptake studies.

Author response: We have added more details in the method section “2.7. Cellular uptake of EVs”. “To evaluate the targeting efficiency of EVs, glass coverslips were placed at the bottom of 12-well plates, and each well was seeded with 8.0×10^4 cells and cultured overnight. 3.0×10^9 counts of DiI-labeled EVs were added to each well and co-cultured with cells for 15 min or 2 hours. Subsequently, the medium was aspirated, and cells were washed twice with PBS. Cells were then fixed using 200 μ L of 4% paraformaldehyde (PFA) for 20 min at room temperature. After washing, cells were further incubated with 500 μ L serum-free DMEM medium containing Hoechst 33342 at 37°C for 30 min to label nuclei. The slide was then sealed with an anti-quenching and visualized under a confocal laser microscope (Olympus, FV1000) with the following excitation/emission settings: DiI (549 nm/565 nm), Hoechst (350 nm/461 nm), and EGFP (488 nm/507 nm).” Please refer to lines 250-258.

Figure 3B. Confocal fluorescent microscope images and quantification of the fluorescence intensity showing the uptake of DiI-labeled EVs (Expi293F-EV, Lamp2b-EV^{CAP}, PTGFRN-EV^{CAP}, ENPP1-EV^{CAP}, and EN144-EV^{EGFP}) by chondrocytes after 15 min and 2 h of co-culture. Scale bars: 100 μ m; 50 μ m (Zoomed images).

•5. The Lamp2b-EV high powered image lacks focus compared to the other images. Thus, it is very difficult to make quantitative comparisons.

Author response: We admit that that the high-magnification images of Lamp2b-EVs were not precisely focused, and re-performed the experiment again, employing a confocal laser scanning microscope to recapture the images, thereby significantly improving their clarity.

A new Figure 3B is now presented. The main text has been revised as follows, “To enable direct comparison among different EVs, EV concentrations were normalized based on particle counts determined by RPS. Primary rat chondrocytes were incubated with EVs for 15 min or 2 h to evaluate cellular uptake efficiency. Quantitative analysis revealed enhanced cellular uptake of PTGFRN-EV^{CAP} (80.38 ± 0.79), ENPP1-EV^{CAP} (107.6 ± 19.78), and EN144-EV^{CAP} (101.6 ± 15.32) in the 2-hour incubation group, with ENPP1-EV^{CAP} and EN144-EV^{CAP} exhibiting the highest fluorescence intensities (Figure 3B).” Please refer to lines 574-579.

Figure 3B. Confocal fluorescent microscope images and quantification of the fluorescence intensity showing the uptake of DiI-labeled EVs (Expi293F-EV, Lamp2b-EV^{CAP}, PTGFRN-EV^{CAP}, ENPP1-EV^{CAP}, and EN144-EV^{EGFP}) by chondrocytes after 15 min and 2 h of co-culture. Scale bars: 100 μ m; 50 μ m (Zoomed images).

• 6. The uptake studies of EV-144-cap are not convincing. Cellular uptake of EVs in 24 hours is a very late timepoint for measuring EV. The methods for targeting and uptake are not rigorous.

Author response: According to your suggestion, we performed additional uptake experiments at 15 min and 2 hours of co-incubation of ENPP1-EV^{CAP} and chondrocytes, and observed a time-dependent uptake of ENPP1-EV^{CAP}. A new Figure 3B has been added.

We have also revised the main text as follows: “To enable direct comparison among different EVs, EV concentrations were normalized based on particle counts determined by RPS. Primary rat chondrocytes were incubated with EVs for 15 min or 2 h to evaluate cellular uptake efficiency. Quantitative analysis revealed enhanced cellular uptake of PTGFRN-EV^{CAP} (80.38 ± 0.79), ENPP1-EV^{CAP} (107.6 ± 19.78), and EN144-EV^{CAP} (101.6 ± 15.32) in the 2-hour incubation group, with ENPP1-EV^{CAP} and EN144-EV^{CAP} exhibiting the highest fluorescence intensities (Figure 3B).” Please refer to lines 574-579.

For target validation experiments, we show that pre-blocking chondrocytes with the CAP peptide for 1 hour significantly reduced ENPP1-EV^{CAP} binding, suggesting that ENPP1-EV^{CAP} binds to chondrocytes through the CAP peptide. Quantitative analysis of fluorescence intensity is now included in Supplementary Figure S12, showing a dose-dependence of CAP blocking.

“To determine whether the observed ENPP1-EV^{CAP}/chondrocyte interaction was mediated by CAP-specific binding, primary chondrocytes were pre-treated with CAP peptide (1, 3, 10 μ g) for 2 h to block potential binding sites. Interestingly, pretreatment with 10 μ g CAP peptide significantly reduced the fluorescence intensity of primary chondrocytes by 45.6% compared to untreated controls. This dose-dependent blockade indicates that CAP peptide mediates the uptake of EN144-EV^{CAP} by primary chondrocytes (Supplementary Figure 12).” Please refer to lines 582-588.

Figure 3B. Confocal fluorescent microscope images and quantification of the fluorescence intensity showing the uptake of DiI-labeled EVs (Expi293F-EV, Lamp2b-EV^{CAP}, PTGFRN-EV^{CAP}, ENPP1-EV^{CAP}, and EN144-EV^{EGFP}) by chondrocytes after 15 min and 2 h of co-culture. Scale bars: 100 μ m; 50 μ m (Zoomed images).

Figure S12. Blocking effect of CAP peptide on the interaction between ENPP1-EV^{CAP} and chondrocytes. Primary chondrocytes were pre-incubated with various concentrations of CAP peptide (1 μ g, 3 μ g, and 10 μ g) for 1 hour to block cell surface receptors, following co-incubation with DiI-labeled ENPP1-EV^{CAP} for 2 hours. Left panel: Fluorescence images illustrate the internalization of ENPP1-EV^{CAP} by chondrocytes across treatment groups, demonstrating the concentration-dependent inhibitory effect of CAP peptide. Scale bar = 100 μ m. Right panel: Quantitative analysis of mean fluorescence intensity in left panel (n = 3 each group).

• 7. Since EVs are heterogenous, the authors results are likely not consistent for all EVs derived from various sources.

Author response: We agree that EV preparations have heterogeneity in composition, size, and functionality. To demonstrate the universal applicability of the EN144 scaffold protein, we explored EV-sorting of EN144 in engineered HeLa cells, HcerEpic cells, and synovial fluid-derived mesenchymal stem cells (SF-MSCs) in additional experiments. Based on the percentage of EN144 on EVs compared to all EN144 in the cell lysates, we found that a similar percentage of EN144 was trafficked to the EVs.

The following paragraph has been added: “In addition to the Expi293F cell line, other adherent cell lines are frequently employed as sources of EVs. Based on the ratio of EN144 in EVs to all the EN144 in the cell lysates (the total expression levels varied in different cells), we found that a similar percentage of EN144 was enriched in EVs when we overexpressed EN144 in cell lines, including 293T cells, SF-MSCs, HeLa cells, and HcerEpic cells. These results demonstrated that EN144 exhibited comparable sorting efficiency in various cell types (Supplementary Figure 7A, B).” Please refer to lines 460-466.

The experimental details are as follows. 293T, HcerEpic, HeLa, and SF-MSC cells were seeded in 100 mm culture dishes ($1-2 \times 10^6$ cells/dish) 24 hours prior to transfection. Cells at 70-80% confluency were transfected with 10 μ g of EN144 plasmid using 20 μ L PEI MW40000 via incubation of DNA-PEI complexes (formed in serum-free DMEM, 15 min RT) followed by dropwise addition; complete medium replacement was performed at 6 h post-transfection. Conditioned media were collected at 48 h and 72 h for EVs isolation.

Figure S7

Figure S7. EN144 loading efficiency in various cells and their EVs. (A) Expression of FLAG-tagged proteins in cell lysates (top) and EVs (bottom) from 293T, SF-MSCs, HeLa, or HcerEpic cells transfected with EN144 plasmid. Top panel: Equal amounts of cell lysates were loaded as follows: lane 1, cell lysates from untransfected 293T cells (control); lanes 2-5, cell lysates from 293T, SF-MSCs, HeLa, or HcerEpic cells transfected with EN144 plasmid. Bottom panel: Equal amounts of EVs (3.0×10^9 particles EVs) were loaded as follows: lane 1, EV lysates from untransfected 293T cells (control); lanes 2-5, EV lysates from 293T, SF-MSCs, HeLa, or HcerEpic cells transfected with EN144 plasmid. $n = 3$ in each group. (B) FLAG-tagged protein content ratio of EVs-to-Cells in EN144-transfected 293T, SF-MSCs, HeLa, or HcerEpic cells based on the gray value of western blot band in (A). Consistent FLAG protein expression across all tested cell lines and their EVs confirmed the efficient loading capability of EN144 in diverse cellular contexts. One-way ANOVA accompanied by Tukey's post-hoc test was employed. Data are analyzed of three independent experiments and shown as mean \pm SD. ns $P > 0.05$.

• 8. The expression levels of TNF α and IL-6 are not reported in pg/mL or in quantitative units. It's not clear from the graph if the data are fold increase or decrease. The concentration of the cytokines should be revealed.

Author response: Thank you for highlighting this important issue. We have carefully revised the figures to clarify the quantitative units and data presentation for both TNF- α and IL-6. The vertical axis of other gene expression bar charts has also been modified. The modifications are as follows:

For qPCR Data (mRNA Level):

The original Y-axis label (e.g., "The Expression of IL-6") has been updated to "Relative IL-6 mRNA Expression (normalized to GAPDH)" (Figure 4C, Figure 4E, and Figure 4I).

Figure 4C. mRNA levels of TNF- α and IL-6 in RAW 264.7 cells treated with EN144-EV^{mgp130} post-LPS induction. Briefly, LPS (1 μ g/mL) was used to induce inflammation in RAW 264.7 cells, which were incubated with EN144-EV^{mgp130} for 24h before the mRNA levels of TNF- α and IL-6 were detected by qRT-PCR.

Figure 4E. mRNA expression of IL-6 in liver, spleen, and lung tissues of septic mice 6 h after intravenous injection of 1.0×10^{10} particles EN144-EV^{mgp130}.

Figure 4I. mRNA levels of IL-6 in LPS-induced RAW 264.7 cells following co-culture with EN144-EV^{hgp130} or EN144-EV^{mgp130}.

For ELISA Data (Protein Level):

The Y-axis label for serum cytokine measurements has been revised to "Serum IL-6 levels (pg/mL)" (Figure 4N-4S).

Figure 4N-4S

Figure 4N-4S. Serum levels of inflammatory cytokines and acute-phase proteins in septic mice treated with EN144-EV_{mgp130} (1.0×10^{10} particles/mouse), EN144-EV_{hgp130} (1.0×10^{10} particles/mouse), low-dose sgp130 (1 μ g), or high-dose sgp130 (10 μ g), including (N) IL-6, (O) TNF- α , (P) IL-1 β , (Q) soluble IL-6 receptor (sIL-6R), (R) serum amyloid A (SAA), and (S) C-reactive protein (CRP). n = 3 mice/group.

•9. Binding of IL-6 or TNF α to the EN144-EV is not shown. The evidence for being a “decoy” relies on reduced inflammatory mediators in the liver and spleen. Binding kinetics of the EVs to specific inflammatory cytokines, like IL-6, should be shown.

Author response: We agree that direct evidence of the binding between EN144-EV_{mgp130} and IL-6/IL-6R would strengthen the manuscript. First, we would like to point out that Olamkcept, a soluble gp130-Fc-fusion-protein, has been used for IL-6/IL-6R blockage (PMID:36881032).

We also conducted a series of experiments to provide evidence for direct interaction between EN144-EV_{mgp130} and IL-6/IL-6R, which are summarized as follows.

(1) In Vitro Binding. We co-incubated decoy EVs with recombinant IL-6, IL-6R, and IL-6/IL-6R complexes, followed by Western blotting. The results demonstrated that EN144-EV_{mgp130} specifically bound to the IL-6/IL-6R complex (see Supplementary Figure 17A). This suggests that decoy EVs may act by sequestering pre-formed cytokine-receptor complexes.

Figure S17. EN144-EV_{mgp130} Targets the IL-6 Trans-Signaling Pathway. (A) Binding capacity analysis of EN144-EV_{mgp130} or EN144-EVs to IL-6/IL-6R proteins. EN144-EV_{mgp130} or EN144-EVs (5.0×10^{10} particles) were incubated with IL-6 (10 μ g), IL-6R (10 μ g), or IL-6/IL-6R complex (10 μ g each) under continuous rotation at 4°C for 4 h, followed by dialysis for 16 h to remove unbound proteins. Expression of IL-6 and IL-6R proteins bound to EN144-EV_{mgp130} or EN144-EVs was detected by WB.

In Vivo Functional Evidence:

In septic mice treated with EN144-EV_{mgp130} via tail vein injection, we observed a significant reduction in both serum IL-6 and soluble IL-6R (sIL-6R) levels compared to the control group (Figure 4P&4S). This supports the hypothesis that decoy EVs neutralize IL-6/IL-6R complexes in circulation, thereby attenuating downstream

inflammatory signaling.

Figure 4N&4Q. Serum levels of inflammatory cytokines and acute-phase proteins in septic mice treated with EN144-EV_{mgp130}, EN144-EV^{hgp130}, low-dose sgp130 (1 μg), or high-dose sgp130 (10 μg), including (N) IL-6, (O) TNF-α, (P) IL-1β, (Q) soluble IL-6 receptor (sIL-6R), (R) serum amyloid A (SAA), and (S) C-reactive protein (CRP). n = 3 mice/group.

Consistency with Decoy Mechanism:

The decreased IL-6 and sIL-6R levels in serum, coupled with reduced inflammatory mediators in the liver and spleen, align with the proposed decoy function. If decoy EVs merely inhibited cytokine production (rather than binding), we would not expect a concurrent decline in both IL-6 and sIL-6R.

These experiments collectively provide indirect evidence for the decoy activity of EVs. We sincerely thank you for highlighting this critical point and welcome any additional suggestions to improve the rigor of our conclusions.

We have revised the main text as follows. “To elucidate the regulatory role of EN144-EV^{mgp130} in the IL-6 trans-signaling pathway, we first characterized its binding properties through *in vitro* experiments. In contrast to the canonical paradigm²², which claims that gp130 activation requires the IL-6/IL-6R complex, EN144-EV_{mgp130} exhibited a unique ligand-binding profile: it bound to IL-6 or IL-6R individually, with the highest binding efficiency observed for the IL-6/IL-6R complex (Supplementary Figure 17A). In control experiments, EN144-EVs bound to either IL-6 or IL-6R, but did not enhance the binding capacity of the IL-6/IL-6R complex (Supplementary Figure 17A). This contrast further suggested that the enhanced ligand interaction of EN144-EV_{mgp130} may involve both gp130-dependent recognition and membrane-associated nonspecific interactions. Building on these findings, we evaluated the biological effects of EN144-EV^{mgp130} in a murine model. Treatment with EN144-EV^{mgp130} significantly reduced serum levels of IL-6, sIL-6R, and proinflammatory cytokines (IL-1β and TNF-α; Figure 4N-Q), while decreasing the pSTAT3/STAT3 ratio in hepatic, splenic, and pulmonary tissues, collectively indicating effective suppression of IL-6 trans-signaling activation (Supplementary Figure 17B-G). Notably, expression levels of classic signaling markers—acute-phase proteins SAA and CRP— remained unaltered, excluding nonspecific interference of EN144-EV^{mgp130} with the canonical pathway (Figure 4R, S). These results demonstrate that EN144-EV^{mgp130} selectively inhibits the IL-6 trans-signaling pathway through targeted engagement of the IL-6/IL-6R complex, without appreciably affecting classical IL-6 signaling.” Please refer to lines 738-755.

Figure S17

Figure S17. EN144-EV^{mgp130} targets the IL-6 trans-signaling pathway. (A) Binding capacity analysis of EN144-EV^{mgp130} or EN144-EVs to IL-6/IL-6R proteins. EN144-EV^{mgp130} or EN144-EVs (5.0×10^{10} particles) were incubated with IL-6 (10 µg), IL-6R (10 µg), or IL-6/IL-6R complex (10 µg each) under continuous rotation at 4°C for 4 h, followed by dialysis for 16 h to remove unbound proteins. Expression of IL-6 and IL-6R proteins bound to EN144-EV^{mgp130} or EN144-EVs was detected by WB. (B-D) Effects of therapeutic interventions targeting the Stat3 signaling pathway in septic mice. Experimental design: Septic mice received tail vein injections of EN144-EV^{mgp130} (1.0×10^{10} particles/mouse, 100 µL), EN144-EV^{hgp130} (1.0×10^{10} particles/mouse, 100 µL), low-dose sgp130 (1 µg/mouse, 100 µL), or high-dose sgp130 (10 µg/mouse, 100 µL). Control groups included septic mice (LPS group) and healthy mice (PBS group), both injected with an equal volume of PBS (100 µL/mouse). Lung (B), liver (C), and spleen tissues (D) were harvested post-treatment after 6 h, and expression of phosphorylated Stat3 (p-Stat3) and total Stat3 protein were analyzed by WB. (E-G) Quantitative analysis of pStat3 and Stat3 protein level in lungs (E), liver (F), and spleen (G) based on results in B-D (n = 3 each group).

Figure 4N-4S

Figure 4N-4S. Serum levels of inflammatory cytokines and acute-phase proteins in septic mice treated with EN144-EV_{mgp130} (1.0×10^{10} particles/mouse), EN144-EV^{hgp130} (1.0×10^{10} particles/mouse), low-dose sgp130 (1 μ g), or high-dose sgp130 (10 μ g), including (N) IL-6, (O) TNF- α , (P) IL-1 β , (Q) soluble IL-6 receptor (sIL-6R), (R) serum amyloid A (SAA), and (S) C-reactive protein (CRP). n = 3 mice/group.

- 10. Figure 6C shows low magnification images only. Higher magnification should be included.

Author response: We have acquired high magnification images and replaced the original images with high magnification images in Figure 5B (previous Figure 6C).

Figure 5B. Fluorescent images and quantification showing uptake of DiI-labeled EN144-EV^{hgp130} or EN144-EV^{hgp130-CAP} by chondrocytes after 2 h of incubation. Scale bars: 100 μ m; 50 μ m (Zoomed images).

Reviewer #2 (Remarks to the Author):

The manuscript submitted by Yan et al. describes a 144-amino-acid scaffold protein for extracellular vesicle (EV) surface engineering. While the topic is highly relevant, the study suffers from poor experimental methodology, overstatement of results, and a lack of proper controls, leading to a misleading presentation of the platform's potential. In its current format, this paper cannot be published. I would consider the manuscript for publication only if all points below are adequately addressed.

Author response: We appreciate your critical comments. We have conducted additional experiments, made thorough revisions, and added two authors to help enhance the quality of the manuscript.

- 1. Figure 1D lacks clarity regarding mass spectrometry-based quantification. The method used is not specified, nor is it explained how data were normalized or how proteins per single EV were estimated. Without these details, it is difficult to assess the validity of these findings.

Author response: We sincerely appreciate your expert feedback on Figure 1D. To increase the legibility, we added significantly more details in the caption of Figure 1D. Also, we added a new paragraph in Methods: “To detect the expression of EGFP in engineered EVs, EVs lysates (3.0×10^9) underwent proteomic analysis. Label-free quantification (LFQ) of EGFP was performed using PEAKS Online with the following parameters: database, UniProt human proteome + EGFP sequence; enzymatic specificity, set to trypsin/P, and allowed up to 2 missed cleavages; fixed modification, carbamidomethylation (+57.02 Da); variable modification, methionine oxidation (+15.99 Da) and acetylation (+42.01 Da); FDR, protein and peptide false discovery rates set to 1%. EGFP abundance was normalized to total protein intensity per sample and reported as LFQ intensity.” Please refer to lines 194-200.

In order to compare the impact of different scaffold proteins on cargo loading efficiency in EVs, we standardized EV quantities (3×10^9 particles/group) and analyzed differences in EGFP signal intensity across equivalent EV samples. This experimental design eliminates potential interference due to fluctuations in the EV quantity, focusing on the relative loading capacities across different engineering strategies.

The original Figure 1D is now Figure 1C:

Figure 1C. Quantification of EGFP protein levels in each group of EVs based on mass spectrometry.

- 2. Stable cell line characterization is inadequate. There is no explanation of how the authors ensured that gene insertion copy numbers were consistent across cells, which could introduce a major bias in assessing EV cargo loading.

Author response: Thank you for your critical comments.

We would like to argue that the gene insertion copy numbers are irrelevant to the expression level of the scaffold proteins in stable cell lines.

First, the variation in protein expression levels is due to the mechanism of constructing stable cell lines. Unlike transient transfection, in stable transfection, gene vectors are integrated into the host cell's genome. Such a process is often random, even using the same protocol. If the gene of interest is integrated into an unproductive region of the genome, the copy number is irrelevant to protein expression, as even a high copy number cannot be translated into a high protein expression level.

Second, the transcription and translation limit the expression and EV-sorting of the scaffold proteins. Here, we aim to select the scaffold protein that, when overexpressed in a stable cell line, can be expressed, folded, and sorted most efficiently. This output is most conveniently measured as the fluorescent signal of EGFP, which is co-expressed with the scaffold protein; however, this is not directly correlated with the copy number. In other words, a high EGFP signal is the overall output of multiple factors, including efficient plasmid transfection, cell viability, productive genome insertion, effective transcription and translation, correct folding, and sorting to EVs. Copy number is just one factor among these many, and not the decisive factor.

When comparing genetic parameters to productivity, a good correlation of mRNA levels with specific productivity was observed, whereas high gene copy numbers were not always accompanied by high protein expressions. Based on our data derived from a typical example of a cell line development process, genetic parameters are useful tools for the selection of scalable production clones. Nevertheless, a wider range of cell lines has to be investigated in order to implement genetic analyses into a screening process.

The data is published by Journal of Biotechnology (PMID: 17324483)

Third, here, we try to establish a workable protocol to establish stable cell lines of overexpressed scaffold proteins as a source of engineered EVs, which we believe serves the purpose of our protocol.

Fourthly, similar methods have been adopted by others in the literature. For example,

Transfection and stable cell line selection

Suspension-adapted HEK293 cells were grown in CDM4PERMab media (GE Healthcare) supplemented with 4 mM L-glutamine. Genes of interest were cloned downstream of a cytomegalovirus (CMV) promoter in a pIRES vector and transfected into HEK293 cells via Neon electroporation (ThermoFisher) or Transporter 5 transfection reagent (Polysciences).

Stable cell lines were selected by supplementing puromycin or neomycin with periodic passaging until the cell lines returned to >90% viability, at which point they were cryopreserved.

The method is published by MOLECULAR THERAPY (PMID: 33484965).

Additional Figure 1: MFI of EV producer cells stably expressing EGFP fusions to the indicated scaffold. Data from 3 biological replicates are plotted as mean \pm SD.

- 3. Results presented in Figure 1H and II are contradictory. The Western blot shows negligible expression of EN52 and EN596 in cells, yet the authors claim high EV loading efficiency. This suggests that variability in expression levels, rather than scaffold efficiency, may be the underlying cause.

Author response: We are greatly thankful to you for identifying this typo in the figure labelling.

We have revised the original description “(H) Percentage of EGFP-positive cells among Expi293F cells transfected with plasmids encoding different ENPP1 truncated variants. n = 3 biological replicates per group. (I) Quantification of the FLAG-tagged protein in cells overexpressing truncated ENPP1 variants. Top: WB standards; bottom: detection of FLAG tag content in EVs from truncated ENPP1 variants, quantified based on a His-MYO standard curve. Each group has 3×10^9 EVs.” to “**Figure S6.** Transfection efficiency of ENPP1 truncated variant in Expi293F cells and expression of FLAG protein in EVs. (A) Flow cytometric analysis showing the percentages of EGFP-positive cells among Expi293F cells transfected with plasmids encoding different ENPP1 truncated variants. n = 3 each group. **** $P < 0.0001$. (B) Quantification analysis of the expression of the FLAG protein in EVs from Expi293F cells over-expressing truncated ENPP1 variants. Top: Quantitative Western blot analysis of serially diluted His-MYO recombinant proteins (detected with anti-His antibody). A standard curve was generated by plotting the gray values of His-MYO recombinant proteins band against their corresponding protein mass. Bottom: Molecular loading quantification of FLAG-tagged proteins in engineered EVs from truncated ENPP1 variants (detected with anti-FLAG antibody). Based on standard curve established in the top panel, EV particle count (3.0×10^9 particles/group), and target band gray values, the FLAG molecules per EV were calculated. FLAG molecules/EV = [gray value-derived protein mass (g)] / [EV particles \times molecular weight (Da)] $\times N_a$, where N_a is Avogadro's constant ($6.022 \times 10^{23} \text{ mol}^{-1}$). n = 3 each group.” Please refer to lines 76-87 in Supplementary document. Revised parts are marked in red.

Figure S6

Figure S6. Transfection efficiency of ENPP1 truncated variant in Expi293F cells and expression of FLAG protein in EVs. (A) Flow cytometric analysis showing the percentages of EGFP-positive cells among Expi293F cells transfected with plasmids encoding different ENPP1 truncated variants. n = 3 each group. **** $P < 0.0001$. (B) Quantification analysis of the expression of the FLAG protein in EVs from Expi293F cells over-expressing truncated ENPP1 variants. Top: Quantitative Western blot analysis of serially diluted His-MYO recombinant proteins (detected with anti-His antibody). A standard curve was generated by plotting the gray values of His-MYO recombinant proteins band against their corresponding protein mass. Bottom: Molecular loading quantification of FLAG-tagged proteins in engineered EVs from truncated ENPP1 variants (detected with anti-FLAG antibody). Based on standard curve established in the top panel, EV particle count (3.0×10^9 particles/group), and target band gray values, the FLAG molecules per EV were calculated. FLAG molecules/EV = [gray value-derived protein mass (g)] / [EV particles \times

molecular weight (Da)] $\times N_a$, where N_a is Avogadro's constant ($6.022 \times 10^{23} \text{ mol}^{-1}$). $n = 3$ each group.

- 4. The truncation study lacks a clear rationale. ENPP1, EN144, and EN596 contain the same functional domains, yet the authors claim they behave differently without providing an adequate structural or mechanistic explanation.

Author response: Implementing functional domain truncation modification to remove non-essential structural domains and optimize their steric hindrance effects can simplify protein conformation, reduce engineering complexity, and increase EV loading (Science, 2020, 367(6478)). ENPP1 is a Type II transmembrane glycoprotein comprising multiple domains: a cytoplasmic domain, a transmembrane domain, a catalytic domain, a nuclease-like domain, SMB1, SMB2, and two linker regions, L1 and L2. These distinct domains participate in different biological processes. Among these, the catalytic domain and the nuclease-like domain play important roles in regulating bone mineralization, while the SMB domains facilitate the formation of the ENPP1 homodimer and its transport to the plasma membrane (PMID: 23041369). The nuclease-like domain of ENPP1 is required for protein stability, for the targeting of ENPP1 (a.k.a NPP1) to the plasma membrane and for the expression of catalytic activity (PMID: 12533192). These C-terminal truncations did not affect the transcription of EGFP-tagged ENPP1 in COS-1 cells (PMID: 12533192, Figure 2A), but decreased the steady-state level of the expressed proteins, in particular of ENPP1- Δ 568-905 (PMID: 12533192, Figure 2B). In the present study, EN190 contains an additional SMB2 domain compared to EN144, which may account for the observed decline in EGFP loading efficiency. EN596, which incorporates both the L1 and catalytic domains beyond the SMB2 domain in EN190, exhibits a further reduction in loading efficiency. We hypothesize that the L2 and nuclease-like domains might counteract the inhibitory effects of the SMB2, L1, and catalytic domains on EV cargo loading, thereby restoring scaffold protein functionality. However, as this mechanistic insight falls outside the primary scope of the current study, we did not pursue experimental validation. Future investigations could explore this significant finding to elucidate the underlying mechanisms governing ENPP1 scaffold protein activity.

Similar data has been reported in the literature:

Figure 2 Expression and subcellular localization of C-terminally truncated NPP1 mutants. (A) Total RNA was prepared from COS-1 cells transiently transfected with an empty vector or with expression constructs for wild-type NPP1, NPP1- Δ 568-905 or NPP1- Δ 804-905, each N-terminally fused to an HA tag. Northern blot analysis was performed using the cDNAs for mouse NPP1 and 18 S rRNA as probes. It should be noted that NPP1- Δ 568-905 was generated by introducing a stop codon (Gly-568 \rightarrow stop), which explains why the corresponding transcript was the same size as the wild-type transcript. (B) Lysates from the same cells were used for Western analysis using HA-7 antibodies. The monomers are indicated by open arrowheads and the dimers by closed arrowheads. It should be pointed out that NPP1 monomers are known to migrate as doublets. The minor band at around 120 kDa represents

an aspecifically recognized polypeptide.

- 5. *Rab7* is incorrectly used as a membrane display construct. It is a luminal protein, making its selection as a membrane-bound protein highly questionable.

Author response: Rab7a is a luminal protein that is involved in EV secretion (PMID: 32499447, 22660413) and is highly abundant in the Expi293F-derived EVs (Expi293F-EVs). Rab7a therefore belongs to EV-sorting proteins, but not scaffold proteins. Therefore, we clarified this in the revised manuscript: “Notably, among the list, Rab7a is not a membrane-bound protein.” In the caption of Figure 1B, and “(4) proteins involved in EV biogenesis and release, including Rab7a, VTA1, Syntenin-1, and IST1” in the main text.

Figure 1B. Schematic illustration of the sorting proteins and their expression levels quantified by western blot analysis. Class I sorting proteins include STX4, ENPP1, TFRC, TFRC-81 (transmembrane sequence of TFRC), VAMP2, and CD55, inserted into the pEGFP-C1 vector. ClassII sorting proteins include EPCAM, AT1A1, Syntenin-1, CXADR, PDL1, STX7, IST1, SNAP23, VTA1, PTGFRN, and Lamp2b, inserted into the pCDNA3.1-EGFP vector. EV-sorting protein Rab7a, VTA1, Syntenin-1, and IST1 are also included. EGFP levels of the overexpressed sorting proteins in the purified EVs were analyzed by western blot analysis and quantified. Briefly, EVs were collected from transfected cells for western blot analysis (3.0×10^9 particles; Figure S3) and quantification.

- 6. *The study does not assess whether EN144 works across different cell types or is limited to Expi293F cells. A broader range of cell lines should be tested to evaluate its generalizability.*

Author response: We fully appreciate your suggestion. In the revised manuscript, we added other cell lines in addition to Expi293F, including HeLa, HcerEpic, and synovial-fluid mesenchymal stem (SF-MSC) cells to validate the functional consistency of EN144-mediated engineering outcomes. Although transfection efficiency varied across cells, normalization to EV protein expression versus cellular protein expression showed no significant difference, showing its generality as a universal scaffold platform.

We have revised the manuscript as follows: “In addition to the Expi293F cell line, other adherent cell lines are frequently employed as sources of EVs. Based on the ratio of EN144 in EVs to all the EN144 in the cell lysates (the total expression levels varied in different cells), we found that a similar percentage of EN144 was enriched in EVs when we overexpressed EN144 in cell lines, including 293T cells, SF-MSCs, HeLa cells, and HcerEpic cells. These results demonstrated that EN144 exhibited comparable sorting efficiency in various cell types (Supplementary Figure 7A, B).” Please refer to lines 460-466.

Two new figures have been added in the Supporting Information file.

Figure S7

Figure S7. EN144 loading efficiency in various cells and their EVs. (A) Expression of FLAG-tagged proteins in cell lysates (top) and EVs (bottom) from 293T, SF-MSCs, HeLa, or HcerEpic cells transfected with EN144 plasmid. Top panel: Equal amounts of cell lysates were loaded as follows: lane 1, cell lysates from untransfected 293T cells (control); lanes 2-5, cell lysates from 293T, SF-MSCs, HeLa, or HcerEpic cells transfected with EN144 plasmid. Bottom panel: Equal amounts of EVs (3.0×10^9 particles EVs) were loaded as follows: lane 1, EV lysates from untransfected 293T cells (control); lanes 2-5, EV lysates from 293T, SF-MSCs, HeLa, or HcerEpic cells transfected with EN144 plasmid. $n = 3$ in each group. (B) FLAG-tagged protein content ratio of EVs-to-Cells in EN144-transfected 293T, SF-MSCs, HeLa, or HcerEpic cells based on the gray value of western blot band in (A). Consistent FLAG protein expression across all tested cell lines and their EVs confirmed the efficient loading capability of EN144 in diverse cellular contexts. One-way ANOVA accompanied by Tukey's post-hoc test was employed. Data are analyzed of three independent experiments and shown as mean \pm SD. ns $P > 0.05$.

- 7. The toxicity study is poorly controlled. There is no benchmarking against unengineered Expi293 EVs, making it impossible to determine whether the observed bell-shaped viability effect is specific to EN144-engineered EVs.

Author response: We sincerely appreciate your critical feedback. To address this concern, we have performed additional experiments comparing EN144-EVs and unengineered Expi293F-EVs, with five biological replicates per group. The revised results are summarized as follows: **Bell-shaped dose-response:** Contrary to our initial observation, the updated data (Supplementary Figure 8C-8D) revealed that neither EN144-EVs nor Expi293-EVs exhibited a pronounced bell-shaped effect on cell viability across the tested concentration range (ranging from 5×10^8 to 5×10^{10} EV particles/mL). **Proliferation-promoting effects:** Notably, both EV types demonstrated similar trends in enhancing cell proliferation after 48 hours of co-culture, suggesting that the observed proliferative activity may be a general property of EVs derived from Expi293 cells rather than specific to EN144 engineering.

Figure S8

Figure S8C-S8D. Analysis of doses-response of EN144-EV doses (5.0×10^8 , 1.0×10^9 , 5.0×10^9 , 1.0×10^{10} , and 5.0×10^{10} particles/mL) on viability at 24 h (C) and viability at 48 h (D) in rat primary chondrocytes, RAW 264.7

cells, 293T cells, or SF-MSC cells. n = 3 each group.

Additional Figure 2. Analysis of doses-response of Expi293F-EV doses (5.0×10^8 , 1.0×10^9 , 5.0×10^9 , 1.0×10^{10} , and 5.0×10^{10} particles/mL) on viability at 24 h (A) and viability at 48 h (B) in rat primary chondrocytes, RAW 264.7 cells, 293T cells, or SF-MSC cells. n = 3 each group.

- 8. The biodistribution study is severely limited by small sample size. Signal quantification should also be presented graphically.

Author response: All animal experiments were performed with at least three biological replicates per group (n = 3). The complete datasets are archived in the Source Data files (SourceData_2G-2H, SourceData_S15G). Additionally, we have now included a quantitative presentation of signal intensity in the revised figures (Figures 2G-2H, Supplementary Figures 15G).

Revised figures:

Figure 2G-2H

Figure 2G. Representative fluorescent images of tissues showing the distribution of intravenously injected DiR-labeled Expi293F-EVs and EN144-EVs at different doses (1.0×10^9 , 1.0×10^{10} , or 1.0×10^{11} particles) in C57BL/6 mice post-injection after 1 h and 24 h. (H) Quantification of EVs based on the fluorescence signal of DiR. The data shown are the average \pm standard deviation of 3 replicates.

Figure S15G

Figure S15G. Biodistribution of EN144-EV^{mgp130} in septic mice. Left panel: Representative *in vivo* bioluminescence imaging and *ex vivo* tissue imaging (heart, liver, spleen, lung, kidney, and brain) 6 hours after intravenous administration of EN144-EV^{mgp130} (1.0×10^{10} particles). Right panel: Quantification of EVs in tissues based on the

fluorescence signal of DiR. n = 3 each group.

Source Data:

Additional Figure 3. Source data of Figure 2G and 2H.

Additional Figure 4. Source data of Figure S15G.

- 9. The increase in RBC count following high-dose EV injection is unexplored. The authors should *discuss* whether this is an artifact or a physiologically relevant observation.

Author response: In Figure S9C, while high-dose Expi293F-EV administration transiently increased RBC counts, we observed no significant changes in either Red Cell Distribution Width (RDW, a marker of erythrocyte morphology) or hemoglobin concentration (HGB, reflecting oxygen-carrying capacity). Although this RBC elevation (mean= $9.98 \times 10^{12}/L$, 95% CI: $8.98-10.61 \times 10^{12}/L$) observed in the high-dose Expi293F-EV group is statistically significant, as compared to the PBS group, the values still fall within the normal range (Charles River data: mean= $9.48 \times 10^{12}/L$, 95% CI: $7.14-12.2 \times 10^{12}/L$) (see the attached table of C57BL/6 mouse haematology below). Based on this, we conclude that such RBC increase is not physiologically relevant.

Figure S9C

Figure S9C. EV injections did not cause toxicity to mice based on peripheral blood parameters (white blood cells, red blood cells, hemoglobin, granulocytes, lymphocytes, and monocytes). Blood samples were analyzed 24 hours after intravenous injection of DiR-labeled Expi293F-EVs or EN144-EVs at doses of 1.0×10^{11} particles/mouse via the tail vein ($n = 3$ mice/group).

C57BL/6 Mouse Hematology

North American Colonies*
January 2008 - December 2012

C57BL/6NCr		WBC (K/ μ L)	NEUT (K/ μ L)	LYMPH (K/ μ L)	MONO (K/ μ L)	EOS (K/ μ L)	BASO (K/ μ L)	NEUT (%)	LYMPH (%)	MONO (%)	EOS (%)
Male (σ)	Mean	8.90	1.44	6.87	0.41	0.14	0.03	15.89	77.41	4.61	1.55
	95% interval										
	Low	4.45	0.53	3.24	0.15	0.01	0.00	7.36	61.26	2.18	0.13
	High	13.96	3.09	11.15	0.94	0.42	0.13	28.59	87.18	11.02	4.42
	N	123	123	123	123	123	123	123	121	121	121
Female (φ)	Mean	8.44	1.19	6.71	0.36	0.15	0.03	14.00	79.55	4.37	1.69
	95% interval										
	Low	3.90	0.42	2.88	0.17	0.01	0.00	7.44	70.19	2.19	0.20
	High	13.94	2.55	10.92	0.69	0.50	0.14	22.67	87.82	7.06	4.51
	N	125	125	125	125	125	125	125	123	123	123

C57BL/6NCr		BASO (%)	RBC (M/ μ L)	HGB (g/dL)	HCT (%)	MCV (fL)	MCH (pg)	MCHC (g/dL)	RDW (%)	PLT (K/ μ L)	MPV (fL)
Male (σ)	Mean	0.38	9.48	14.2	46.6	49.2	14.8	30.2	17.9	1347	5.0
	95% interval										
	Low	0.01	7.14	10.8	37.3	42.7	11.7	24.6	15.9	841	4.3
	High	1.24	12.20	19.2	62.0	56.0	16.3	34.9	20.3	2159	6.1
	N	121	121	121	121	121	121	121	121	115	115
Female (φ)	Mean	0.34	9.24	13.8	45.4	49.2	15.0	30.7	17.9	1167	4.9
	95% interval										
	Low	0.02	7.37	10.9	37.2	42.6	13.0	26.0	16.1	565	4.3
	High	1.26	11.50	18.1	58.0	55.6	16.8	35.9	21.1	1849	5.6
	N	123	123	123	123	123	123	123	123	117	117

The data is from C57BL/6 Mouse Hematology published by Charles River.

- 10. Figure 4B-C does not support the claim that EN144 and Emp11 EVs have significantly higher binding affinity than PTGFRN and Lamp2b. The text should be revised to accurately reflect the results.

Author response: We have redesigned the experiments in this section to demonstrate the functionality of targeting EVs and revised the original description, “EN144-EV^{CAP} and ENPP1-EV^{CAP} showed a pronounced affinity with rat primary chondrocytes (Figure 3B-C)” to “To enable direct comparison among different EVs, EV concentrations were normalized based on particle counts determined by RPS. Primary rat chondrocytes were incubated with EVs for 15 min or 2 h to evaluate cellular uptake efficiency. Quantitative analysis revealed enhanced cellular uptake of PTGFRN-EV^{CAP} (80.38 ± 0.79), ENPP1-EV^{CAP} (107.6 ± 19.78), and EN144-EV^{CAP} (101.6 ± 15.32) in the 2-hour incubation group, with ENPP1-EV^{CAP} and EN144-EV^{CAP} exhibiting the highest fluorescence intensities (Figure 3B). Notably, after 24-hour incubation, the cellular uptake rates of ENPP1-EV^{CAP} and EN144-EV^{CAP} reached approximately 90%, significantly surpassing those of PTGFRN-EV^{CAP} ($87.97\% \pm 0.38\%$), Lamp2b-EV^{CAP} ($69.37\% \pm 0.97\%$), and Expi293F-EV ($47.93\% \pm 1.27\%$, Figure 3C). To determine

whether the observed ENPP1-EV^{CAP}/chondrocyte interaction was mediated by CAP-specific binding, primary chondrocytes were pre-treated with CAP peptide (1, 3, 10 jig) for 2 h to block potential binding sites. Interestingly, pretreatment with 10 jig CAP peptide significantly reduced the fluorescence intensity of primary chondrocytes by 45.6% compared to untreated controls. This dose-dependent blockade indicates that CAP peptide mediates the uptake of EN144-EV^{CAP} by primary chondrocytes (Supplementary Figure 12).” Please refer to lines 574-588.

Figure 3B. Confocal fluorescent microscope images and quantification of the fluorescence intensity showing the uptake of DiI-labeled EVs (Expi293F-EV, Lamp2b-EV^{CAP}, PTGFRN-EV^{CAP}, ENPP1-EV^{CAP}, and EN144-EV^{EGFP}) by chondrocytes after 15 min and 2 h of co-culture. Scale bars: 100 µm; 50 µm (Zoomed images).

Figure S12. Blocking effect of CAP peptide on the interaction between ENPP1-EV^{CAP} and chondrocytes. Primary chondrocytes were pre-incubated with various concentrations of CAP peptide (1 jig, 3 jig, and 10 jig) for 1 hour to block cell surface receptors, following co-incubation with DiI-labeled ENPP1-EV^{CAP} for 2 hours. Left panel: Fluorescence images illustrate the internalization of ENPP1-EV^{CAP} by chondrocytes across treatment groups, demonstrating the concentration-dependent inhibitory effect of CAP peptide. Scale bar = 100 µm. Right panel: Quantitative analysis of mean fluorescence intensity in left panel (n = 3 each group).

- 11. Figures 4J and 4N contain *overstated* claims about delivery efficiency. Editing efficiency must be quantified by *sequencing to establish actual percentage editing rates*. The authors should also clarify whether they used VSVG to enhance endosomal escape, as EVs inherently are poor at performing endosomal escape.

Author response: We sincerely appreciate your insightful feedback regarding Figures 3J and 3N (Original Figure 4J and 4N). Please allow us to clarify our methodology and interpretations.

Regarding Figure 3J: The data reflect intracellular EV uptake efficiency, quantified by detection of exogenous sgRNA content in cells via qRT-PCR (not editing outcomes). We have clarified this distinction in the revised manuscript. “Next, we evaluated the capacity of engineered EVs to deliver sgRNA into cells. The GFP-Nanobody system exhibited the highest uptake efficiency, with intracellular levels of exogenous sgRNA

exceeding those delivered by CIBN-CRY2, PRB-PDAR, and NbALFA^{PE}-ALFA systems by approximately 4.32-, 5.57-, and 18.71-fold, respectively (Figure 3J).” Please refer to lines 621-625.

Regarding Figure 3N: The detectable red fluorescence provides qualitative evidence of functional genome editing. We fully agree that sequencing-based quantification is essential to establish editing efficiency, and this will be addressed in future work.

On the use of VSVG: We did not utilize VSVG to enhance endosomal escape. The observed signals suggest that the endogenous EV escape mechanism, while inherently limited, enabled detectable delivery of functional CRISPR-Cas9 components under our experimental conditions. We recognize that engineering strategies (e.g., VSVG incorporation) could significantly improve efficiency and thank you for this suggestion. Future studies will implement VSVG modification to systematically optimize EV delivery.

Figure 3J&3N

Figure 3J. sgRNA expression levels in 293 T cells transfected by different CRISPR/Cas9-harboring EVs after 24 h. **(N)** Observation of red fluorescence in 293T stop-dsRed cells following transient transfection with EN144-EGFP, Cas9-sgRNA1-Nanobody, Cas9-sgRNA2-Nanobody plasmids, or treatment with EVs derived from these plasmids.

- 12. The orientation of displayed receptors is not addressed. ENPP1 is a type II transmembrane protein, whereas TNFR and IL6ST are type I transmembrane proteins. The effect of this difference on receptor functionality is completely ignored.

Author response: Different types of transmembrane proteins will have different orientations. We cloned the TNFR and IL6ST at the extravesicular domain of ENPP1 to display these two receptors on the surface of EVs. Although the structures of ENPP1-TNFR and ENPP1-IL6ST fusion proteins were not characterized due to their complexity, we have shown that the EVs carrying these fusion proteins were able to bind to IL-6 and reduce the level of serum IL-6 and sIL-6R both *in vitro* and *in vivo*.

In Vitro Binding Assays (Supplementary Figure 17A): The effective binding of EN144-EV^{mgp130} to the IL-6/IL-6R complex demonstrates that the ligand-binding capability of the mgp130 remains intact.

In Vivo Efficacy (Figures 4N & 4Q): The significant reduction in serum IL-6 and sIL-6R levels in sepsis models suggests that EN144-EV^{mgp130} does not compromise the receptor's ability to neutralize circulating IL-6/IL-6R complexes.

Figure S17. EN144-EV^{mgp130} Targets the IL-6 Trans-Signaling Pathway. (A) Binding capacity analysis of EN144-EV^{mgp130} or EN144-EVs to IL-6/IL-6R proteins. EN144-EV^{mgp130} or EN144-EVs (5.0×10^{10} particles) were incubated with IL-6 (10 μ g), IL-6R (10 μ g), or IL-6/IL-6R complex (10 μ g each) under continuous rotation at 4°C for 4 h, followed by dialysis for 16 h to remove unbound proteins. Expression of IL-6 and IL-6R proteins bound to EN144-EV^{mgp130} or EN144-EVs was detected by WB.

Figure 4N&4Q. Serum levels of inflammatory cytokines and acute-phase proteins in septic mice treated with EN144-EV^{mgp130}, EN144-EV^{hgp130}, low-dose sgp130 (1 μ g), or high-dose sgp130 (10 μ g), including (N) IL-6, (O) TNF- α , (P) IL-1 β , (Q) soluble IL-6 receptor (sIL-6R), (R) serum amyloid A (SAA), and (S) C-reactive protein (CRP). $n = 3$ mice/group.

- 13. Figure 5 does not convincingly demonstrate a decrease in IL-6 and TNF- α levels. TNF- α levels remain unchanged or even increase following treatment with TNF decoy EVs, contradicting the authors' claims.

Author response: According to your suggestion, we have replaced the original statement: “Notably, both EN144-EV^{mgp130} and EN144-EV^{hTGNFR1} treatments markedly reduced the levels of secreted TNF- α and IL-6 proteins from inflammatory RAW 264.7 cells (Figure 5C-E).” with “Quantitative analysis revealed that EN144-EV^{mgp130} achieved significantly greater suppression of TNF- α and IL-6 in inflammatory RAW 264.7 cells (Figure 4C, D).” Please refer to lines 694-696.

- 14. No dose-response study has been conducted for the experiments in Figure 5, this is important to perform in order to determine the true efficacy of the treatment.

Author response: Thank you for your insightful feedback. We systematically performed dose-response analyses for both EN144-EV^{mgp130} and EN144-EV^{hgp130} in a LPS-induced murine sepsis model. Four escalating doses (1.0×10^9 , 5.0×10^9 , 1.0×10^{10} , and 5.0×10^{10} particles/mouse) were administered intravenously, with survival time recorded. At 1.0×10^{10} particles/mouse, survival rates reached 100% (EN144-EV^{mgp130}) and 80% (EN144-EV^{hgp130}), while the highest dose (5.0×10^{10} particles/mouse) achieved 100% survival for both treatments (Figure 4L-M). Furthermore, *in vitro* studies demonstrated a dose-dependent reduction in IL-6 levels: EN144-EV^{hgp130} decreased IL-6 by 30% at 5.0×10^8 particles/mL and by 55% at 1.0×10^{10} particles/mL (Supplementary Figure 16B), corroborating the dose-response relationship observed *in vivo*.

We have revised the main text as follows: “Furthermore, we observed that EN144-EV^{hgp130} exhibited a dose-dependent inhibitory effect on IL-6. Specifically, EN144-EV^{hgp130} decreased IL-6 levels by 30% at a concentration of 5.0×10^8 particles/mL and by 55% at 1.0×10^{10} particles/mL (Supplementary Figure 16B). Building on these encouraging *in vitro* findings, the therapeutic potential of decoy EVs was subsequently evaluated *in vivo* using a sepsis model. Administration of varying EV doses demonstrated that intravenous injection of EN144-EV^{mgp130} or EN144-EV^{hgp130} (1.0×10^{10} particles/mouse) significantly improved survival outcomes in septic mice, achieving survival rates exceeding 80% within 120 hours post-administration (Figure 4J, K). Notably, when administered at the highest dose of 5.0×10^{10} particles/mouse, both preparations achieved complete (100% up to 120 h after injection) survival protection (Figure 4J, K).” Please refer to lines 715-724.

Figure 4J-4K. Survival rates of septic mice treated with varying doses of EN144-EV^{mcp130} (J) and EN144-EV^{hgp130} (K). n = 5 mice/group.

Figure S16B. The mRNA levels of IL-6 in RAW 264.7 treated with 1000 ng/mL LPS following co-culture with various concentrations (1.0×10^8 , 5.0×10^8 , 1.0×10^9 , or 1.0×10^{10} particles/mL) of EN144-EV^{hgp130} were measured by qRT-PCR.

- 15. The biodistribution study in Supplementary Figure 11H is based on a single animal and cannot be used to support conclusions.

Author response: As this is the same as Question 8, please refer to answers to Question 8.

- 16. Figure 5M does not clarify the differences in effects between mouse GP130 EVs and human GP130 EVs, despite both performing similarly in suppressing IL-6 trans-signaling.

Author response: Thank you for raising this important point. As highlighted in the revised Figure 4L

(previously Figure 5M), our survival data demonstrate that both EN144-EV^{mgp130} and EN144-EV^{hgp130} exhibit comparable efficacy in improving survival rates within the LPS-induced sepsis model, with no statistically significant differences observed between the two treatments ($P > 0.05$). The specific revisions are as follows: “Comparative analysis revealed that EN144-EV^{hgp130} (80%) and EN144-EV^{mgp130} (80%) demonstrated significantly enhanced therapeutic efficacy in septic mice, outperforming sgp130 treatment at 10 μ g (40%) and 1 μ g (20%) doses (Figure 4L).” Please refer to lines 724-726.

Figure 4L. Survival rates changes of septic mice after tail vein injection of EN144-EV^{mgp130}, EN144-EV^{hgp130}, sgp130 (1 μ g), or sgp130 (10 μ g). n = 5 mice/group. Statistical analysis was performed using the Log-rank (Mantel-Cox) test. * $P < 0.05$, ** $P < 0.01$.

- 17. Supplementary Figure 9 does not explain how translation activity was measured. The method must be described in detail to prevent misinterpretation of the results.

Author response: We have revised the original statement “Further, protein profiling confirmed the translational activity of EGFP mRNA delivered by EN144-EV^{EGFP} in co-cultured cells (Supplementary Figure 9F)” to “Further, the results of protein profiling indicated that EGFP protein expression was detected in cells co-cultured with EN144-EV^{EGFP} (Supplementary Figure 13F).” Please refer to lines 604-606.

The corresponding legend has also been revised, as follow: “(F) Cells were incubated with EN144-EV^{EGFP} (1.0×10^{10} particles), and intracellular EGFP protein levels were measured using LC-MS/MS (n = 2 each group).” Please refer to line 170-172 in Supplementary document.

Figure S13F. Cells were incubated with EN144-EV^{EGFP} (1.0×10^{10} particles), and intracellular EGFP protein levels were measured using LC-MS/MS (n = 2 each group).

Reviewer #3 (Remarks to the Author):

The manuscript, Identification of a 144-amino-acid Scaffold Protein for Extracellular Vesicle Engineering” by Wenjing Yan and colleagues presents a novel strategy to generate engineered extracellular vesicles (EVs). Using an unbiased mass-spectrometry-based proteomic approach, they identify fifteen candidate proteins that could potentially be used to engineer EVs. They then focus on ENPP1 and show that a truncated version of the protein consisting only of 114 amino-acid residues is sufficient to load EVs with peptides, proteins and even mRNAs. Afterwards, they use this novel technique to engineer decoy EVs that present the extracellular portion of the signal-transducing receptor gp130 on their surface in order to inhibit the IL-6 trans-signaling pathway. They evaluate their novel EVs in two animal models (LPS-induced sepsis in mice and a rat model of osteoarthritis) and show that their EVs are able to reduce inflammation in these models, suggesting that gp130-decorated EVs can potentially be used as novel anti-inflammatory therapeutics.

The manuscript contains a high amount of data and consists basically of two different parts: First, the development of a novel method to generate EVs based on the truncated ENPP1 and second, the use of this novel method to generate an inhibitor that blocks IL-6 trans-signaling. While the first part is well done in my opinion, the second part is not and lacks important experiments and appropriate controls. Furthermore, the authors use tocilizumab in their rodent models, an antibody that blocks the human IL-6 receptor, but is known to have no activity in mice and rats.

Author response: We sincerely thank you for your positive comments and constructive criticism. We noticed the report that tocilizumab does not block interleukin-6 (IL-6) signaling in murine cells (PLoS One. 2020 May 4;15(5):e0232612.). So, we have removed tocilizumab from the manuscript and re-performed most of the animal studies.

Major points:

*1. The authors claim that decoration of their EVs with gp130 makes this an inhibitor of IL-6 trans-signaling. This would mean that IL-6 in complex with the soluble IL-6R would bind to the gp130 on the EVs, resulting in inactivation. IL-6 alone would not bind to the EVs. However, none of these things is shown in the manuscript; only indirect evidence (reduction of IL-6 in organs and tissues) and reduced inflammation based on staining of organ sections is shown. **Please provide evidence that** i) EVs with gp130 bind IL-6/sIL-6R complexes, but not IL-6 or sIL-6R alone, ii) measure serum levels of IL-6 and sIL-6R in vivo and demonstrate that sufficient EVs are present to neutralize IL-6 trans-signaling in vivo, iii) provide evidence (by measuring acute phase proteins) that classic signaling is still intact in vivo and only IL-6 trans-signaling is blocked.*

Author response: Thank you for the thorough evaluation and helpful guidance in enhancing the mechanistic validation of our study. We have systematically addressed the concerns raised with the following new experimental evidence:

(1) In Vitro Confirmation of Specific Binding of gp130EVs to the IL-6/sIL-6R Complex

We conducted *in vitro* binding assays by incubating EN144-EV^{mgp130}/Control EVs (EN144-EV) with recombinant IL-6, sIL-6R, or pre-formed IL-6/sIL-6R complexes (Supplementary Figure 17A). Compared to incubation with IL-6 or IL-6R alone, EN144-EV^{mgp130} showed higher binding to the IL-6/sIL-6R complex. Unlike soluble gp130, both EN144-EV and EN144-EV^{mgp130} groups exhibited non-specific binding to free IL-6/sIL-6R, which may be due to the inherent adsorptive properties of the EV membrane.

(2) In Vivo Neutralization Capacity and Pharmacodynamic Validation

We treated septic mice via tail vein injection with EN144-EV^{mgp130}, resulting in a significant reduction in serum IL-6/sIL-6R levels ($P < 0.001$ compared to the PBS group, Figure 4N&4Q). Concurrently, downstream pro-inflammatory cytokines (IL-1 β , TNF- α , Figure 4O-P) were also reduced. If EN144-EV^{mgp130} merely inhibited cytokine production (rather than binding), we would not observe a simultaneous decrease in both IL-6 and sIL-6R. Thus, these experiments suggest that EN144-EV^{mgp130} can specifically bind to the IL-6/sIL-6R complex.

(3) Inhibition of Trans-Signaling Pathway While Preserving Intact Classical IL-6 Signaling

In mice treated with EN144-EV^{mgp130}, the expression of acute-phase proteins (CRP, SAA, Figure 4R-S) did not significantly change compared to the PBS group ($P > 0.05$), indicating the integrity of the classical IL-6 signaling pathway. Additionally, the ratio of pStat3/Stat3 in the liver, spleen, and lungs of treated mice was reduced, indicating the blockade of the trans-signaling pathway (Supplementary Figure 17B-G).

In summary, our experimental results indicate that EN144-EV^{mgp130} can specifically bind to the IL-6/sIL-6R complex, blocking the IL-6 trans-signaling pathway rather than the IL-6 cis-signaling pathway. We hope these modifications meet your expectations and thank you again for your meticulous review.

The manuscript has been revised as follows: “To elucidate the regulatory role of EN144-EV^{mgp130} in the IL-6 trans-signaling pathway, we first characterized its binding properties through *in vitro* experiments. In contrast to the canonical paradigm²², which claims that gp130 activation requires the IL-6/IL-6R complex, EN144-EV^{mgp130} exhibited a unique ligand-binding profile: it bound to IL-6 or IL-6R individually, with the highest binding efficiency observed for the IL-6/IL-6R complex (Supplementary Figure 17A). In control experiments, EN144-EVs bound to either IL-6 or IL-6R, but did not enhance the binding capacity of the IL-6/IL-6R complex (Supplementary Figure 17A). This contrast further suggested that the enhanced ligand interaction of EN144-EV^{mgp130} may involve both gp130-dependent recognition and membrane-associated nonspecific interactions. Building on these findings, we evaluated the biological effects of EN144-EV^{mgp130} in a murine model. Treatment with EN144-EV^{mgp130} significantly reduced serum levels of IL-6, sIL-6R, and proinflammatory cytokines (IL-1 β and TNF- α ; Figure 4N-Q), while decreasing the pSTAT3/STAT3 ratio in hepatic, splenic, and pulmonary tissues, collectively indicating effective suppression of IL-6 trans-signaling activation (Supplementary Figure 17B-G). Notably, expression levels of classic signaling markers—acute-phase proteins SAA and CRP—remained unaltered, excluding nonspecific interference of EN144-EV^{mgp130} with the canonical pathway (Figure 4R, S). These results demonstrate that EN144-EV^{mgp130} selectively inhibits the IL-6 trans-signaling pathway through targeted engagement of the IL-6/IL-6R complex, without appreciably affecting classical IL-6 signaling.” Please refer to lines 738-755.

Figure S17

Figure S17. EN144-EV^{mgp130} targets the IL-6 trans-signaling pathway. (A) Binding capacity analysis of EN144-EV^{mgp130} or EN144-EVs to IL-6/IL-6R proteins. EN144-EV^{mgp130} or EN144-EVs (5.0×10^{10} particles) were incubated with IL-6 (10 μg), IL-6R (10 μg), or IL-6/IL-6R complex (10 μg each) under continuous rotation at 4°C for 4 h, followed by dialysis for 16 h to remove unbound proteins. Expression of IL-6 and IL-6R proteins bound to EN144-EV^{mgp130} or EN144-EVs was detected by WB. (B-D) Effects of therapeutic interventions targeting the Stat3 signaling pathway in septic mice. Experimental design: Septic mice received tail vein injections of EN144-EV^{mgp130} (1.0×10^{10} particles/mouse, 100 μL), EN144-EV^{hgp130} (1.0×10^{10} particles/mouse, 100 μL), low-dose sgp130 (1 μg/mouse, 100 μL), or high-dose sgp130 (10 μg/mouse, 100 μL). Control groups included septic mice (LPS group) and healthy mice (PBS group), both injected with an equal volume of PBS (100 μL/mouse). Lung (B), liver (C), and spleen tissues (D) were harvested post-treatment after 6 h, and expression of phosphorylated Stat3 (p-Stat3) and total Stat3 protein were analyzed by WB. (E-G) Quantitative analysis of pStat3 and Stat3 protein level in lungs (E), liver (F), and spleen (G) based on results in B-D (n = 3 each group).

Figure 4N-4S. Serum levels of inflammatory cytokines and acute-phase proteins in septic mice treated with EN144-EV_{mgp130} (1.0×10^{10} particles/mouse), EN144-EV_{hgp130} (1.0×10^{10} particles/mouse), low-dose sgp130 (1 μg), or high-dose sgp130 (10 μg), including (N) IL-6, (O) TNF-α, (P) IL-1β, (Q) soluble IL-6 receptor (sIL-6R), (R) serum amyloid A (SAA), and (S) C-reactive protein (CRP). n = 3 mice/group.

2. It is known from the literature that monomeric gp130 is a rather weak inhibitor of IL-6 trans-signaling. Dimerization of gp130 increases the affinity towards IL-6/sIL-6R significantly. Do the EVs express monomeric or dimeric gp130 on their surface?

Author response: Compared with monomeric gp130, displaying gp130 on the surface of EVs clusters gp130 in finite space, which increases the avidity of gp130, similar to dimerization of gp130. We reason that this explains why EV-displayed gp130 showed higher therapeutic efficacy than monomeric gp130. In this regard, we concur with you on this effect.

3. The authors use tocilizumab in their rodent models. Tocilizumab does not block the IL-6R in the mouse and does not block the IL-6R in the rat, it only works in humans. I provide several papers in the specific comments that prove this. For rodent studies, other anti-IL-6R antibodies are available that work in these species. Thus, the *in vivo* experiments as done by the authors are worthless, unfortunately.

Author response: We sincerely appreciate your comments on the use of tocilizumab in the manuscript, which will help improve the quality of our work. Although some literature suggests that tocilizumab can exert anti-inflammatory effects in mice, there are indeed concerns about its use in mouse and rat models. We have carefully considered your feedback and made adjustments to our experimental design. Therefore, we have replaced tocilizumab with sgp130 (mouse) protein as the positive control in our sepsis treatment experiments. We have conducted new *in vivo* experiments to confirm the anti-inflammatory effects of EN144-EV^{mgp130} again, ensuring that our results are more relevant and accurate. In the rat osteoarthritis model, we have removed the tocilizumab treatment group, as you suggested, and retained the original positive control, which is hyaluronic acid treatment. Due to space limitations, we only present selected representative figures. Detailed descriptions can be found in lines 717-755 of the manuscript. We hope these changes address your concerns while maintaining the integrity and relevance of our study.

Figure 4J-4S

Figure 4J-4S. Survival rates of septic mice treated with varying doses of EN144-EV^{mgp130} (J) and EN144-EV^{hgp130} (K). $n = 5$ mice/group. Survival rates (L) and body weight (M) changes of septic mice after tail vein injection of EN144-EV^{mgp130} (1.0×10^{10} particles/mouse), EN144-EV^{hgp130} (1.0×10^{10} particles/mouse), sgp130 (1 μ g), or sgp130 (10 μ g). $n = 5$ mice/group. Statistical analysis was performed using the Log-rank (Mantel-Cox) test. $*P < 0.05$, $**P < 0.01$. (N-S) Serum levels of inflammatory cytokines and acute-phase proteins in septic mice treated with EN144-EV^{mgp130} (1.0×10^{10} particles/mouse), EN144-EV^{hgp130} (1.0×10^{10} particles/mouse), low-dose sgp130 (1 μ g), or high-dose sgp130 (10 μ g), including (N) IL-6, (O) TNF- α , (P) IL-1 β , (Q) soluble IL-6 receptor (sIL-6R), (R) serum amyloid A (SAA), and (S) C-reactive protein (CRP). $n = 3$ mice/group.

Figure S16

Figure S16. Loading capacity assessment and anti-inflammatory evaluation of EN144-EV^{hgp130}. (A) Quantitative analysis of FLAG protein in EN144-EV^{hgp130}. Left panel: A His-MYO standard curve was established via WB by comparing immunoblot signals between serially diluted His-MYO standards and FLAG protein in EN144-EV^{hgp130} samples. Right panel: The relative quantity of FLAG protein in EN144-EV^{hgp130} was calculated based on band intensity measured by ImageJ software using the standard curve from the left panel. (B) The mRNA levels of IL-6 in RAW 264.7 treated with 1000 ng/mL LPS following co-culture with various concentrations (1.0×10^8 , 5.0×10^8 , 1.0×10^9 , or 1.0×10^{10} particles/mL) of EN144-EV^{hgp130} were measured by qRT-PCR. (C-F) EV preparations (EN144-EV^{hgp130} and EN144-EV^{hgp130-CAP}) of 1.0×10^{10} EV particles or sgp130 (1 µg or 10 µg) were injected into septic mice. The mice were sacrificed post-injection after 6 hours, and the organs were dissected for analysis. (C) Immunohistochemical (IHC) staining of IL-6 protein in lung, liver, and spleen tissues (brown: positive signals; blue: nuclear counterstaining with hematoxylin). Scale bar = 100 µm. (D) Therapeutic evaluation of intervention groups

on inflammatory injury in organs of septic mice. Representative H&E stained images showing histopathological alterations in lung, liver, and spleen tissues of intervention and control groups (scale bar = 100 μ m). (E) Semi-quantitative analysis of IHC-positive areas using ImageJ software based on IHC results in (C). (F) Semi-quantitative inflammatory scores (0-3: none/mild/moderate/severe injury) based on H&E staining results in (D).

Figure 5I-5O

Figure 5I. ACLT-induced OA rat model. In situ injections (weekly, 4 weeks) of indicated treatments. Cartilage recovery and toxicity assessments in the 5th week using staining, immunofluorescence, immunohistochemistry, and biochemical analyses (n = 3). (I) Schematic illustration showing cartilage repair by EV treatment *in vivo*. Briefly, a rat model of OA was established based on the ACLT method. EV preparations (EN144-EVs, EN144-EV^{hgp130}, and EN144-EV^{hgp130-CAP}) of 1.0×10^{10} EV particles or HA (10 μ g) were injected intraarticularly into OA rats weekly for four weeks. The rats were sacrificed in the fifth week, and the cartilages were dissected for analysis. (J) OARSI scores indicating cartilage damage severity in different treatment groups (n = 3). (K) Representative images of HE, TB, and SO-FG staining for histological analysis. The sham group serves as control. Scale bars: 200 μ m (top panel), 50 μ m (bottom panel). (L-O) Immunohistochemical expression levels of COL2A1, IL-6, and MMP13 in cartilage tissues from various treatment groups (n = 3). Scale bar: 50 μ m.

Figure S20

Figure S20. Modulation of synovial gene expression by EN144-EV^{hgp130-CAP}. A rat model of OA was established based on the ACLT method. EV preparations (EN144-EVs, EN144-EV^{hgp130}, and EN144-EV^{hgp130-CAP}) of 1.0×10^{10} EV particles or HA (10 μ g) were injected intra-articularly into OA rats weekly for four weeks. Sham-operated controls underwent joint cavity exposure without ligament transection. The rats were sacrificed in the fifth week, and the synovial tissues were dissected for following analysis: (A) Immunofluorescence staining analysis of iNOS (red; M1 macrophage marker) and CD206 (green; M2 macrophage marker) expression in synovial tissues (scale bar = 100 μ m); (B) Immunohistochemical staining of IL-6 expression in synovial tissues (scale bar = 50 μ m); (C-E) Semi-quantitative analysis of protein level of CD206 (C), iNOS (D), and IL-6 (E) in synovial tissues across groups (n = 3 each group).

General comments:

- Quality of the western blots is insufficient and has to be improved.
- Quality of the figures has to be improved.

Author response: We sincerely appreciate the opportunity to improve our manuscript based on your valuable feedback. In response to the concerns regarding Western blots (WB) and figure quality, we have implemented the following substantial revisions. (1) **Western Blot Optimization:** We optimized experimental protocols by increasing sample loading quantities and adjusting transfer/blocking durations. Clearly labeled molecular weight marker positions. Provided full-length blot membranes for verification. (2) **Comprehensive Figure Upgrades:** We streamlined the layout of main figures for enhanced clarity. Upgraded fluorescence image resolution through optimized acquisition settings. Revised data visualization methods to improve interpretability. Due to space limitations, only a portion of the revised WB and images are displayed. Other revised images can be viewed in the **Manuscript, Supplementary document, and SourceData file.**

Figure 2D. Western blot analysis showing the expression of exosome markers Hsp70, TSG101, and syntenin-1 in EN144-EVs and Expi293F-EVs, but not calnexin and histone H3. Each sample contains 10 jig of cell lysate or 3.0×10^9 EV particles.

Figure 3K. Immunoprecipitation (IP) analysis of cells or EVs (EN144-EV^{M-CRISPR/Cas9}) treated with GFP-Nanobody plasmids. Cell lysates or EN144-EV^{M-CRISPR/Cas9} were immunoprecipitated with a Cas9 antibody and blotted using Cas9 antibody. IP with an anti-IgG antibody served as the control (top). sgRNA analysis of immunoprecipitants following the IP assay in lysates of cells or EN144-EV^{M-CRISPR/Cas9} (bottom).

Original Figure 5K

Revised Figure 4H

SourceData_4H

Figure 4H. Protein levels of IL-6 in LPS-induced RAW 264.7 cells following co-culture with EN144-EV^{hgp130} or EN144-EV^{mgp130}.

Original Figure 1

Revised Figure 1

Figure 1. Screening for EV-sorting proteins and identification of EN144 as the smallest EV-specific scaffold protein. (A) Schematic illustration showing the workflow. Briefly, three methods (ultracentrifugation, magnetic bead-based EVtrap technology, and a purification process integrating tangential flow industrial extraction) were used to purify Expi293F-EVs from cell lysates, which were subsequently subjected to LC-MS/MS analysis. (B) Schematic illustration of the sorting proteins and their expression levels quantified by western blot analysis. Class I sorting proteins include STX4, ENPP1, TFRC, TFRC-81 (transmembrane sequence of TFRC), VAMP2, and CD55, inserted into the pEGFP-C1 vector. Class II sorting proteins include EPCAM, AT1A1, Syntenin-1, CXADR, PDL1, STX7, IST1, SNAP23, VTA1, PTGFRN, and Lamp2b, inserted into the pCDNA3.1-EGFP vector. EV-sorting protein Rab7a, VTA1, Syntenin-1, and IST1 are also included. EGFP levels of the overexpressed sorting proteins in the purified EVs were analyzed by western blot analysis and quantified. Briefly, EVs were collected from transfected cells for western blot analysis (3.0×10^9 particles; Figure S3) and quantification. (C) Quantification of EGFP protein levels in each group of EVs based on mass spectrometry. (D) Quantification of EGFP molecules per EV particle based on the grayscale values and formula in Figure S4. (E) Flow cytometric analysis showing the percentages of EGFP-positive

EVs among the isolated EV populations. (F) Schematic illustration showing truncated ENPP1 variants. Abbreviations: CD, cytoplasmic domain; TM, transmembrane domain; SMB, somatomedin B-like domain. (G) Quantification of ENPP1 variant numbers per EV particle based on western blot analysis of the FLAG tag (Figure S6B). (H) Schematic illustration showing the comparison of the architecture of the selected scaffold proteins.

Figure 2. Proteomic analysis and *in vivo* distribution of EN144-EVs. (A) Western blot analysis showing that EN144-EVs enrich the overexpressed EN144 based on the FLAG tag. (B) Representative TEM images of Expi293F-EVs and EN144-EVs. Scale bar: 100 nm. (C) Particle size distribution of Expi293F-EVs and EN144-EVs based on Resistive Pulse Sensing (RPS) analysis. (D) Western blot analysis showing the expression of exosome markers Hsp70, TSG101, and syntenin-1 in EN144-EVs and Expi293F-EVs, but not calnexin and histone H3. Each sample contains 10 jig of cell lysate or 3.0×10^9 EV particles. (E) The volcano plot showing 188 differentially expressed proteins (DEPs) in EN144-EVs compared to Expi293F-EVs. (Selection threshold: $|\log_2(\text{FC})| > 2$, $\text{adj.}P < 0.05$, with 143 upregulated DEPs in red dots and 45 downregulated DEPs in blue). (F) Gene Ontology (GO) and Kyoto Encyclopedia of Genes and Genomes (KEGG) pathway analysis ($\text{adj.}P < 0.05$) of the 188 DEPs. CC, Cellular Component; MF, Molecular Function. (G) Representative fluorescent images of tissues showing the distribution of intravenously injected DiR-labeled Expi293F-EVs and EN144-EVs at different doses (1.0×10^9 , 1.0×10^{10} , or 1.0×10^{11} particles) in C57BL/6 mice post-injection after 1 h and 24 h. (H) Quantification of EVs based on the fluorescence signal of DiR. The data shown are the average \pm standard deviation of 3 replicates.

Figure S9. Safety of EN144-EVs based on hematological and biochemical indices through intravenous administration to mice. (A-C) EV injections did not cause toxicity to mice based on peripheral blood parameters (white blood cells, red blood cells, hemoglobin, granulocytes, lymphocytes, and monocytes). Blood samples were analyzed 24 hours after intravenous injection of DiR-labeled Expi293F-EVs or EN144-EVs at doses of 1.0×10^9 , 1.0×10^{10} , and 1.0×10^{11} particles/mouse via the tail vein ($n = 3$ mice/group). (D-F) Safety assessment based on hepatic function markers (ALT, ALP, AST), renal function markers (BUN, CREA), and lactate dehydrogenase (LDH) 24 hours post-injection ($n = 3$ mice/group).

Figure 3. Surface and luminal cargo loading on EN144-EVs. (A) Schematic diagram showing the construction of chondrocyte-targeting EVs with the CAP sequence fused to the N-termini of Lamp2b or PTGFRN, or the C-termini

of ENPP1 or EN144. (B) Confocal fluorescent microscope images and quantification of the fluorescence intensity showing the uptake of DiI-labeled EVs (Expi293F-EV, Lamp2b-EV^{CAP}, PTGFRN-EV^{CAP}, ENPP1-EV^{CAP}, and EN144-EV^{EGFP}) by chondrocytes after 15 min and 2 h of co-culture. Scale bars: 100 μ m; 50 μ m (Zoomed images).

Further specific comments:

• 1. Line 104: Etanercept is not a monoclonal antibody, but rather a designer protein consisting of the extracellular part of TNFR2 and the constant part of a IgG antibody

Author response: We have removed the mention of Etanercept and revised the sentence to: “For example, monoclonal antibodies targeting inflammatory cytokines, such as Tocilizumab, have been approved as first-line therapies for treating rheumatoid arthritis (RA).” Please refer to lines 103-105.

• 2. Line 109: The canonical term is “classic signaling”, not “cis-signaling”

Author response: We have made the corresponding corrections. Please refer to lines 109, 114, and 117.

• 3. Lines 119/120: #22 is not the correct citation for this statement. The correct citation is Jostock et al. Eur J Biochem. 2001 Jan;268(1):160-7

Author response: We have made corrections to the references.

• 4. Figure 1G: “domin” should probably be “domain” and “cataytic” should be “catalytic” **Author response:** We have made the corresponding corrections.

Figure 1F. Schematic illustration showing truncated ENPP1 variants. Abbreviations: CD, cytoplasmic domain; TM, transmembrane domain; SMB, somatomedin B-like domain.

• 5. For some Western blots molecular weight markers are indicated, while this is missing for others. Please add them to all western blots

Author response: We have added all Western blot molecular weight markers. Due to space limitations, we have only presented a portion of the revised results here. The complete set of revised results is stored in the manuscript and supplementary files.

Figure 2A. Proteomic analysis and *in vivo* distribution of EN144-EVs. (A) Western blot analysis showing that EN144-EVs enrich the overexpressed EN144 based on the FLAG tag. (D) Western blot analysis showing the expression of exosome markers Hsp70, TSG101, and syntenin-1 in EN144-EVs and Expi293F-EVs, but not calnexin and histone H3. Each sample contains 10 μ g of cell lysate or 3.0×10^9 EV particles.

Figure 3K-3L

Figure 3. Surface and luminal cargo loading on EN144-EVs. (K) Immunoprecipitation (IP) analysis of cells or EVs (EN144-EV^{M-CRISPR/Cas9}) treated with GFP-Nanobody plasmids. Cell lysates or EN144-EV^{M-CRISPR/Cas9} were immunoprecipitated with a Cas9 antibody and blotted using Cas9 antibody. IP with an anti-IgG antibody served as the control (top). sgRNA analysis of immunoprecipitants following the IP assay in lysates of cells or EN144-EV^{M-CRISPR/Cas9} (bottom). (L) Lysates were subjected to IP using control IgG or anti-Cas9 antibody, followed by Western blot analysis with anti-FLAG antibody to assess FLAG-EN144 protein levels.

• 6. Original, uncropped blots are currently missing and should be added so that the reviewers can assess the quality of the western blots presented in this manuscript

Author response: We have added uncropped blots in **SourceData file**. Due to space limitations, we have only presented a portion of the revised results here. The complete set of revised results is stored in the **supplementary files**.

Additional Figure 7. Source data of Figure S7A.

SourceData_S14B

Additional Figure 8. Source data of Figure S14B.

SourceData_S18A

Additional Figure 8. Source data of Figure S18A.

•7. The legends of the Supplemental Figures is way too brief. Please expand them and describe in more detail what is shown in the figures.

Author response: We have thoroughly revised and expanded the descriptions to provide more detailed explanations of what is shown in each figure. The revisions are extensive and cover all the supplementary figures, as you suggested. For your convenience, the detailed legends are included in the **supplementary files**. Revised

parts are marked in red.

•8. Figure S3: I presume these are western blots – against what? GFP?

Author response: We have labeled the target protein as “EGFP” in both the figure legend and the image itself within the revised **supplementary file**.

The specific revisions are as follows: “**Figure S3**. Expression of EGFP proteins in engineered EVs. A panel of proteins was selected as candidates for new EV-sorting proteins. Class I sorting proteins include STX4, ENPP1, TFRC, TFRC-81 (transmembrane sequence of TFRC), VAMP2, and CD55, inserted into the pEGFP-C1 vector. Class II sorting proteins include EPCAM, AT1A1, Syntenin-1, CXADR, PDL1, STX7, IST1, SNAP23, VTA1, PTGFRN, and Lamp2b, inserted into the pCDNA3.1-EGFP vector. EV-sorting protein Rab7a is also included. Briefly, EVs were collected from transfected cells, and EGFP levels of the overexpressed sorting proteins in the purified EVs were analyzed by western blot analysis (3.0×10^9 particles).” Please refer to lines 47-53 in **supplementary file**.

Figure S3. Expression of EGFP proteins in engineered EVs. A panel of proteins was selected for new EV-sorting proteins. Class I sorting proteins include STX4, ENPP1, TFRC, TFRC-81 (transmembrane sequence of TFRC), VAMP2, and CD55, inserted into the pEGFP-C1 vector. Class II sorting proteins include EPCAM, AT1A1, Syntenin-1, CXADR, PDL1, STX7, IST1, SNAP23, VTA1, PTGFRN, and Lamp2b, inserted into the pCDNA3.1-EGFP vector. EV-sorting protein Rab7a is also included. Briefly, EVs were collected from transfected cells, and EGFP levels of the overexpressed sorting proteins in the purified EVs were analyzed by western blot analysis (3.0×10^9 particles).

•9. Figure S4: The legend states “Quantitative analysis of EGFP in engineered EVs”. The blot on the left hand site shows “His-MYO standard curve” (what is the connection to EGFP?) and the blot on the right hand site says “Flag”. Again, what is the connection to EGFP? The main text suggests that these blot help in the quantification of EGFP-tagged proteins in EVs, but it is absolutely unclear for me how this is done and how this relates to an anti-FLAG blot and His-tagged MYO.

Author response: We apologize for any confusion caused by the brevity of our original description. The His-MYO standard curve (left panel) was established using serially diluted His-tagged MYO proteins to correlate Western blot (WB) band intensity with protein quantity. This calibration allows quantification of EGFP-tagged proteins (right panel) by comparing their EGFP-labeled WB signals (EGFP was co-expressed with FLAG-tag) against the standardized curve. We have now explicitly detailed this methodological linkage in the revised figure legend to better demonstrate how anti-EGFP blot data enables EGFP quantification through established reference metrics.

Figure S4

Figure S4. Quantitative analysis of the expression of EGFP protein in engineered EVs. Left panel: Quantitative Western blot analysis of serially diluted His-MYO recombinant proteins (detected with anti-His antibody). A standard curve was generated by plotting the gray values of His-MYO recombinant proteins band against their corresponding protein mass. Right panel: Molecular loading quantification of EGFP in engineered EVs (detected with anti-GFP antibody). Based on standard curve established in the left panel, EV particle count (3.0×10^9 particles/group), and target band gray values, the EGFP molecules per EV were calculated. EGFP molecules/EV = $11 \text{gray value-derived protein mass (g)} / 11 \text{EV particles} \times \text{molecular weight (Da)} \times N_a$, where N_a is Avogadro's constant ($6.022 \times 10^{23} \text{ mol}^{-1}$).

The specific revisions are as follows: **Figure S4.** Quantitative analysis of the expression of EGFP protein in engineered EVs. Left panel: Quantitative Western blot analysis of serially diluted His-MYO recombinant proteins (detected with anti-His antibody). A standard curve was generated by plotting the gray values of His-MYO recombinant proteins band against their corresponding protein mass. Right panel: Molecular loading quantification of EGFP in engineered EVs (detected with anti-GFP antibody). Based on standard curve established in the left panel, EV particle count (3.0×10^9 particles/group), and target band gray values, the EGFP molecules per EV were calculated. EGFP molecules/EV = $11 \text{gray value-derived protein mass (g)} / 11 \text{EV particles} \times \text{molecular weight (Da)} \times N_a$, where N_a is Avogadro's constant ($6.022 \times 10^{23} \text{ mol}^{-1}$). Please refer to lines 5865 in supplementary file.

• 10. Lines 385-387: The text mentions quantification of EGFP, the corresponding Figure 1e “flag molecules per EV”. Please make this consistent and explain better what has been done.

Author response: Since FLAG and EGFP are co-expressed, we have now corrected this inconsistency and updated the y-axis label in Figure 1D to “EGFP molecules per EV” to align with the text description. Additionally, we have revised the figure legend to “Quantification of EGFP molecules per EV particle based on the grayscale values and formula in Figure S4.” Please refer to lines 485-486.

D **Figure 1D**

Figure 1D. Quantification of EGFP molecule numbers per EV particle based on the grayscale values and formula in Figure S4.

• 11. Line 393/394: “Although Rab7a-EVs exhibited the highest EGFP loading, they were internalized less

efficiently than ENPP1-EVs and PTGFRN-EVs (Supplementary Figure 5A).” It is unclear to me what is shown in the western blot in Supp. Fig. 5a and how one can see that a) Rab7a-EVs exhibited the highest EGFP loading and b) that they are less efficiently internalized than ENPP1 -EVs and PTGFRN-EVs

Author response: We have made the necessary revisions to address the concerns raised. Regarding the statement that Rab7a-EVs exhibited the highest EGFP loading, this observation was based on the data presented in Figure 1D. To clarify the internalization efficiency of different EVs, we conducted co-culture experiments with Rab7a-EVs, ENPP1-EVs, and PTGFRN-EVs and measured the expression of FLAG proteins in Expi293F cells using Western Blot. This experiment was designed to assess the cellular uptake of EVs.

To avoid any confusion for the readers, we have revised the legend of Supplementary Figure 5A to “**Expression of FLAG-tagged proteins in Expi293F cells co-cultured with Rab7a-EVs, ENPP1-EVs, or PTGFRN-EVs by western blot analysis. Each sample contains 1.0×10^{10} particles EVs.**” Please refer to lines 70-72 in the supplementary file. Additionally, we have updated the corresponding text in the manuscript to: “**Although Rab7a-EVs exhibited the highest EGFP loading (Figure 1D), they were internalized less efficiently than ENPP1-EVs and PTGFRN-EVs (Supplementary Figure 5A).**” Please refer to lines 449-450.

- 12. *Supplementary Figure 5C: The x-axis is not correct.*

Author response: We have revised the x-axis label in Supplementary Figure 5C to accurately reflect the data presented.

Figure S5C. Time-course analysis of FLAG-tagged proteins in Expi293F cells exposed to ENPP1-EVs (B) and corresponding quantitative assessment (C).

- 13. *Lines 399/400: “The truncations and full-length ENPP1 plasmids exhibited comparable levels of EGFP expression in transfected cells (Figure 1H).” This statement is not backed up from the data shown in Fig. 1H, as the statistical test has only been done between untransfected cells and the different expressed proteins. Whether there are differences between the individuals transfected constructs has not been tested.*

Author response: To address your concern, we have made the following revisions: (1) The original statement has been removed; (2) The text now states: “**Transfection efficiencies of truncated and full-length ENPP1 plasmids ranged from 38% to 58% (Supplementary Figure 6A).**” Please refer to lines 456-457.

- 14. *Figure 2A: The GAPDH blot looks strange. Why would GAPDH be present as two different bands in EN144 expressing cells, but only as a single band in untransfected cells?*

Author response: In Figure 2A, the concurrent use of mouse-derived primary antibodies against both FLAG and GAPDH on the same membrane could indeed lead to detection interference, particularly in EV samples

from EN144-transfected cells. To resolve this issue, we have repeated the experiment with methodological refinement: sequential immunoblotting (FLAG detection -* antibody stripping -* GAPDH detection). We have also replaced all Figures with such issues, including **Figure 2A**, **Figure 4A**, **Figure 4F**, and **Supplementary Figure 18A**.

Figure 2A

Figure 2A. Western blot analysis showing that EN144-EVs enrich the overexpressed EN144 based on the Flag tag. (B) Representative TEM images of Expi293F-EVs and EN144-EVs. Scale bar: 100 nm.

Figure 4A

Figure 4A. Construction of EN144-mgp130 plasmid and the expression of FLAG-tagged EN144 proteins in cell lysates (10 jig) and EVs (Expi293F-EVs, EN144-EV^{m_{gp130}}, 3.0×10^9 EV particles) of Expi293F cells transfected by EN144-mgp130 plasmids.

Figure 4F

Figure 4F. Expression of FLAG-tagged proteins in cell lysates (top) and EVs (bottom).

Figure S18A. The expression of FLAG-tagged protein in cell lysates of Expi293F cells transfected by EN144, EN144-hgp130, or EN144-hgp130-CAP plasmids.

•15. *Figure S7: scale bar is not defined*

Author response: We have added Scale bars: 100 μm to the revised Supplementary Figure 10A-10B. Please refer to lines 134 in the supplementary file.

•16. *Lines 554/555: “Here, we also non-covalent loading of Cas9 protein through selected docking domains, ...” Please correct and rephrase.*

Author response: This sentence has been revised to: “We systematically evaluated multiple protein heterodimerization strategies for EV cargo loading, including a GFP-GFP nanobody interacting pair³⁷, a light-dependent CIBN-CRY2 interacting pair³⁸, an NbALFA^{PE}-ALFA interacting pair³⁹, and a PRB-PDAR interacting pair⁴⁰.” Please refer to lines 612-615.

•17. *Lines 604-606: “Genetic fusion of mgp130 or hTNFR1 to the C-terminus of EN144 created constructs, which were transiently expressed in Expi293F cells.” Please explain why murine gp130, but human TNFR1 was used. Further, please state which part of mgp130 and hTNFR1 were fused to EN144 (I assume the whole ectodomains without transmembrane and intracellular parts, but this should be stated here).*

Author response: We have carefully considered your concerns and have revised the manuscript accordingly.

(1) **Choice of mgp130 and hTNFR1:** Previous studies (PMID: 34616047, Supplementary Table 2) have demonstrated that EVs loaded with mgp130 or hTNFR1 can exert anti-inflammatory effects. Therefore, in this study, we directly loaded these proteins onto the newly identified scaffold protein EN144 to compare their anti-inflammatory effects. To avoid potential confusion, we have omitted the hTNFR1 data as they yielded negative results. This omission does not compromise the overall structure of the manuscript.

(2) **Fusion Details:** “Based on previous studies^{41, 42}, the C-terminus of EN144 was fused to mouse gp130 (mgp130), comprising the extracellular, transmembrane, and partial cytoplasmic domains, and transiently expressed in Expi293F cells.” Please refer to lines 686-688. This configuration precisely replicates the successful structural framework reported in the references we cited, ensuring the functionality of mgp130.

•18. *Figure S11B is not explained in the text and should be removed*

Author response: We have removed Figure S11B.

•19. *Lines 612-614: “Notably, both EN144-EV^{mgp130} and EN144-EV^{hTNFR1} treatments markedly reduced the levels of secreted TNF- α and IL-6 proteins from inflammatory RAW 264.7 cells (Figure 5C-E).” There is no difference between LPS treated and EN144-EV^{hTNFR1}-treated cells in Fig. 5C, and even an increase in the EN144-EV^{hTNFR1}-treated cells in Fig. 5D. The statement is wrong – only EN144-EV^{mgp130} reduces TNF and IL-6 gene expression. Whether any of this treatments has an influence on the protein level is unclear: The quality*

of the western blots presented in Fig. 5E is too bad to answer this question, and appear to only represent a single experiment. An answer would require better western blots and the quantification of at least three independent experiments.

Author response: We repeated the Western blotting with increased loading protein and performed quantitative analysis in three experiments. The problematic sentence has been revised to: “Quantitative analysis revealed that EN144-EV^{mgp130} achieved significantly greater suppression of TNF- α and IL-6 in inflammatory RAW 264.7 cells (Figure 4C, D).” Please refer to lines 694-695.

Figure 4D. Protein levels of TNF- α and IL-6 in RAW 264.7 cells treated with EN144-EV^{mcp130} post-LPS induction. Briefly, LPS (1 jg/mL) was used to induce inflammation in RAW 264.7 cells, which were incubated with EN144-EV^{mcp130} for 24h before the mRNA levels of TNF- α and IL-6 were detected by qRT-PCR (C) and protein levels by western blot analysis

• 20. Lines 621/622: “These data prove the anti-inflammatory efficacy of EN144-EV^{mcp130}.” No, they do not. Reduced IL-6 expression level do not prove that something has an anti-inflammatory effect.

Author response: We have revised it to “*In vitro* and *in vivo* investigations demonstrated that EN144-EV^{mcp130} significantly attenuated IL-6 and TNF- α expression levels in a LPS-induced inflammation model.” Please refer to lines 701-703.

• 21. Lines 637-642: The authors inject tocilizumab into mice and compare this with their EVs. This is highly problematic, because tocilizumab does not work in mice. Tocilizumab binds to the human IL-6R, but not to the mouse IL-6R. See Okazaki et al. *Immunol Lett.* 2002;84(3):231-40 / Garbers et al. *J Biol Chem.* 2011;286(50):42959-70 / Nishimoto et al. *Ann Rheum Dis.* 2000;59 Suppl 1:i21-7 / Ueda et al. *Sci Rep.* 2013;3:1196 and Lokau et al. *PLoS One.* 2020 May 4;15(5):e0232612

Author response: We sincerely appreciate your valuable comments and suggestions. In response to your concern regarding the use of tocilizumab in our study, we have made the following revisions: We have replaced tocilizumab with different doses of sgp130 (1 jg, 10 jg) as the positive control. The results are consistent with previous studies, showing that both EN144-EV^{mcp130} and EN144-EV^{hgp130} exhibit excellent anti-inflammatory effects.

The specific revisions are as follows: “Comparative analysis revealed that EN144-EV^{hgp130} (80%) and EN144-EV^{mcp130} (80%) demonstrated significantly enhanced therapeutic efficacy in septic mice, outperforming sgp130 treatment at 10 jg (40%) and 1 jg (20%) doses (Figure 4L). Moreover, mice receiving decoy EVs therapy exhibited accelerated weight recovery (72 hours) compared to the sgp130-treated group (96 hours posttreatment, Figure 4M). The decoy EVs exhibited superior protection against LPS -induced inflammation, as evidenced by reduced IL-6 levels in the liver, lung, and spleen (Supplementary Figure 16C, E). We then assessed the impact of various treatments on the histopathology of sepsis mice. Mice in the PBS group exhibited massive

inflammatory cell infiltration in the liver, lungs, and spleen, and elevated histologic scores. Conversely, decoy EVs treatment mitigated inflammation in the liver, lungs and spleen, effectively alleviating LPS-induced inflammation (Supplementary Figure 16D, F).” Please refer to lines 724-734.

Figure 4L-4M

Figure 4L-4M. Survival rates (L) and body weight (M) changes of septic mice after tail vein injection of EN144-EV^{mgp130}, EN144-EV^{hgp130}, sgp130 (1 μg), or sgp130 (10 μg). n = 5 mice/group. Statistical analysis was performed using the Log-rank (Mantel-Cox) test. **P* < 0.05, ***P* < 0.01.

Figure S16C-16F

Figure S16. Loading capacity assessment and anti-inflammatory evaluation of EN144-EV^{hgp130}. (C-F) EV preparations (EN144-EV^{hgp130} and EN144-EV^{hgp130-CAP}) of 1.0×10^{10} EV particles or sgp130 (1 μg or 10 μg) were injected into septic mice. The mice were sacrificed post-injection after 6 hours, and the organs were dissected for

analysis. (C) Immunohistochemical (IHC) staining of IL-6 protein in lung, liver, and spleen tissues (brown: positive signals; blue: nuclear counterstaining with hematoxylin). Scale bar = 100 μm . (D) Therapeutic evaluation of intervention groups on inflammatory injury in organs of septic mice. Representative H&E stained images showing histopathological alterations in lung, liver, and spleen tissues of intervention and control groups (scale bar = 100 μm). (E) Semi-quantitative analysis of IHC-positive areas using ImageJ software based on IHC results in (C). (F) Semi-quantitative inflammatory scores (0-3: none/mild/moderate/severe injury) based on H&E staining results in (D).

•22. Line 705: *The authors evaluate their engineered EVs in a rat model. Again, they use tocilizumab as a control treatment, and again, this is highly problematic, because tocilizumab does also not work in rats. Please see Okazaki et al. Immunol Lett. 2002;84(3):231-40 / George et al. JACC Basic Transl Sci. 2021 Apr 7;6(5):431-443*

Author response: Thank you for your insightful and constructive feedback. We have meticulously revised the manuscript and all tocilizumab-related data have been systematically and thoroughly removed from the OA therapeutic evaluation. To ensure the integrity and relevance of our analysis, we have now utilized hyaluronic acid (HA), a clinically validated conservative treatment for OA, as the sole positive control. This modification not only addresses your concern but also maintains the original experimental architecture, ensuring the consistency and validity of our study design.

This email has been sent through the Springer Nature Tracking System NY-610A-NPG&MTS

REVIEWER COMMENTS

Reviewer #2 (Remarks to the Author):

Overall, the manuscript has been improved several items still need to be properly addressed and, hence, further revisions are needed:

1. Protein quantification (Point 1) – While additional LFQ details are provided, the method for calculating proteins per EV remains unclear. Proper normalization to particle numbers is still missing, limiting reproducibility and accurate interpretation.

Author response: We sincerely thank you for your comments. Regarding the method for calculating proteins per EV particle, it involves two aspects: 1) the quantification of the protein, and 2) the quantification of the EV particle numbers. Both are explained as follows.

The number of protein molecules was calculated from the grayscale values of the protein bands, using a standard curve with MYO protein as a reference. The detailed experimental procedure and calculation method are also described in the manuscript and the caption of Figure S4. A standard curve was generated by plotting the gray values of His-MYO recombinant protein bands against their corresponding protein mass. The EGFP western blot signal in the EV lysate was then quantified by comparing the band intensity to the standard curve.

Figure S4. Quantitative analysis of the expression of EGFP protein in engineered EVs. Left panel: Quantitative Western blot analysis of serially diluted His-MYO recombinant proteins (detected with anti-His antibody). A standard curve was generated by plotting the gray values of His-MYO recombinant proteins band against their corresponding protein mass. Right panel: Molecular loading quantification of EGFP in engineered EVs (detected with anti-GFP antibody). Based on the standard curve established in the left panel, EV particle count (3.0×10^9 particles/group), and target band gray values, the EGFP molecules per EV were calculated. EGFP molecules/EV = [gray value-derived protein mass (g)] / [EV particles \times molecular weight (Da)] \times N_a , where N_a is Avogadro's constant (6.022×10^{23} mol⁻¹).

Second, for the quantification of the number of EVs, the resistance-pulse sensing (RPS) method was used. We rewrote this method to increase its clarity, as follows: "Resistive Pulse Sensing (RPS) was employed for the determination of particle size and concentration. Using a NanoCoulter counter (Resun Technology, China), a chip with a measurement range of 60 nm to 200 nm was installed, and 200 μ L of PBS buffer was rapidly dispensed into each sample well on both sides of the detection card, ensuring the system was free of air bubbles or leaks. Before sample analysis, a background verification was performed, confirming that the particle count in PBS was fewer than 5 over 60 seconds. In each sample, 100 EV particles were selected for size analysis." Please refer to lines 808-814. Revised parts are marked in red.

Combining these two methods together, the EGFP molecules per EV were calculated according to the following equation: $\text{EGFP molecules/EV} = [\text{gray value-derived protein mass (g)}] / [\text{EV particles} \times \text{molecular weight (Da)}] \times N_a$, where N_a is Avogadro's constant ($6.022 \times 10^{23} \text{ mol}^{-1}$). The results derived from this calculation are presented in Figure 1D.

Figure 1D. Quantification of EGFP molecules per EV particle based on the grayscale values and formula in Figure S4.

To address any potential ambiguity in the original wording, the sentence has been rephrased for clarity and precision. The revised text is as follows: “After normalization, we estimated an average of 523 EGFP proteins on each ENPP1-EV particle and 629 EGFP proteins on each Rab7a-EV particle, exceeding the EGFP number on the PTGFRN-EV particle by 1.32 and 1.58 folds, respectively (Figure 1D, Supplementary Figure 4).” Please refer to lines 178-181.

2. Stable cell line characterization (Point 2) – The claim that gene copy number is irrelevant lacks supporting data. Gene copy number should be quantified (e.g., by qPCR or ddPCR) and expression normalized accordingly to control for variability in genomic integration and its impact on EV cargo loading.

Author response: We sincerely thank you for your comments. Admittedly, due to the lack of access to a ddPCR system (which is considered the gold standard for absolute copy number quantification), we were unable to directly measure gene copy number. Instead, we performed a series of experiments at the mRNA and protein levels to prove the success of gene manipulation.

First, we reason that mRNA levels can serve as a surrogate indicator of gene integration, as the functional output of a gene (i.e., protein expression) depends not only on DNA copy number but also on transcriptional activity. According to Lattenmayer, C. and coworkers: “When comparing genetic parameters to productivity, a good correlation of mRNA levels with specific productivity was observed, whereas high gene copy numbers were not always accompanied by high protein expressions (PMID:17324483).” Therefore, we consider the mRNA level a reliable and functionally relevant indicator of the successfully integrated and effectively transcribed gene load.

Therefore, we selected nine stable cell lines (CXADR, ENPP1, EPCAM, Lamp2b, PDL1, PTGFRN, Rab7a, SNAP23, and STX7) that exhibited high EV-associated EGFP loading in previous screening and quantified their EGFP mRNA levels using qPCR.

As shown in Additional Table 1, EGFP mRNA expression in all cell lines was significantly higher than in the

negative control. Even in the ENPP1 cell line, which showed the lowest expression among the nine, the mRNA level was approximately 8000-fold higher than that of the control. Using ENPP1 as a reference, we normalized the EGFP mRNA expression of the other cell lines (Additional Figure 1). The results indicated that the expression levels in the other cell lines were generally tens of times higher than that of ENPP1, demonstrating considerable transcriptional heterogeneity among the cell lines.

Additional Table 1: Relative expression of EGFP in cells

Expi293F	CXADR	ENPP1	EPCAM	Lamp2b	PDL1	PTGFRN	Rab7a	SNAP23	STX7
1.0	102837.0	8599.3	160253.9	180295.0	109456.6	126607.2	201441.3	174153.5	130166.6
1.1	72716.7	8192.0	136638.1	181549.1	149522.2	109456.6	167059.2	141456.6	160253.9
1.0	64633.8	7434.4	118128.7	185363.8	163621.2	174153.5	198668.0	140479.5	128374.6

Additional Figure 1

Additional Figure 1: Relative EGFP expression in stable cell lines, normalized to the ENPP1 stable cell line.

Furthermore, to eliminate the influence of basal transcriptional variation on the assessment of EV loading capacity, we calculated the ratio of “EGFP protein in EVs to EGFP mRNA in cells” for each cell line (Additional Figure 2). This ratio essentially reflects the amount of functional EGFP protein loaded into EVs per mRNA transcript—i.e., the loading efficiency. We found that, even after normalizing for mRNA expression levels, the loading efficiency of ENPP1 and Rab7a remained significantly higher than that of other constructs, including PTGFRN.

These two facts combined support our conclusion that ENPP1 is an outstanding scaffold protein.

Additional Figure 2

Additional Figure 2: EGFP protein in EVs to EGFP mRNA in cells, normalized to the ENPP1 stable cell line.

3. *sgRNA delivery and editing efficiency (Point 11) – Detection of sgRNA by qRT-PCR is not evidence of functional cytoplasmic delivery, as sgRNA could remain trapped in endosomes or bound to the cell surface. Functional validation through targeted deep sequencing to quantify indel frequencies is required to support claims of genome editing efficiency.*

Author response: We sincerely thank the reviewer for this valuable comment. The point raised about qRT-PCR's inability to distinguish between functional delivery of sgRNA and non-functional scenarios, such as endosomal entrapment or cell-surface binding, is indeed crucial. Accordingly, we have performed functional validation of genome-editing efficiency, as suggested, in cells treated with Cas9-sgRNA-loaded EVs.

Specifically, after co-culturing EVs carrying Cas9-sgRNA with the stop-dsRed reporter cell line for 48 hours, the red fluorescent cells were sorted, and genomic DNA was extracted. The target site was then amplified by PCR and analyzed using Sanger sequencing. The resulting sequencing data were further examined using the TIDE online platform, which accurately quantifies the editing efficacy and simultaneously identifies the predominant types of insertions and deletions (indels). The TIDE analysis confirmed gene editing in the treated group, with a detected editing efficacy of 3.1% to 13.6%.

Accordingly, we have added the following content to the manuscript:

Methods section: “Sanger sequencing

Sanger sequencing was employed to assess the editing efficiency at the target genomic loci. PCR amplification was performed in a 20 μ L reaction mixture containing 3 μ L of template DNA, 1 μ L each of forward and reverse primers (10 μ M), 10 μ L of 2 \times Hieff[®] PCR Master Mix, and 5 μ L of ddH₂O. The amplification protocol consisted of an initial denaturation at 95°C for 5 min, followed by 40 cycles of denaturation at 95°C for 25 s, annealing at 62°C for 30 s, and extension at 72°C for 20 s, with a final hold at 4°C. The resulting DNA fragments were separated by electrophoresis on a 2% agarose gel. Subsequently, the purified PCR products were subjected to Sanger sequencing, and the editing efficacy was analyzed using the TIDE algorithm⁷⁰.” Please refer to lines 899-907.

Results section: “Sanger sequencing followed by TIDE (Tracking of Indels by Decomposition) analysis further revealed that the editing efficacy in the stop-dsRed gene upon EVs treatment ranged from 3.1% to 13.6% (Supplementary Table 5).” Please refer to lines 388-390 in the Manuscript document and lines 285-286 in the Supplementary document.

4. gp130 binding – The claim that gp130 binds IL-6 or sIL-6R alone contradicts established literature, where binding occurs only to the IL-6/sIL-6R complex during trans-signaling. Please clarify

Author response: We sincerely thank you for your insightful comments on the gp130 binding mechanism. We fully agree with you that in the classical trans-signaling model, gp130 primarily binds to the IL-6/sIL-6R complex rather than to IL-6 or sIL-6R alone.

In our initial experiments, the observed apparent binding of gp130 to IL-6 or sIL-6R alone was likely attributable to the "protein corona" effect on the surface of EVs. Unlike conventional soluble proteins, EVs possess a high surface-area-to-volume ratio and high surface energy, which facilitate the adsorption of surrounding biomolecules and lead to the formation of a protein corona (see PMID: 40848102). This may have resulted in nonspecific binding signals.

To minimize potential interference from the protein corona, we optimized our experimental procedures, including extending the dialysis time, increasing the frequency of dialysis buffer changes, and reducing the concentrations of IL-6 and sIL-6R used. Under these optimized conditions, gp130 on EN144-EVs exhibited specificity consistent with the classical literature: it bound specifically to the IL-6/sIL-6R complex, but not to IL-6 or sIL-6R alone.

We revised the main text as follows: “To investigate the regulatory role of EN144-EV^{mgp130} in the IL-6 trans-signaling pathway, we first characterized its binding properties *in vitro*. Consistent with the canonical theory that gp130 activation requires the preformed IL-6/IL-6R complex²², EN144-EV^{mgp130} did not bind to IL-6 or IL-6R individually, and binding was observed exclusively with the IL-6/IL-6R complex (Supplementary Figure 17A). In control experiments, EN144-EVs showed no binding to IL-6, IL-6R, or the IL-6/IL-6R complex (Supplementary Figure 17A). These results demonstrate that EN144-EV^{mgp130} specifically interacts with the IL-6/IL-6R complex.” Please refer to lines 485-491.

Figure S17A

Figure S17A. (A) Binding capacity analysis of EN144-EV^{mgp130} or EN144-EVs to IL-6/IL-6R proteins. EN144-EV^{mgp130} or EN144-EVs (5.0×10^{10} particles) were incubated with IL-6 (50 ng), IL-6R (100 ng), or IL-6/IL-6R complex (50 ng/100 ng) under continuous rotation at 4°C for 4 h. Unbound proteins were removed by dialysis against PBS for 16 h, with buffer changes every 2 h. Expression of IL-6 and IL-6R proteins bound to

EN144-EV^{mgp130} or EN144-EVs was detected by WB.

5. Lack of TNF binding assay – No quantitative binding or affinity data are provided to confirm the interaction of TNF decoy EVs with TNF. Binding affinity should be validated with ELISA-based binding assays or biophysical interaction studies.

Author response: We thank you for raising the need for binding data. Although the EN144-EV^{TNFR1} data have been removed as the study now focuses on EN144-EV^{mgp130}, we have addressed your question by performing an ELISA-based binding assay as suggested.

The binding of EN144-EV^{TNFR1} with TNF- α was evaluated using the ELISA method. Briefly, anti-TNFR1 antibodies were coated on the ELISA plate wells, and 4.00×10^9 counts of EN144-EV^{TNFR1} particles were added to each well. Subsequently, five concentrations of TNF- α (0.16 ng/mL, 0.8 ng/mL, 4 ng/mL, 20 ng/mL, 100 ng/mL) were added to each well and incubated to allow TNF- α to bind to TNFR1 on immobilized EVs. Subsequent detection was conducted following the operational steps of a commercial TNF- α ELISA kit. This procedure involved the sequential addition of HRP-conjugated detection antibodies and a chromogenic substrate. The absorbance of each well was then measured at a wavelength of 450 nm using a microplate reader. Data analysis revealed that the amount of TNF- α that bound to a fixed number of EN144-EV^{TNFR1} particles increased in a dose-dependent manner with increasing concentrations. By fitting a dose-response curve, the half maximal effective concentration (EC₅₀) was calculated to be approximately 8.3 ng/mL. This result indicates that TNFR1 expressed on the surface of EN144-EV^{TNFR1} can effectively bind TNF- α .

Additional Figure 3. ELISA curve showing that TNF- α binds to immobilized EN144-EV^{TNFR1} in a dose-dependent manner, with a simulated EC₅₀ of 8.3 ng/mL.

Besides ELISA, other evidence is also available. For example, co-IP experiment also proves the binding between TNF- α and EN144-EV^{TNFR1}. Briefly, a total of 5.0×10^{10} particles of EN144-EV^{TNFR1} or control EN144-EVs were co-incubated with TNF- α under continuous rotation at 4 °C for 4 hours, followed by 16 hours of dialysis to remove unbound TNF- α . The level of TNF- α bound to the EVs was detected by co-immunoprecipitation (co-IP). The results showed that the level of TNF- α bound to EN144-EV^{TNFR1} was significantly higher than that bound to the control EN144-EVs, indicating that EN144-EV^{TNFR1} can function as an effective decoy by specifically binding and capturing TNF- α .

Additional Figure 3

Additional Figure 4: Analysis of the binding capacity between EN144-EV^{TNFR1} or EN144-EV and TNF- α . EN144-EV^{TNFR1} or EN144-EV was incubated with TNF- α under continuous rotation at 4 °C for 4 h, followed by dialysis for 16 h to remove unbound proteins. Flag-tagged magnetic beads were used to enrich EVs, and the expression of TNF- α bound to EN144-EV^{TNFR1} or EN144-EV was analyzed by Western blot.

Original data for Additional Figure 3

6. Translational activity assessment – The possibility that observed EGFP signals are due to passively loaded EGFP protein has not been excluded. Appropriate controls, such as EVs from EGFP-overexpressing cells without mRNA packaging, are necessary.

Author response: We thank you for pointing out this possibility. To rule out the possibility that the observed EGFP signals resulted from passively loaded EGFP protein, we followed your suggestion. We measured EGFP protein levels in three types of EV samples and in cells after co-culture with each EV type using LC-MS/MS. The three EV groups were as follows: Expi293F-EVs (no plasmid transfection), EGFP-CD box EVs (transfected only with the EGFP-C/D box plasmid), and EN144-EV^{EGFP} (co-transfected with EN144-L7Ae and EGFP-C/D box plasmids). Although passive loading of EGFP protein was detected in the EN144-EV^{EGFP} and EGFP-C/D box EVs groups.. However, clear EGFP protein expression was only detected in cells co-cultured with EN144-EV^{EGFP}. These findings indicate that the EGFP signals observed in the cells mainly originated from active translation of EGFP mRNA delivered by EN144-EV^{EGFP}, rather than from passively carried EGFP protein within the EVs.

We revised the main text as follows: “To determine whether the observed EGFP signal originated from active mRNA translation rather than from pre-existing protein passively carried in EVs, LC-MS/MS was used to quantify EGFP levels in three types of EVs (Expi293F-EVs, EGFP-C/D box EVs, and EN144-EV^{EGFP}) as well as in recipient cells after co-culture (Supplementary Figure 13F, G). The results demonstrated that passive loading of EGFP protein was detected in both the EN144-EV^{EGFP} and EGFP-C/D box EVs groups (Supplementary Figure 13F). However, EGFP expression was significantly elevated in recipient cells co-cultured with EN144-EV^{EGFP} (Supplementary Figure 13G). These findings indicate that the EGFP signal in the recipient cells primarily resulted from active translation of EGFP mRNA delivered by EN144-EV^{EGFP}, rather than from pre-packaged EGFP protein in the EVs.” Please refer to lines 342-351.

Figure S13F

Figure S13F. Quantification of EGFP protein in EVs using LC-MS/MS. Three types of EVs were analyzed: Expi293F-EVs, EGFP-C/D box, and EN144-EV^{EGFP}, with each sample containing 2×10^{10} particles. The sources of the EVs are as follows: EN144-EV^{EGFP} was derived from the supernatant of cells co-transfected with EN144-L7Ae and EGFP-C/Dbox plasmids; EGFP-CD box was obtained from the supernatant of cells transfected with the EGFP-C/D box plasmid only; and Expi293F-EVs were isolated from the supernatant of untransfected cells.

Figure S13G

Figure S13G. Cells were incubated with EN144-EV^{EGFP} (1.0×10^{10} particles), and intracellular EGFP protein levels were measured using LC-MS/MS (n = 3 each group).

7. *Overstatement of results* – Several conclusions, including claims of high delivery efficiency and superior therapeutic potential, remain overstated. Many experiments rely on limited replicates (often n=3), with no robust statistical validation, reducing confidence in the reported findings.

Author response: We thank you for raising this critical point. We fully agree that the interpretation of our results should be more cautious, given the limited number of experimental replicates (typically n=3). Accordingly, we have thoroughly reviewed the entire manuscript and revised all statements that could be perceived as overstated.

(1) The original text: “Notably, our study demonstrates that for the EN144 scaffold, the GFP-Nanobody system yields the highest capacity and delivery efficiency. The findings demonstrate that the protein heterodimerization on the EN144 scaffold significantly boosted the loading of Cas9 in EVs,...” has been revised to: “Our study demonstrates that for the EN144 scaffold, the GFP-Nanobody system yields a **higher capacity and delivery efficiency than other tested dimerization systems**. The findings demonstrate that the protein heterodimerization on the EN144 scaffold **enabled the efficient** loading of Cas9 in EVs,...” Please refer to lines 693-695.

(2) The original text: “Comparative analysis revealed that EN144-EV^{hgp130} (80%) and EN144-EV^{mgp130} (80%) demonstrated significantly enhanced therapeutic efficacy in septic mice, outperforming sgp130 treatment at 10 μg (40%) and 1 μg (20%) doses (Figure 4L).” has been revised to: “Comparative analysis revealed that EN144-EV^{hgp130} (80%) and EN144-EV^{mgp130} (80%) demonstrated **an** enhanced therapeutic efficacy in septic mice, **with survival rates higher than those achieved by** sgp130 treatment at 10 μg (40%) and 1 μg (20%) doses (Figure 4L)”. Please refer to lines 472-473.

(3) The original text: “Notably, the EN144-EV^{hgp130-CAP} group showed the highest COL2A1 level and the lowest levels of IL-6 and MMP13, outperforming the EN144-EV^{hgp130} and HA groups (Figure 5L-O).” has been revised to: “The EN144-EV^{hgp130-CAP} group showed the highest COL2A1 level and the lowest levels of IL-6 and MMP13, **which were superior to the levels in** the EN144-EV^{hgp130} and HA groups (Figure 5L-O).” Please refer to lines 575-576.

(4) The original text: “Notably, ENPP1 emerged as an effective sorting protein, outperforming PTGFRN, already in clinical trials⁴⁵, and Lamp2b, widely used in preclinical studies⁴⁶.” has been revised to: “In our screening, ENPP1 emerged as an effective sorting protein, which showed higher efficiency than PTGFRN, already in clinical trials⁴⁵, and Lamp2b, widely used in preclinical studies⁴⁶.” Please refer to lines 629-630.

(5) The original text: “Consequently, EN144 augments EV loading efficiency and avoids the potential biological side effects of ENPP1.” has been revised to: “Consequently, EN144 is designed to enhance EV loading efficiency and to minimize the potential biological side effects of ENPP1.” Please refer to lines 656-657.

Reviewer #3 (Remarks to the Author):

The authors have responded well to my previous comments. They have added lots of additional experimental data and substantially strengthened the manuscript. I have no further concerns.

Author response: We thank you for your positive feedback and for acknowledging our efforts in strengthening the manuscript. We are pleased that the revisions and additional experimental data have addressed all your previous concerns.

REVIEWER COMMENTS

Reviewer #2 (Remarks to the Author):

Overall, the authors have made substantial revisions in response to our previous comments. Nevertheless, several technical issues remain and should be addressed to further strengthen the manuscript.

1. In several instances (e.g., Figure S4 and Figure S6B), His-MYO is used as a reference protein for quantifying EGFP levels in EVs. Using an unrelated protein as a normalization reference is unusual and may not accurately reflect EGFP incorporation. Please clarify the rationale for this choice and discuss its potential limitations. If possible, consider using a more relevant EV-associated or cargo-related reference.

Author response: We thank you for this important comment. We fully agree that using an unrelated protein (His-MYO) as a reference for quantification is not ideal, as the potential errors it may introduce could affect the accurate assessment of the number of EGFP fusion proteins. Therefore, after thorough discussion, we decided to revise the quantification method: instead of providing an exact number of EGFP fusion proteins based on the His-MYO standard curve (as in the previous version, shown in the left panel of Figure S4, the top panel of Figure S6B, and Figure S16A), we now use a relative semi-quantitative method. In the revised manuscript, we compare the relative changes in protein levels by measuring the band intensity (optical density) of the target bands in Western blots.

We have revised the statement at lines 178-181 to “Semi-quantitative protein analysis indicated that the levels of EGFP protein loaded into ENPP1-EVs and Rab7a-EVs were approximately three- and four-fold higher, respectively, than those in PTGFRN-EVs (Figure 1D, Supplementary Figure 4).”

The statement “EVs expressing the 144-amino-acid truncation, EN144 (EN144-EVs), exhibited a marked enhancement in cargo loading, a three-fold increase over ENPP1-EVs (Figure 1G, Supplementary Figure 6B).” has been updated in the manuscript. Please refer to lines 195-197.

The statement “Protein gray value analysis was performed to quantify engineered receptors on EVs, based on which we estimated that each EV displayed an average of 400 hgp130 receptors (Supplementary Figure 16A).” has been removed, as this conclusion was based on the outdated quantification method that has now been eliminated.

Correspondingly, the following figures were revised:

Supplementary Figure 4. Semi-quantitative analysis of EGFP protein in engineered EVs. EGFP protein levels in engineered EVs were detected by Western blot using an anti-GFP antibody. A consistent amount of EVs (3.0×10^9 particles per lane) was loaded for each sample. Grayscale intensity analysis of the target band was performed using

Image J software.

Figure 1D. Semi-quantitative analysis of EGFP protein in engineered EVs. The relative load levels of EGFP in engineered EVs were analyzed semi-quantitatively based on the band grayscale values shown in Figure S4.

Supplementary Figure 6B

Supplementary Figure 6B. FLAG-tagged protein in engineered EVs was semi-quantitatively analyzed by Western blot using an anti-FLAG antibody, with a standardized load of 3.0×10^9 EV particles per lane. Grayscale intensity analysis of the target band was performed using Image J software.

Figure 1G. Semi-quantitative analysis of FLAG protein in engineered EVs, as determined by band intensity measurements from Figure S6B.

2. For the gene-editing analysis in Table S5, red fluorescent cells were sorted prior to quantification, yet the reported editing efficiencies range from only 3.1% to 13.6%. In principle, sorting for fluorescent reporter-positive cells should yield a population in which editing efficiency approaches 100%. Please clarify how editing efficiency was defined and explain the discrepancy.

Author response: We thank you for raising this important point and for providing us with the opportunity to clarify this key methodological detail.

We introduced the stop-dsRed reporter plasmid into 293T cells using a lentivirus transfection system, resulting in a multiclonal cell line. Multiple (N) copies of the plasmid are randomly integrated into the genome of each cell. The EV-delivered gene-editing system removes the stop sequence, enabling the cell to express dsRed and emit red fluorescence. The values reported in Table S5 (3.1% to 13.6%) represent the frequency of indels as determined by TIDE analysis of the target site, which indicates that among the N copies of the plasmid, roughly 3.1-13.6% of the copies were edited.

To avoid confusion and strengthen the manuscript, we have explicitly stated in the revised text: “Editing efficiency was determined by sequencing analysis of the target locus in genomic DNA extracted from the FACS-sorted, red fluorescent-positive cell population. It represents the frequency of indel mutations among all the stop-dsRed copies in the fluorescent 293T cells.” Please refer to lines 905-908.

3. *Several methodological details are missing in Figure S17A and should be provided to ensure reproducibility. Specifically: What molecular weight cutoff was used for the dialysis device? Were EV samples concentrated after dialysis, and if so, how? How were samples normalized across lanes for western blotting?*

Author response: Thank you for pointing out these methodological details, which are crucial for ensuring experimental reproducibility. We have supplemented the Figure S17A legend with the following information: “...Unbound proteins were removed by dialysis (100 kDa MWCO) against PBS for 16 h, with buffer changes every 2 h. EVs were then recovered using EVtrap magnetic beads, and samples were normalized by total protein (20 µg per lane) for western blotting.” Please refer to lines 207-209 in the revised Supplementary document.

4. *In Additional Figure 3, a negative control using non-TNF α -binding EVs should be included to exclude the possibility of nonspecific binding.*

Author response: We sincerely thank you for the opportunity to improve our study and further strengthen its rigor. In accordance with your suggestion, we have performed additional experiments using EVs without TNFR1 as a negative control for the nonspecific binding. The detailed experimental procedure is as follows:

We evaluated the binding capacity of EVs with or without TNFR1 to TNF- α using an ELISA-based approach. Briefly, 1.00×10^{10} particles of EVs were immobilized in each well of the ELISA plate. Then, TNF- α (0 ng/mL, 0.2 ng/mL, 15 ng/mL, 25 ng/mL, 50 ng/mL, 100 ng/mL, 200 ng/mL) was added to bind to TNFR1 on the immobilized EVs. Subsequent detection was performed using HRP-conjugated detection antibodies and a chromogenic substrate, according to the protocol of a commercial TNF- α ELISA kit. Absorbance in each well was measured at 450 nm using a microplate reader.

The ELISA data showed that the amount of TNF- α bound to a fixed number of EV-TNFR1 particles increased in a dose-dependent manner with rising TNF- α concentrations. We did observe a baseline binding signal to EV-293F that does not overexpress TNFR1. However, the signal plateaus at an absorbance of around 0.13, which is significantly lower than that in the EV-TNFR1 group (~0.5). We reason that this baseline signal is due to 1) a baseline level of TNFR1 in EVs derived from 293F cells, and 2) a low level of non-specific binding between TNF

and EVs. Taken together, these data indicate that EV-TNFR1 preparations display functional TNFR1 on the surface of EVs, enabling binding to TNF- α with equivalent affinity as reported according to the measured EC₅₀ of around 12.5 ng/mL.

Additional Figure 1. ELISA characterization of TNF- α binding affinity to EVs with or without TNFR1 surface expression. The dose-response curves depict the binding kinetics, and the calculated half-maximal effective concentrations (EC₅₀) are compared between the two groups.

Reviewer #2 (Remarks to the Author):

The authors have adressed all concerns. Happy to accept the manuscript

Reply: We, the authors, are very grateful for your acceptance.